



# Global Carbon Budget 2022

Pierre Friedlingstein[1,2], Michael O'Sullivan[1], Matthew W. Jones[3], Robbie M. Andrew[4], Luke Gregor[5], Judith Hauck[6], Corinne Le Quéré[3], Ingrid T. Luijkx[7], Are Olsen[8,9,], Glen P. Peters[4], Wouter Peters[7,10], Julia Pongratz[11,12,] Clemens Schwingshackl[11], Stephen Sitch[1], Josep G. Canadell[13], Philippe Ciais[14], Robert B. Jackson[15], Simone Alin[16], Ramdane Alkama[17], Almut Arneth[18], Vivek K. Arora[19], Nicholas R. Bates[20,21], Meike Becker[8,9], Nicolas Bellouin[22], Henry C. Bittig[23], Laurent Bopp[2], Frédéric Chevallier[14], Louise P. Chini[24], Margot Cronin[25], Wiley Evans[26], Stefanie Falk[11], Richard A. Feely[16], Thomas Gasser[27], Marion Gehlen[14], Thanos Gkritzalis[28], Lucas Gloege[29,30], Giacomo Grassi[17], Nicolas Gruber[5], Özgür Gürses[6], Ian Harris[31], Matthew Hefner[32,33], Richard A. Houghton[34], George C. Hurtt[24], Yosuke Iida[35], Tatiana Ilyina[12], Atul K. Jain[36], Annika Jersild[12], Koji Kadono[35], Etsushi Kato[37], Daniel Kennedy[38], Kees Klein Goldewijk[39], Jürgen Knauer[40,41], Jan Ivar Korsbakken[4], Peter Landschützer[12,28], Nathalie Lefèvre[42,] Keith Lindsay[43], Junjie Liu[44], Zhu Liu[45], Gregg Marland[32,33,] Nicolas Mayot[3], Matthew J. McGrath[14], Nicolas Metzl[42], Natalie M. Monacci[46], David R. Munro[47,48], Shin-Ichiro Nakaoka[49], Yosuke Niwa[49,50], Kevin O´Brien[51,16], Tsuneo Ono[52], Paul I. Palmer[53,54], Naiqing Pan[55,56], Denis Pierrot[57], Katie Pocock[26], Benjamin Poulter[58], Laure Resplandy[59], Eddy Robertson[60], Christian Rödenbeck[61], Carmen Rodriguez[62,] Thais M. Rosan[1], Jörg Schwinger[63,9], Roland Séférian[64], Jamie D. Shutler[1], Ingunn Skjelvan[63,9], Tobias Steinhoff[65], Qing Sun[66], Adrienne J. Sutton[16], Colm Sweeney[48], Shintaro Takao[49], Toste Tanhua[65], Pieter P. Tans[67], Xiangjun Tian[68], Hanqin Tian[56], Bronte Tilbrook[69,70], Hiroyuki Tsujino[50], Francesco Tubiello[71], Guido R. van der Werf[72], Anthony P. Walker[73], Rik Wanninkhof[57], Chris Whitehead[74], Anna Willstrand Wranne[75], Rebecca Wright[3], Wenping Yuan[76], Chao Yue[77], Xu Yue[78], Sönke Zaehle[61], Jiye Zeng[49], Bo Zheng[79]

[1] Faculty of Environment, Science and Economy, University of Exeter, Exeter EX4 4QF, UK
[2] Laboratoire de Météorologie Dynamique / Institut Pierre-Simon Laplace, CNRS, Ecole Normale Supérieure / Université PSL, Sorbonne Université, Ecole Polytechnique, Paris, France
[3] Tyndall Centre for Climate Change Research, School of Environmental Sciences, University of East Anglia, Norwich Research Park, Norwich NR4 7TJ, UK
[4] CICERO Center for International Climate Research, Oslo 0349, Norway
[5] Environmental Physics Group, ETH Zürich, Institute of Biogeochemistry and Pollutant Dynamics and Center for Climate Systems Modeling (C2SM), Zurich, Switzerland
[6] Alfred-Wegener-Institut Helmholtz-Zentum für Polar- und Meeresforschung, Postfach 120161, 27515 Bremerhaven, Germany
[7] Wageningen University, Environmental Sciences Group, P.O. Box 47, 6700AA, Wageningen, The Netherlands
[8] Geophysical Institute, University of Bergen, Bergen, Norway
[9] Bjerknes Centre for Climate Research, Bergen, Norway
[10] University of Groningen, Centre for Isotope Research, Groningen, The Netherlands
[11] Ludwig-Maximilians-Universität Munich, Luisenstr. 37, 80333 München, Germany
[12] Max Planck Institute for Meteorology, Hamburg, Germany
[13] CSIRO Oceans and Atmosphere, Canberra, ACT 2101, Australia
[14] Laboratoire des Sciences du Climat et de l'Environnement, LSCE/IPSL, CEA-CNRS-UVSQ, Université Paris-Saclay, F-91191 Gif-sur-Yvette, France
[15] Department of Earth System Science, Woods Institute for the Environment, and Precourt Institute for Energy, Stanford University, Stanford, CA 94305–2210, United States of America
[16] National Oceanic & Atmospheric Administration, Pacific Marine Environmental Laboratory (NOAA/PMEL), 7600 Sand Point Way NE, Seattle, WA 98115, USA
[17] Joint Research Centre, European Commission, Ispra, Italy
[18] Karlsruhe Institute of Technology, Institute of Meteorology and Climate Research/Atmospheric Environmental Research, 82467 Garmisch-Partenkirchen, Germany



[19] Canadian Centre for Climate Modelling and Analysis, Climate Research Division, Environment
and Climate Change Canada, Victoria, BC, Canada
[20] Bermuda Institute of Ocean Sciences (BIOS), 17 Biological Lane, St. Georges, GE01, Bermuda
[21] Department of Ocean and Earth Science, University of Southampton, European Way,
Southampton, SO14 3ZH, UK
[22] Department of Meteorology, University of Reading, Reading, UK
[23] Leibniz Institute for Baltic Sea Research Warnemuende (IOW), Seestrasse 15; 18119 Rostock,
Germany
[24]Department of Geographical Sciences, University of Maryland, College Park, Maryland 20742,
USA
[25] Marine Institute, Galway, Ireland
[26] Hakai Institute, Heriot Bay, BC, Canada
[27] International Institute for Applied Systems Analysis (IIASA), Schlossplatz 1
A-2361 Laxenburg, Austria
[28] Flanders Marine Institute (VLIZ), InnovOceanSite, Wandelaarkaai 7, 8400 Ostend, Belgium
[29] Lamont-Doherty Earth Observatory and Department of Earth and Environmental Sciences,
Columbia University, New York, NY, USA
[30] Open Earth Foundation, Marina del Rey, CA, USA
[31] NCAS-Climate, Climatic Research Unit, School of Environmental Sciences, University of East
Anglia, Norwich Research Park, Norwich, NR4 7TJ, UK
[32] Research Institute for Environment, Energy, and Economics, Appalachian State University,
Boone, North Carolina, USA
[33] Department of Geological and Environmental Sciences, Appalachian State University, Boone,
North Carolina, USA
[34] Woodwell Climate Research Center, Falmouth, MA 02540, USA
[35] Atmosphere and Ocean Department, Japan Meteorological Agency, Minato-Ku, Tokyo 105-
8431, Japan
[36] Department of Atmospheric Sciences, University of Illinois, Urbana, IL 61821, USA
[37] Institute of Applied Energy (IAE), Minato-ku, Tokyo 105-0003, Japan
[38] National Center for Atmospheric Research, Climate and Global Dynamics, Terrestrial Sciences
Section, Boulder, CO 80305, USA
[39] Utrecht University, Faculty of Geosciences, Department IMEW, Copernicus Institute of
Sustainable Development, Heidelberglaan 2, P.O. Box 80115, 3508 TC, Utrecht, the Netherlands
[40] Hawkesbury Institute for the Environment, Western Sydney University, Penrith, New South
Wales, Australia
[41] Climate Science Centre, CSIRO Oceans and Atmosphere, Canberra, ACT, Australia
[42] LOCEAN/IPSL laboratory, Sorbonne Université, CNRS/IRD/MNHN, Paris, France
[43] National Center for Atmospheric Research, Climate and Global Dynamics, Oceanography
Section, Boulder, CO 80305, USA
[44] Jet Propulsion Laboratory, California Institute of Technology, Pasadena, CA, USA
[45] Department of Earth System Science, Tsinghua University, Beijing, China
[46] University of Alaska Fairbanks, College of Fisheries and Ocean Sciences, PO Box 757220,
Fairbanks, AK, USA
[47] Cooperative Institute for Research in Environmental Sciences, University of Colorado, Boulder,
CO, 80305, USA
[48] National Oceanic & Atmospheric Administration/Global Monitoring Laboratory (NOAA/GML),
Boulder, CO, 80305, USA
[49] Earth System Division, National Institute for Environmental Studies (NIES), 16-2 Onogawa,
Tsukuba Ibaraki, 305-8506, Japan
[50] Meteorological Research Institute, 1-1 Nagamine, Tsukuba, Ibaraki, 305-0052 Japan
[51] Cooperative Institute for Climate, Ocean and Ecosystem Studies (CICOES), University of
Washington, Seattle, WA, USA
[52] Japan Fisheries Research and Education Agency, 2-12-4 Fukuura, Kanazawa-Ku, Yokohama 236-
8648, Japan
[53] National Centre for Earth Observation, University of Edinburgh, UK
[54] School of GeoSciences, University of Edinburgh, UK
[55] College of Forestyry, Wildlife and Environment, Auburn University, Auburn, AL 36849, USA
[56] Schiller Institute for Integrated Science and Society, Department of Earth and Environmental
Sciences, Boston College, Chestnut Hill, MA 02467, USA



[57] National Oceanic & Atmospheric Administration/Atlantic Oceanographic & Meteorological
Laboratory (NOAA/AOML), Miami, FL 33149, USA
[58] NASA Goddard Space Flight Center, Biospheric Sciences Laboratory, Greenbelt, Maryland
20771, USA
[59] Princeton University, Department of Geosciences and Princeton Environmental Institute,
Princeton, NJ, USA
[60] Met Office Hadley Centre, FitzRoy Road, Exeter EX1 3PB, UK
[61] Max Planck Institute for Biogeochemistry, P.O. Box 600164, Hans-Knöll-Str. 10, 07745 Jena,
Germany
[62] University of Miami, RSMAS, 4600 Rickenbacker Causeway, Miami, FL 33149, USA
[63] NORCE Norwegian Research Centre, Jahnebakken 5, 5007 Bergen, Norway
[64] CNRM, Université de Toulouse, Météo-France, CNRS, Toulouse, France
[65] GEOMAR Helmholtz Centre for Ocean Research Kiel, Düsternbrooker Weg 20, 24105 Kiel,
Germany
[66] Climate and Environmental Physics, Physics Institute and Oeschger Centre for Climate Change
Research, University of Bern, Bern, Switzerland
[67] National Oceanic & Atmospheric Administration, Earth System Research Laboratory
(NOAA ESRL), Boulder, CO 80305, USA
[68] Institute of Tibetan Plateau Research, Chinese Academy of Sciences, Beijing 100101, China
[69] CSIRO Oceans and Atmosphere, PO Box 1538, Hobart, Tasmania 7001, Australia
[70] Australian Antarctic Partnership Program, University of Tasmania, Hobart, Australia
[71] Statistics Division, Food and Agriculture Organization of the United Nations, Via Terme di
Caracalla, Rome 00153, Italy
[72] Department of Earth sciences, Faculty of Science, Vrije Universiteit, Amsterdam, the
Netherlands
[73] Environmental Sciences Division and Climate Change Science Institute, Oak Ridge National
Laboratory, Oak Ridge, TN, 37831, USA
[74] Sitka Tribe of Alaska, 456 Katlian Street, Sitka, Alaska 99835, USA
[75] Swedish Meteorological and Hydrological Institute, Sven Källfeltsgata 15, 426 68 Västra
Frölunda, Sweden
[76] School of Atmospheric Sciences, Sun Yat-sen University, Zhuhai, Guangdong 510245, China.
[77] Institute of Soil and Water Conservation, Northwest A&F University, Yangling, Shaanxi 712100,
P.R. China
[78] School of Environmental Science and Engineering, Nanjing University of Information Science
and Technology (NUIST), China
[79] Institute of Environment and Ecology, Tsinghua Shenzhen International Graduate School,
Tsinghua University, Shenzhen 518055, China
*Correspondence to*: Pierre Friedlingstein (p.friedlingstein@exeter.ac.uk)



**Abstract**
Accurate assessment of anthropogenic carbon dioxide (CO2) emissions and their redistribution among the
atmosphere, ocean, and terrestrial biosphere in a changing climate is critical to better understand the global
carbon cycle, support the development of climate policies, and project future climate change. Here we describe
and synthesise data sets and methodology to quantify the five major components of the global carbon budget
and their uncertainties. Fossil CO2 emissions (EFOS) are based on energy statistics and cement production data,
while emissions from land-use change (ELUC), mainly deforestation, are based on land-use and land-use
change data and bookkeeping models. Atmospheric CO2 concentration is measured directly, and its growth rate
(GATM) is computed from the annual changes in concentration. The ocean CO2 sink (SOCEAN) is estimated
with global ocean biogeochemistry models and observation-based data-products. The terrestrial CO2 sink
(SLAND) is estimated with dynamic global vegetation models. The resulting carbon budget imbalance (BIM),
the difference between the estimated total emissions and the estimated changes in the atmosphere, ocean, and
terrestrial biosphere, is a measure of imperfect data and understanding of the contemporary carbon cycle. All
uncertainties are reported as $\pm 1\sigma$.
For the year 2021, EFOS increased by 5.1% relative to 2020, with fossil emissions at $10.1 \pm 0.5$ GtC yr-1 ($9.9 \pm$
0.5 GtC yr-1 when the cement carbonation sink is included), ELUC was $1.1 \pm 0.7$ GtC yr-1, for a total
anthropogenic CO2 emission of $11.1 \pm 0.8$ GtC yr-1 ($40.8 \pm 2.9$ GtCO2). Also, for 2021, GATM was $5.2 \pm 0.2$
GtC yr-1 ($2.5 \pm 0.1$ ppm yr-1), SOCEAN was $2.9 \pm 0.4$ GtC yr-1 and SLAND was $3.5 \pm 0.9$ GtC yr-1, with a
BIM of -0.6 GtC yr-1 (i.e. total estimated sources too low or sinks too high). The global atmospheric CO2
concentration averaged over 2021 reached $414.71 \pm 0.1$ ppm. Preliminary data for 2022, suggest an increase in
EFOS relative to 2021 of +1.1% (0% to 1.7%) globally, and atmospheric CO2 concentration reaching 417.3
ppm, more than 50% above pre-industrial level. Overall, the mean and trend in the components of the global
carbon budget are consistently estimated over the period 1959-2021, but discrepancies of up to 1 GtC yr$^{-1}$ persist
for the representation of annual to semi-decadal variability in CO$_2$ fluxes. Comparison of estimates from
multiple approaches and observations shows: (1) a persistent large uncertainty in the estimate of land-use
changes emissions, (2) a low agreement between the different methods on the magnitude of the land CO$_2$ flux in
the northern extra-tropics, and (3) a discrepancy between the different methods on the strength of the ocean sink
over the last decade. This living data update documents changes in the methods and data sets used in this new
global carbon budget and the progress in understanding of the global carbon cycle compared with previous
publications of this data set (Friedlingstein et al., 2022a; Friedlingstein et al., 2020; Friedlingstein et al., 2019;
Le Quéré et al., 2018b, 2018a, 2016, 2015b, 2015a, 2014, 2013). The data presented in this work are available at
https://doi.org/10.18160/GCP-2022 (Friedlingstein et al., 2022b).




### Executive Summary

**Global fossil $CO_2$ emissions (excluding cement carbonation) further increased in 2022, being now slightly above their pre-COVID19 pandemic level.** The 2021 emission increase was 0.46 GtC yr$^{-1}$ (1.7 GtCO$_2$ yr$^{-1}$), bringing 2021 emissions to 10.1 ± 0.5 GtC yr$^{-1}$ (36.9 ± 1.8 GtCO$_2$ yr$^{-1}$), slightly below the emissions level of 2019. Preliminary estimates based on data available suggest fossil $CO_2$ emissions continued to increase in 2022, by 1.1% relative to 2021 (0% to 1.7%), bringing emissions at 10.2 GtC yr$^{-1}$ (37.3 GtCO$_2$ yr$^{-1}$), slightly above the 2019 level (10.1 ± 0.5 GtC yr$^{-1}$, 37.0 ± 1.8 GtCO$_2$ yr$^{-1}$). Emissions from coal, oil, and gas in 2022 are expected to be above their 2021 levels (by 0.8%, 2.2% and 1.1% respectively). Regionally, emissions in 2022 are expected to have been decreasing by 1.5% in China (3.0 GtC, 11.1 GtCO$_2$), and 1% in the European Union (0.8 GtC, 2.8 GtCO$_2$), but increasing by 1.6% in the United States (1.4 GtC, 5.1 GtCO$_2$), 5.6% in India (0.8 GtC, 2.9 GtCO$_2$) and 2.5% for the rest of the world (4.2 GtC, 15.5 GtCO$_2$).

**Fossil $CO_2$ emissions decreased in 24 countries during the decade 2010-2019.** Altogether, these 24 countries contribute to about 2.4 GtC yr$^{-1}$ (8.8 GtCO$_2$) fossil fuel $CO_2$ emissions over the last decade, only about one quarter of world $CO_2$ fossil emissions.

**Global $CO_2$ emissions from land-use, land-use change, and forestry (LUC) averaged at 1.2 ± 0.7 GtC yr$^{-1}$ (4.5 ± 2.6 GtCO$_2$ yr$^{-1}$) for the 2012-2021 period with a preliminary projection for 2022 of 1.0 ± 0.7 GtC yr$^{-1}$ (3.6 ± 2.6 GtCO$_2$ yr$^{-1}$). A small decrease over the past two decades is not robust given the large model uncertainty.** Deforestation emissions remain high at 1.8 ± 0.4 GtC yr$^{-1}$ over the 2012-2021 period, highlighting a substantial mitigation potential for emissions reductions. Sequestration of 0.9 ± 0.3 GtC yr$^{-1}$ through re-/afforestation and forestry offsets one half of the deforestation emissions. Emissions from other transitions and from peat drainage and peat fire add further, small contributions. The highest emitters during 1959-2021 in descending order were Brazil, Indonesia, and the Democratic Republic of the Congo, with these 3 countries contributing more than half of the global total land-use emissions.

**The remaining carbon budget for a 50% likelihood to limit global warming to 1.5°C, 1.7°C and 2°C has respectively reduced to 105 GtC (380 GtCO$_2$), 200 GtC (730 GtCO$_2$) and 335 GtC (1230 GtCO$_2$) from the beginning of 2023, equivalent to 9, 18 and 30 years, assuming 2022 emissions levels.** Total anthropogenic emissions were 11.1 GtC yr$^{-1}$ (40.8 GtCO$_2$ yr-1) in 2021, with a preliminary estimate of 11.1 GtC yr$^{-1}$ (40.9 GtCO2 yr$^{-1}$) for 2022. The remaining carbon budget to keep global temperatures below these climate targets has shrunk by 33 GtC (121 GtCO$_2$) since the release of the IPCC AR6 Working Group 1 assessment in 2019. Reaching zero $CO_2$ emissions by 2050 entails cutting total anthropogenic $CO_2$ emissions by about 0.4 GtC (1.4 GtCO$_2$) each year on average, comparable to the decrease during 2020, highlighting the scale of the action needed.

**The concentration of $CO_2$ in the atmosphere is set to reach 417.3 ppm in 2022, 51% above pre-industrial levels.** The atmospheric $CO_2$ growth was 5.2 ± 0.02 GtC yr$^{-1}$ during the decade 2012-2021 (48% of total $CO_2$ emissions) with a preliminary 2022 growth rate estimate of around 5.5 GtC yr$^{-1}$ (2.6 ppm).

**The ocean $CO_2$ sink resumed a more rapid growth in the past decade after low or no growth during the 1991-2002 period.** However, the growth of the ocean $CO_2$ sink in the past decade has an uncertainty of a factor





of three, with estimates based on data products and estimates based on models showing an ocean sink trend of
+0.7 GtC yr$^{-1}$ decade$^{-1}$ and +0.2 GtC yr$^{-1}$ decade$^{-1}$ since 2010, respectively. The discrepancy in the trend
originates from all latitudes but is largest in the Southern Ocean. The ocean $CO_2$ sink was 2.9 ± 0.4 GtC yr$^{-1}$
during the decade 2011-2020 (26% of total $CO_2$ emissions), with a similar preliminary estimate of 2.9 GtC yr$^{-1}$
for 2022.
**The land $CO_2$ sink continued to increase during the 2012-2021 period primarily in response to increased**
**atmospheric $CO_2$, albeit with large interannual variability.** The land $CO_2$ sink was 3.1 ± 0.6 GtC yr$^{-1}$
during the 2012-2021 decade (29% of total $CO_2$ emissions), 0.4 GtC yr$^{-1}$ larger than during the previous decade
(2000-2009), with a preliminary 2022 estimate of around 3.4 GtC yr$^{-1}$. Year to year variability in the land sink is
about 1 GtC yr$^{-1}$, making small annual changes in anthropogenic emissions hard to detect in global atmospheric
$CO_2$ concentration.



## 1    Introduction


The concentration of carbon dioxide ($CO_2$) in the atmosphere has increased from approximately 277 parts per
million (ppm) in 1750 (Joos and Spahni, 2008), the beginning of the Industrial Era, to $414.7 \pm 0.1$ ppm in 2021
(Dlugokencky and Tans, 2022); Figure 1). The atmospheric $CO_2$ increase above pre-industrial levels was,
initially, primarily caused by the release of carbon to the atmosphere from deforestation and other land-use
change activities (Canadell et al., 2021). While emissions from fossil fuels started before the Industrial Era, they
became the dominant source of anthropogenic emissions to the atmosphere from around 1950 and their relative
share has continued to increase until present. Anthropogenic emissions occur on top of an active natural carbon
cycle that circulates carbon between the reservoirs of the atmosphere, ocean, and terrestrial biosphere on time
scales from sub-daily to millennia, while exchanges with geologic reservoirs occur at longer timescales (Archer
et al., 2009).
The global carbon budget (GCB) presented here refers to the mean, variations, and trends in the perturbation of
$CO_2$ in the environment, referenced to the beginning of the Industrial Era (defined here as 1750). This paper
describes the components of the global carbon cycle over the historical period with a stronger focus on the
recent period (since 1958, onset of atmospheric $CO_2$ measurements), the last decade (2012-2021), the last year
(2021) and the current year (2022). Finally, it provides cumulative emissions from fossil fuels and land-use
change since the year 1750, the pre-industrial period; and since the year 1850, the reference year for historical
simulations in IPCC AR6 (Eyring et al., 2016).
We quantify the input of $CO_2$ to the atmosphere by emissions from human activities, the growth rate of
atmospheric $CO_2$ concentration, and the resulting changes in the storage of carbon in the land and ocean
reservoirs in response to increasing atmospheric $CO_2$ levels, climate change and variability, and other
anthropogenic and natural changes (Figure 2). An understanding of this perturbation budget over time and the
underlying variability and trends of the natural carbon cycle is necessary to understand the response of natural
sinks to changes in climate, $CO_2$ and land-use change drivers, and to quantify emissions compatible with a given
climate stabilisation target.
The components of the $CO_2$ budget that are reported annually in this paper include separate and independent
estimates for the $CO_2$ emissions from (1) fossil fuel combustion and oxidation from all energy and industrial
processes; also including cement production and carbonation ($E_{FOS}$; GtC yr$^{-1}$) and (2) the emissions resulting
from deliberate human activities on land, including those leading to land-use change ($E_{LUC}$; GtC yr$^{-1}$); and their
partitioning among (3) the growth rate of atmospheric $CO_2$ concentration ($G_{ATM}$; GtC yr$^{-1}$), and the uptake of
$CO_2$ (the '$CO_2$ sinks') in (4) the ocean ($S_{OCEAN}$; GtC yr$^{-1}$) and (5) on land ($S_{LAND}$; GtC yr$^{-1}$). The $CO_2$ sinks as
defined here conceptually include the response of the land (including inland waters and estuaries) and ocean
(including coastal and marginal seas) to elevated $CO_2$ and changes in climate and other environmental
conditions, although in practice not all processes are fully accounted for (see Section 2.7). Global emissions and
their partitioning among the atmosphere, ocean and land are in balance in the real world. Due to the combination
of imperfect spatial and/or temporal data coverage, errors in each estimate, and smaller terms not included in our
budget estimate (discussed in Section 2.7), the independent estimates (1) to (5) above do not necessarily add up
to zero. We therefore (a) additionally assess a set of global atmospheric inversion system results that by design
close the global carbon balance (see Section 2.6), and (b) estimate a budget imbalance ($B_{IM}$), which is a measure



of the mismatch between the estimated emissions and the estimated changes in the atmosphere, land and ocean,
as follows:
$$B_{IM} = E_{FOS} + E_{LUC} - (G_{ATM} + S_{OCEAN} + S_{LAND}) \qquad (1)$$
$G_{ATM}$ is usually reported in ppm yr$^{-1}$, which we convert to units of carbon mass per year, GtC yr$^{-1}$, using 1 ppm
= 2.124 GtC (Ballantyne et al., 2012; Table 1). All quantities are presented in units of gigatonnes of carbon
(GtC, $10^{15}$ gC), which is the same as petagrams of carbon (PgC; Table 1). Units of gigatonnes of $CO_2$ (or billion
tonnes of $CO_2$) used in policy are equal to 3.664 multiplied by the value in units of GtC.
We also quantify $E_{FOS}$ and $E_{LUC}$ by country, including both territorial and consumption-based accounting for
$E_{FOS}$ (see Section 2), and discuss missing terms from sources other than the combustion of fossil fuels (see
Section 2.7).
The global $CO_2$ budget has been assessed by the Intergovernmental Panel on Climate Change (IPCC) in all
assessment reports (Prentice et al., 2001; Schimel et al., 1995; Watson et al., 1990; Denman et al., 2007; Ciais et
al., 2013; Canadell et al., 2021), and by others (e.g. Ballantyne et al., 2012). The Global Carbon Project (GCP,
www.globalcarbonproject.org, last access: 25 September 2022) has coordinated this cooperative community
effort for the annual publication of global carbon budgets for the year 2005 (Raupach et al., 2007; including
fossil emissions only), year 2006 (Canadell et al., 2007), year 2007 (GCP, 2008), year 2008 (Le Quéré et al.,
2009), year 2009 (Friedlingstein et al., 2010), year 2010 (Peters et al., 2012b), year 2012 (Le Quéré et al., 2013;
Peters et al., 2013), year 2013 (Le Quéré et al., 2014), year 2014 (Le Quéré et al., 2015a; Friedlingstein et al.,
2014), year 2015 (Jackson et al., 2016; Le Quéré et al., 2015b), year 2016 (Le Quéré et al., 2016), year 2017 (Le
Quéré et al., 2018a; Peters et al., 2017), year 2018 (Le Quéré et al., 2018b; Jackson et al., 2018),  year 2019
(Friedlingstein et al., 2019; Jackson et al., 2019; Peters et al., 2020), year 2020 (Friedlingstein et al.,  2020; Le
Quéré et al., 2021) and more recently the year 2021 (Friedlingstein et al., 2022a; Jackson et al., 2022). Each of
these papers updated previous estimates with the latest available information for the entire time series.
We adopt a range of ±1 standard deviation (σ) to report the uncertainties in our estimates, representing a
likelihood of 68% that the true value will be within the provided range if the errors have a Gaussian distribution,
and no bias is assumed. This choice reflects the difficulty of characterising the uncertainty in the $CO_2$ fluxes
between the atmosphere and the ocean and land reservoirs individually, particularly on an annual basis, as well
as the difficulty of updating the $CO_2$ emissions from land-use change. A likelihood of 68% provides an
indication of our current capability to quantify each term and its uncertainty given the available information.
The uncertainties reported here combine statistical analysis of the underlying data, assessments of uncertainties
in the generation of the data sets, and expert judgement of the likelihood of results lying outside this range. The
limitations of current information are discussed in the paper and have been examined in detail elsewhere
(Ballantyne et al., 2015; Zscheischler et al., 2017). We also use a qualitative assessment of confidence level to
characterise the annual estimates from each term based on the type, amount, quality, and consistency of the
evidence as defined by the IPCC (Stocker et al., 2013).
This paper provides a detailed description of the data sets and methodology used to compute the global carbon
budget estimates for the industrial period, from 1750 to 2022, and in more detail for the period since 1959. This
paper is updated every year using the format of 'living data' to keep a record of budget versions and the changes

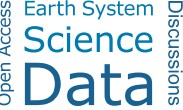

in new data, revision of data, and changes in methodology that lead to changes in estimates of the carbon
budget. Additional materials associated with the release of each new version will be posted at the Global Carbon
Project (GCP) website (http://www.globalcarbonproject.org/carbonbudget, last access: 25 September 2022),
with fossil fuel emissions also available through the Global Carbon Atlas (http://www.globalcarbonatlas.org,
last access: 25 September 2022). All underlying data used to produce the budget can also be found at
https://globalcarbonbudget.org/ (last access: 25 September 2022). With this approach, we aim to provide the
highest transparency and traceability in the reporting of $CO_2$, the key driver of climate change.

## 2    Methods

Multiple organisations and research groups around the world generated the original measurements and data used
to complete the global carbon budget. The effort presented here is thus mainly one of synthesis, where results
from individual groups are collated, analysed, and evaluated for consistency. We facilitate access to original
data with the understanding that primary data sets will be referenced in future work (see Table 2 for how to cite
the data sets). Descriptions of the measurements, models, and methodologies follow below, and detailed
descriptions of each component are provided elsewhere.
This is the 17th version of the global carbon budget and the 11th revised version in the format of a living data
update in Earth System Science Data. It builds on the latest published global carbon budget of Friedlingstein et
al. (2022a). The main changes are: the inclusion of (1) data to year 2021 and a projection for the global carbon
budget for year 2022; (2) the inclusion of country level estimates of $E_{LUC}$; (3) a process-based decomposition of
$E_{LUC}$ into its main components (deforestation, carbon uptake on forests, emissions from organic soils, and net
flux from other transitions).
The main methodological differences between recent annual carbon budgets (2018-2022) are summarised in
Table 3 and previous changes since 2006 are provided in Table A7.

### 2.1    Fossil $CO_2$ emissions ($E_{FOS}$)

#### 2.1.1    Historical period 1850-2021

The estimates of global and national fossil $CO_2$ emissions ($E_{FOS}$) include the oxidation of fossil fuels through
both combustion (e.g., transport, heating) and chemical oxidation (e.g. carbon anode decomposition in
aluminium refining) activities, and the decomposition of carbonates in industrial processes (e.g. the production
of cement). We also include $CO_2$ uptake from the cement carbonation process. Several emissions sources are not
estimated or not fully covered: coverage of emissions from lime production are not global, and decomposition of
carbonates in glass and ceramic production are included only for the "Annex 1" countries of the United Nations
Framework Convention on Climate Change (UNFCCC) for lack of activity data. These omissions are
considered to be minor. Short-cycle carbon emissions - for example from combustion of biomass - are not
included here but are accounted for in the $CO_2$ emissions from land use (see section 2.2).
Our estimates of fossil $CO_2$ emissions are derived using the standard approach of activity data and emission
factors, relying on data collection by many other parties. Our goal is to produce the best estimate of this flux,
and we therefore use a prioritisation framework to combine data from different sources that have used different
methods, while being careful to avoid double counting and undercounting of emissions sources. The CDIAC-FF



emissions dataset, derived largely from UN energy data, forms the foundation, and we extend emissions to year
Y-1 using energy growth rates reported by BP. We then proceed to replace estimates using data from what we
consider to be superior sources, for example Annex 1 countries' official submissions to the UNFCCC. All data
points are potentially subject to revision, not just the latest year. For full details see Andrew and Peters (2021).
Other estimates of global fossil $CO_2$ emissions exist, and these are compared by Andrew (2020a). The most
common reason for differences in estimates of global fossil $CO_2$ emissions is a difference in which emissions
sources are included in the datasets. Datasets such as those published by the energy company BP, the US Energy
Information Administration, and the International Energy Agency's '$CO_2$ emissions from fuel combustion' are
all generally limited to emissions from combustion of fossil fuels. In contrast, datasets such as PRIMAP-hist,
CEDS, EDGAR, and GCP's dataset aim to include all sources of fossil $CO_2$ emissions. See Andrew (2020a) for
detailed comparisons and discussion.
Cement absorbs $CO_2$ from the atmosphere over its lifetime, a process known as 'cement carbonation'. We
estimate this $CO_2$ sink as the average of two studies in the literature (Cao et al., 2020; Guo et al., 2021). Both
studies use the same model, developed by Xi et al. (2016), with different parameterisations and input data. The
Global Cement and Concrete Association reports a much lower carbonation rate, but this is based on the highly
conservative assumption of 0% mortar (GCCA, 2021). Since carbonation is a function of both current and
previous cement production, we extend these estimates by one year to 2021 by using the growth rate derived
from the smoothed cement emissions (10-year smoothing) fitted to the carbonation data.
We use the Kaya Identity for a simple decomposition of $CO_2$ emissions into the key drivers (Raupach et al.,
2007). While there are variations (Peters et al 2017), we focus here on a decomposition of $CO_2$ emissions into
population, GDP per person, energy use per GDP, and $CO_2$ emissions per energy. Multiplying these individual
components together returns the $CO_2$ emissions. Using the decomposition, it is possible to attribute the change
in $CO_2$ emissions to the change in each of the drivers. This method gives a first order understanding of what
causes $CO_2$ emissions to change each year.

### 376  2.1.2  2022 projection

We provide a projection of global $CO_2$ emissions in 2022 by combining separate projections for China, USA,
EU, India, and for all other countries combined. The methods are different for each of these. For China we
combine monthly fossil fuel production data from the National Bureau of Statistics, import/export data from the
Customs Administration, and monthly coal consumption estimates from SX Coal (2022), giving us partial data
for the growth rates to date of natural gas, petroleum, and cement, and of the consumption itself for raw coal.
We then use a regression model to project full-year emissions based on historical observations. For the USA our
projection is taken directly from the Energy Information Administration's (EIA) Short-Term Energy Outlook
(EIA, 2022), combined with the year-to-date growth rate of cement clinker production. For the EU we use
monthly energy data from Eurostat to derive estimates of monthly $CO_2$ emissions through July, with coal
emissions extended through August using a statistical relationship with reported electricity generation from coal
and other factors. Given the very high uncertainty in European energy markets in 2022, we forego our usual
history-based projection techniques and use instead the year-to-date growth rate as the full-year growth rate for
both coal and natural gas. EU emissions from oil are derived using the EIA's projection of oil consumption for
Europe. EU cement emissions are based on available year-to-date data from three of the largest producers,



Germany, Poland, and Spain. India's projected emissions are derived from estimates through July (August for
oil) using the methods of Andrew (2020b) and extrapolated assuming normal seasonal patterns. Emissions for
the rest of the world are derived using projected growth in economic production from the IMF (2022) combined
with extrapolated changes in emissions intensity of economic production. More details on the $E_{FOS}$ methodology
and its 2022 projection can be found in Appendix C.1.
## 2.2 CO$_2$ emissions from land-use, land-use change and forestry ($E_{LUC}$)
### 2.2.1 Historical Period
The net $CO_2$ flux from land-use, land-use change and forestry ($E_{LUC}$, called land-use change emissions in the
rest of the text) includes $CO_2$ fluxes from deforestation, afforestation, logging and forest degradation (including
harvest activity), shifting cultivation (cycle of cutting forest for agriculture, then abandoning), and regrowth of
forests following wood harvest or abandonment of agriculture. Emissions from peat burning and drainage are
added from external datasets. Compared to our earlier assessments, this year we include spatially explicit
information also for peat drainage and combine three independent datasets for peat drainage.
Three bookkeeping approaches (updated estimates each of BLUE (Hansis et al., 2015), OSCAR (Gasser et al.,
2020), and H&N2017 (Houghton and Nassikas, 2017)) were used to quantify gross sources and sinks and the
resulting net $E_{LUC}$. Uncertainty estimates were derived from the Dynamic Global vegetation Models (DGVMs)
ensemble for the time period prior to 1960, using for the recent decades an uncertainty range of $\pm0.7$ GtC yr$^{-1}$,
which is a semi-quantitative measure for annual and decadal emissions and reflects our best value judgement
that there is at least 68% chance ($\pm1\sigma$) that the true land-use change emission lies within the given range, for the
range of processes considered here. This uncertainty range had been increased from 0.5 GtC yr$^{-1}$ after new
bookkeeping models were included that indicated a larger spread than assumed before (Le Quéré et al., 2018).
Projections for 2021 are based on fire activity from tropical deforestation and degradation as well as emissions
from peat fires and drainage.
Our $E_{LUC}$ estimates follow the definition of global carbon cycle models of $CO_2$ fluxes related to land-use and
land management and differ from IPCC definitions adopted in National GHG Inventories (NGHGI) for
reporting under the UNFCCC, which additionally generally include, through adoption of the IPCC so-called
managed land proxy approach, the terrestrial fluxes occurring on land defined by countries as managed. This
partly includes fluxes due to environmental change (e.g. atmospheric $CO_2$ increase), which are part of $S_{LAND}$ in
our definition. This causes the global emission estimates to be smaller for NGHGI than for the global carbon
budget definition (Grassi et al., 2018). The same is the case for the Food Agriculture Organization (FAO)
estimates of carbon fluxes on forest land, which include, compared to $S_{LAND}$, both anthropogenic and natural
sources on managed land (Tubiello et al., 2021). Using the approach outlined in Grassi et al. (2021), here we
map as additional information the two definitions to each other, to provide a comparison of the anthropogenic
carbon budget to the official country reporting to the climate convention.
### 2.2.2 2022 Projection
We project the 2022 land-use emissions for BLUE, the updated H&N2017 and OSCAR, starting from their
estimates for 2021 assuming unaltered peat drainage, which has low interannual variability, and the highly



variable emissions from peat fires, tropical deforestation and degradation as estimated using active fire data
(MCD14ML; Giglio et al., 2016). More details on the $E_{LUC}$ methodology can be found in Appendix C.2

### 2.3 Growth rate in atmospheric CO₂ concentration ($G_{ATM}$)

**2.3**    **Growth rate in atmospheric CO₂ concentration ($G_{ATM}$)**

#### 2.3.1 Historical period

**2.3.1**    **Historical period**
The rate of growth of the atmospheric $CO_2$ concentration is provided for years 1959-2021 by the US National
Oceanic and Atmospheric Administration Earth System Research Laboratory (NOAA/ESRL; Dlugokencky and
Tans, 2022), which is updated from Ballantyne et al. (2012) and includes recent revisions to the calibration scale
of atmospheric $CO_2$ measurements (Hall et al., 2021). For the 1959-1979 period, the global growth rate is based
on measurements of atmospheric $CO_2$ concentration averaged from the Mauna Loa and South Pole stations, as
observed by the $CO_2$ Program at Scripps Institution of Oceanography (Keeling et al., 1976). For the 1980-2020
time period, the global growth rate is based on the average of multiple stations selected from the marine
boundary layer sites with well-mixed background air (Ballantyne et al., 2012), after fitting a smooth curve
through the data for each station as a function of time, and averaging by latitude band (Masarie and Tans, 1995).
The annual growth rate is estimated by Dlugokencky and Tans (2022) from atmospheric $CO_2$ concentration by
taking the average of the most recent December-January months corrected for the average seasonal cycle and
subtracting this same average one year earlier. The growth rate in units of ppm yr$^{-1}$ is converted to units of GtC
yr$^{-1}$ by multiplying by a factor of 2.124 GtC per ppm, assuming instantaneous mixing of $CO_2$ throughout the
atmosphere (Ballantyne et al., 2012; Table 1).
Since 2020, NOAA/ESRL provides estimates of atmospheric $CO_2$ concentrations with respect to a new
calibration scale, referred to as WMO-CO2-X2019, in line with the recommendation of the World
Meteorological Organization (WMO) Global Atmosphere Watch (GAW) community (Hall et al., 2021). The
WMO-CO2-X2019 scale improves upon the earlier WMO-CO2-X2007 scale by including a broader set of
standards, which contain $CO_2$ in a wider range of concentrations that span the range 250-800 ppm (versus 250–
520 ppm for WMO-CO2-X2007). In addition, NOAA/ESRL made two minor corrections to the analytical
procedure used to quantify $CO_2$ concentrations, fixing an error in the second virial coefficient of $CO_2$ and
accounting for loss of a small amount of $CO_2$ to materials in the manometer during the measurement process.
The difference in concentrations measured using WMO-CO2-X2019 versus WMO-CO2-X2007 is ~+0.18 ppm
at 400 ppm and the observational record of atmospheric $CO_2$ concentrations have been revised accordingly. The
revisions have been applied retrospectively in all cases where the calibrations were performed by NOAA/ESRL,
thus affecting measurements made by members of the WMO-GAW programme and other regionally
coordinated programmes (e.g., Integrated Carbon Observing System, ICOS). Changes to the $CO_2$ concentrations
measured across these networks propagate to the global mean $CO_2$ concentrations. The re-calibrated data were
first used to estimate $G_{ATM}$ in the 2021 edition of the global carbon budget (Friedlingstein et al., 2022a).
Friedlingstein et al. (2022a) verified that the change of scales from WMO-CO2-X2007 to WMO-CO2-X2019
made a negligible difference to the value of $G_{ATM}$ (-0.06 GtC yr$^{-1}$ during 2010-2019 and -0.01 GtC yr$^{-1}$ during
1959-2019, well within the uncertainty range reported below).
The uncertainty around the atmospheric growth rate is due to four main factors. First, the long-term
reproducibility of reference gas standards (around 0.03 ppm for 1σ from the 1980s; Dlugokencky and Tans,



2022). Second, small unexplained systematic analytical errors that may have a duration of several months to two
years come and go. They have been simulated by randomising both the duration and the magnitude (determined
from the existing evidence) in a Monte Carlo procedure. Third, the network composition of the marine boundary
layer with some sites coming or going, gaps in the time series at each site, etc (Dlugokencky and Tans, 2022).
The latter uncertainty was estimated by NOAA/ESRL with a Monte Carlo method by constructing 100
"alternative" networks (Masarie and Tans, 1995; NOAA/ESRL, 2019). The second and third uncertainties,
summed in quadrature, add up to 0.085 ppm on average (Dlugokencky and Tans, 2022). Fourth, the uncertainty
associated with using the average $CO_2$ concentration from a surface network to approximate the true
atmospheric average $CO_2$ concentration (mass-weighted, in 3 dimensions) as needed to assess the total
atmospheric $CO_2$ burden. In reality, $CO_2$ variations measured at the stations will not exactly track changes in
total atmospheric burden, with offsets in magnitude and phasing due to vertical and horizontal mixing. This
effect must be very small on decadal and longer time scales, when the atmosphere can be considered well
mixed. Preliminary estimates suggest this effect would increase the annual uncertainty, but a full analysis is not
yet available. We therefore maintain an uncertainty around the annual growth rate based on the multiple stations
data set ranges between 0.11 and 0.72 GtC yr$^{-1}$, with a mean of 0.61 GtC yr$^{-1}$ for 1959-1979 and 0.17 GtC yr$^{-1}$
for 1980-2020, when a larger set of stations were available as provided by Dlugokencky and Tans (2022) but
recognise further exploration of this uncertainty is required. At this time, we estimate the uncertainty of the
decadal averaged growth rate after 1980 at 0.02 GtC yr$^{-1}$ based on the calibration and the annual growth rate
uncertainty but stretched over a 10-year interval. For years prior to 1980, we estimate the decadal averaged
uncertainty to be 0.07 GtC yr$^{-1}$ based on a factor proportional to the annual uncertainty prior and after 1980
(0.02 * [0.61/0.17] GtC yr$^{-1}$).
We assign a high confidence to the annual estimates of $G_{ATM}$ because they are based on direct measurements
from multiple and consistent instruments and stations distributed around the world (Ballantyne et al., 2012; Hall
et al., 2021).
To estimate the total carbon accumulated in the atmosphere since 1750 or 1850, we use an atmospheric $CO_2$
concentration of 277 ± 3 ppm or 286 ± 3 ppm, respectively, based on a cubic spline fit to ice core data (Joos and
Spahni, 2008). For the construction of the cumulative budget shown in Figure 3, we use the fitted estimates of
$CO_2$ concentration from Joos and Spahni (2008) to estimate the annual atmospheric growth rate using the
conversion factors shown in Table 1. The uncertainty of ±3 ppm (converted to ±1σ) is taken directly from the
IPCC's AR5 assessment (Ciais et al., 2013). Typical uncertainties in the growth rate in atmospheric $CO_2$
concentration from ice core data are equivalent to ±0.1-0.15 GtC yr$^{-1}$ as evaluated from the Law Dome data
(Etheridge et al., 1996) for individual 20-year intervals over the period from 1850 to 1960 (Bruno and Joos,

498 1997).

**2.3.2    2022 projection**
We provide an assessment of $G_{ATM}$ for 2022 based on the monthly calculated global atmospheric $CO_2$
concentration (GLO) through August (Dlugokencky and Tans, 2022), and bias-adjusted Holt–Winters
exponential smoothing with additive seasonality (Chatfield, 1978) to project to January 2023. Additional
analysis suggests that the first half of the year (the boreal winter-spring-summer transition) shows more
interannual variability than the second half of the year (the boreal summer-autumn-winter transition), so that the



exact projection method applied to the second half of the year has a relatively smaller impact on the projection
of the full year. Uncertainty is estimated from past variability using the standard deviation of the last 5 years'
monthly growth rates.
**2.4    Ocean $CO_2$ sink**
**2.4.1    Historical Period**
The reported estimate of the global ocean anthropogenic $CO_2$ sink $S_{OCEAN}$ is derived as the average of two
estimates. The first estimate is derived as the mean over an ensemble of ten global ocean biogeochemistry
models (GOBMs, Table 4 and Table A2). The second estimate is obtained as the mean over an ensemble of
seven observation-based data-products (Table 4 and Table A3). An eighth product (Watson et al., 2020) is
shown, but is not included in the ensemble average as it differs from the other products by adjusting the flux to a
cool, salty ocean surface skin (see Appendix C.3.1 for a discussion of the Watson product). The GOBMs
simulate both the natural and anthropogenic $CO_2$ cycles in the ocean. They constrain the anthropogenic air-sea
$CO_2$ flux (the dominant component of $S_{OCEAN}$) by the transport of carbon into the ocean interior, which is also
the controlling factor of present-day ocean carbon uptake in the real world. They cover the full globe and all
seasons and were recently evaluated against surface ocean carbon observations, suggesting they are suitable to
estimate the annual ocean carbon sink (Hauck et al., 2020). The data-products are tightly linked to observations
of $fCO_2$ (fugacity of $CO_2$, which equals $pCO_2$ corrected for the non-ideal behaviour of the gas; Pfeil et al.,
2013), which carry imprints of temporal and spatial variability, but are also sensitive to uncertainties in gas-
exchange parameterizations and data-sparsity. Their asset is the assessment of interannual and spatial variability
(Hauck et al., 2020). We further use two diagnostic ocean models to estimate $S_{OCEAN}$ over the industrial era

525    (1781-1958).

The global $fCO_2$-based flux estimates were adjusted to remove the pre-industrial ocean source of $CO_2$ to the
atmosphere of 0.65 GtC yr$^{-1}$ from river input to the ocean (Regnier et al., 2022), to satisfy our definition of
$S_{OCEAN}$ (Hauck et al., 2020). The river flux adjustment was distributed over the latitudinal bands using the
regional distribution of Aumont et al. (2001; North: 0.17 GtC yr$^{-1}$, Tropics: 0.16 GtC yr$^{-1}$, South: 0.32 GtC yr$^{-1}$),
acknowledging that the boundaries of Aumont et al (2001; namely 20°S and 20°N) are not consistent with the
boundaries otherwise used in the GCB (30°S and 30°N). A recent study based on one ocean biogeochemical
model (Lacroix et al., 2020) suggests that more of the riverine outgassing is located in the tropics than in the
Southern Ocean; and hence this regional distribution is associated with a major uncertainty. Anthropogenic
perturbations of river carbon and nutrient transport to the ocean are not considered (see section 2.7).
We derive $S_{OCEAN}$ from GOBMs by using a simulation (sim A) with historical forcing of climate and
atmospheric $CO_2$, accounting for model biases and drift from a control simulation (sim B) with constant
atmospheric $CO_2$ and normal year climate forcing. A third simulation (sim C) with historical atmospheric $CO_2$
increase and normal year climate forcing is used to attribute the ocean sink to $CO_2$ (sim C minus sim B) and
climate (sim A minus sim C) effects. A fourth simulation (sim D; historical climate forcing and constant
atmospheric $CO_2$) is used to compare the change in anthropogenic carbon inventory in the interior ocean (sim A
minus sim D) to the observational estimate of Gruber et al. (2019) with the same flux components (steady state
and non-steady state anthropogenic carbon flux). Data-products are adjusted to represent the full ice-free ocean



area by a simple scaling approach when coverage is below 99%. GOBMs and data-products fall within the
observational constraints over the 1990s ($2.2 \pm 0.7$ GtC yr$^{-1}$, Ciais et al., 2013) after applying adjustments.
$S_{OCEAN}$ is calculated as the average of the GOBM ensemble mean and data-product ensemble mean from 1990
onwards. Prior to 1990, it is calculated as the GOBM ensemble mean plus half of the offset between GOBMs
and data-products ensemble means over 1990-2001.
We assign an uncertainty of $\pm 0.4$ GtC yr$^{-1}$ to the ocean sink based on a combination of random (ensemble
standard deviation) and systematic uncertainties (GOBMs bias in anthropogenic carbon accumulation,
previously reported uncertainties in fCO$_2$-based data-products; see section C.3.3). We assess a medium
confidence level to the annual ocean CO$_2$ sink and its uncertainty because it is based on multiple lines of
evidence, it is consistent with ocean interior carbon estimates (Gruber et al., 2019, see section 3.5.5) and the
interannual variability in the GOBMs and data-based estimates is largely consistent and can be explained by
climate variability. We refrain from assigning a high confidence because of the systematic deviation between
the GOBM and data-product trends since around 2002. More details on the $S_{OCEAN}$ methodology can be found in
Appendix C.3.

### 557 2.4.2 2022 Projection

The ocean CO$_2$ sink forecast for the year 2022 is based on the annual historical and estimated 2022 atmospheric
CO$_2$ concentration (Dlugokencky and Tans 2021), the historical and estimated 2022 annual global fossil fuel
emissions from this year's carbon budget, and the spring (March, April, May) Oceanic Niño Index (ONI) index
(NCEP, 2022). Using a non-linear regression approach, i.e., a feed-forward neural network, atmospheric CO$_2$,
the ONI index and the fossil fuel emissions are used as training data to best match the annual ocean CO$_2$ sink
(i.e. combined $S_{OCEAN}$ estimate from GOBMs and data products) from 1959 through 2021 from this year's
carbon budget. Using this relationship, the 2022 $S_{OCEAN}$ can then be estimated from the projected 2021 input
data using the non-linear relationship established during the network training. To avoid overfitting, the neural
network was trained with a variable number of hidden neurons (varying between 2-5) and 20% of the randomly
selected training data were withheld for independent internal testing. Based on the best output performance
(tested using the 20% withheld input data), the best performing number of neurons was selected. In a second
step, we trained the network 10 times using the best number of neurons identified in step 1 and different sets of
randomly selected training data. The mean of the 10 trainings is considered our best forecast, whereas the
standard deviation of the 10 ensembles provides a first order estimate of the forecast uncertainty. This
uncertainty is then combined with the $S_{OCEAN}$ uncertainty (0.4 GtC yr$^{-1}$) to estimate the overall uncertainty of the
2022 projection.

### 574 2.5 Terrestrial CO$_2$ sink

### 575 2.5.1 Historical Period

The terrestrial land sink ($S_{LAND}$) is thought to be due to the combined effects of fertilisation by rising
atmospheric CO$_2$ and N inputs on plant growth, as well as the effects of climate change such as the lengthening
of the growing season in northern temperate and boreal areas. $S_{LAND}$ does not include land sinks directly
resulting from land-use and land-use change (e.g., regrowth of vegetation) as these are part of the land-use flux



($E_{LUC}$), although system boundaries make it difficult to attribute exactly $CO_2$ fluxes on land between $S_{LAND}$ and
$E_{LUC}$ (Erb et al., 2013).
$S_{LAND}$ is estimated from the multi-model mean of 16 DGVMs (Table A1). As described in Appendix C.4,
DGVMs simulations include all climate variability and $CO_2$ effects over land, with 11 DGVMs also including
the effect of N inputs. The DGVMs estimate of $S_{LAND}$ does not include the export of carbon to aquatic systems
or its historical perturbation, which is discussed in Appendix D3. See Appendix C.4 for DGVMs evaluation and
uncertainty assessment for $S_{LAND}$, using the International Land Model Benchmarking system (ILAMB; Collier et
al., 2018). More details on the $S_{LAND}$ methodology can be found in Appendix C.4.

### 2.5.2    2022 Projection

Like for the ocean forecast, the land $CO_2$ sink ($S_{LAND}$) forecast is based on the annual historical and estimated
2022 atmospheric $CO_2$ concentration (Dlugokencky and Tans 2021), historical and estimated 2022 annual
global fossil fuel emissions from this year's carbon budget, and the summer (June, July, August) ONI index
(NCEP, 2022). All training data are again used to best match $S_{LAND}$ from 1959 through 2021 from this year's
carbon budget using a feed-forward neural network. To avoid overfitting, the neural network was trained with a
variable number of hidden neurons (varying between 2-15), larger than for $S_{OCEAN}$ prediction due to the stronger
land carbon interannual variability. As done for $S_{OCEAN}$, a pre-training selects the optimal number of hidden
neurons based on 20% withheld input data, and in a second step, an ensemble of 10 forecasts is produced to
provide the mean forecast plus uncertainty. This uncertainty is then combined with the $S_{LAND}$ uncertainty for
2021 (0.9 GtC yr$^{-1}$) to estimate the overall uncertainty of the 2022 projection.

### 2.6    The atmospheric perspective

The world-wide network of in-situ atmospheric measurements and satellite derived atmospheric $CO_2$ column
(x$CO_2$) observations put a strong constraint on changes in the atmospheric abundance of $CO_2$. This is true
globally (hence our large confidence in $G_{ATM}$), but also regionally in regions with sufficient observational
density found mostly in the extra-tropics. This allows atmospheric inversion methods to constrain the magnitude
and location of the combined total surface $CO_2$ fluxes from all sources, including fossil and land-use change
emissions and land and ocean $CO_2$ fluxes. The inversions assume $E_{FOS}$ to be well known, and they solve for the
spatial and temporal distribution of land and ocean fluxes from the residual gradients of $CO_2$ between stations
that are not explained by fossil fuel emissions. By design, such systems thus close the carbon balance ($B_{IM} = 0$)
and thus provide an additional perspective on the independent estimates of the ocean and land fluxes.
This year's release includes nine inversion systems that are described in Table A4. Each system is rooted in
Bayesian inversion principles but uses different methodologies. These differences concern the selection of
atmospheric $CO_2$ data or x$CO_2$, and the choice of a-priori fluxes to refine. They also differ in spatial and
temporal resolution, assumed correlation structures, and mathematical approach of the models (see references in
Table A4 for details). Importantly, the systems use a variety of transport models, which was demonstrated to be
a driving factor behind differences in atmospheric inversion-based flux estimates, and specifically their
distribution across latitudinal bands (Gaubert et al., 2019; Schuh et al., 2019). Four inversion systems (CAMS-
FT21r2, CMS-flux, GONGGA, THU) used satellite x$CO_2$ retrievals from GOSAT and/or OCO-2, scaled to the



WMO 2019 calibration scale. One inversion this year (CMS-Flux) used these xCO2 datasets in addition to the
in-situ observational $CO_2$ mole fraction records.
The original products delivered by the inverse modellers were modified to facilitate the comparison to the other
elements of the budget, specifically on two accounts: (1) global total fossil fuel emissions including cement
carbonation $CO_2$ uptake, and (2) riverine $CO_2$ transport. Details are given below. We note that with these
adjustments the inverse results no longer represent the net atmosphere-surface exchange over land/ocean areas
as sensed by atmospheric observations. Instead, for land, they become the net uptake of $CO_2$ by vegetation and
soils that is not exported by fluvial systems, similar to the DGVMs estimates. For oceans, they become the net
uptake of anthropogenic $CO_2$, similar to the GOBMs estimates.
The inversion systems prescribe global fossil fuel emissions based on the GCP's Gridded Fossil Emissions
Dataset versions 2022.1 or 2022.2 (GCP-GridFED; Jones et al., 2022), which are updates to GCP-
GridFEDv2021 presented by Jones et al. (2021). GCP-GridFEDv2022 scales gridded estimates of $CO_2$
emissions from EDGARv4.3.2 (Janssens-Maenhout et al., 2019) within national territories to match national
emissions estimates provided by the GCB for the years 1959-2021, which were compiled following the
methodology described in Section 2.1. Small differences between the systems due to for instance regridding to
the transport model resolution, or use of different GridFED versions with different cement carbonation sinks
(which were only present starting with GridFEDv2022.1), are adjusted in the latitudinal partitioning we present,
to ensure agreement with the estimate of $E_{FOS}$ in this budget. We also note that the ocean fluxes used as prior by
6 out of 9 inversions are part of the suite of the ocean process model or fCO2 data products listed in Section 2.4.
Although these fluxes are further adjusted by the atmospheric inversions, it makes the inversion estimates of the
ocean fluxes not completely independent of $S_{OCEAN}$ assessed here.
To facilitate comparisons to the independent $S_{OCEAN}$ and $S_{LAND}$, we used the same corrections for transport and
outgassing of carbon transported from land to ocean, as done for the observation-based estimates of $S_{OCEAN}$ (see
Appendix C.3).
The atmospheric inversions are evaluated using vertical profiles of atmospheric $CO_2$ concentrations (Figure B4).
More than 30 aircraft programs over the globe, either regular programs or repeated surveys over at least 9
months (except for SH programs), have been used to assess system performance (with space-time observational
coverage sparse in the SH and tropics, and denser in NH mid-latitudes; Table A6). The nine systems are
compared to the independent aircraft $CO_2$ measurements between 2 and 7 km above sea level between 2001 and
2021. Results are shown in Figure B4 and discussed in Section 3.7.
With a relatively small ensemble (N=9) of systems that moreover share some a-priori fluxes used with one
another, or with the process-based models, it is difficult to justify using their mean and standard deviation as a
metric for uncertainty across the ensemble. We therefore report their full range (min-max) without their mean.
More details on the atmospheric inversions methodology can be found in Appendix C.5.
**2.7    Processes not included in the global carbon budget**
The contribution of anthropogenic CO and $CH_4$ to the global carbon budget is not fully accounted for in Eq. (1)
and is described in Appendix D1. The contributions to $CO_2$ emissions of decomposition of carbonates not

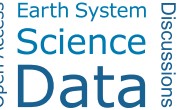

accounted for is described in Appendix D2. The contribution of anthropogenic changes in river fluxes is
conceptually included in Eq. (1) in $S_{OCEAN}$ and in $S_{LAND}$, but it is not represented in the process models used to
quantify these fluxes. This effect is discussed in Appendix D3. Similarly, the loss of additional sink capacity
from reduced forest cover is missing in the combination of approaches used here to estimate both land fluxes
($E_{LUC}$ and $S_{LAND}$) and its potential effect is discussed and quantified in Appendix D4.

## 3 Results

For each component of the global carbon budget, we present results for three different time periods: the full
historical period, from 1850 to 2021, the six decades in which we have atmospheric concentration records from
Mauna Loa (1960-2021), a specific focus on last year (2021), and the projection for the current year (2022).
Subsequently, we assess the combined constraints from the budget components (often referred to as a bottom-up
budget) against the top-down constraints from inverse modelling of atmospheric observations. We do this for
the global balance of the last decade, as well as for a regional breakdown of land and ocean sinks by broad
latitude bands.

### 3.1 Fossil $CO_2$ Emissions

#### 3.1.1 Historical period 1850-2021

Cumulative fossil $CO_2$ emissions for 1850-2021 were $465 \pm 25$ GtC, including the cement carbonation sink
(Figure 3, Table 8, all cumulative numbers are rounded to the nearest 5GtC).
In this period, 46% of fossil $CO_2$ emissions came from coal, 35% from oil, 15% from natural gas, 3% from
decomposition of carbonates, and 1% from flaring.
In 1850, the UK stood for 62% of global fossil $CO_2$ emissions. In 1891 the combined cumulative emissions of
the current members of the European Union reached and subsequently surpassed the level of the UK. Since
1917 US cumulative emissions have been the largest. Over the entire period 1850-2021, US cumulative
emissions amounted to 115GtC (24% of world total), the EU's to 80 GtC (17%), and China's to 70 GtC (14%).
There are three additional global datasets with long time series that include all sources of fossil $CO_2$ emissions:
CDIAC-FF (Gilfillan and Marland, 2021), CEDS version v_2021_04_21 (Hoesly et al., 2018); O'Rourke et al.,
2021) and PRIMAP-hist version 2.3.1 (Gütschow et al., 2016, 2021), although these datasets are not entirely
independent from each other. CDIAC-FF has the lowest cumulative emissions over 1750-2018 at 437 GtC, GCP
has 443 GtC, CEDS 445 GtC, PRIMAP-hist TP 453 GtC, and PRIMAP-hist CR 455 GtC. CDIAC-FF excludes
emissions from lime production, while neither CDIAC-FF nor GCP explicitly include emissions from
international bunker fuels prior to 1950. CEDS has higher emissions from international shipping in recent years,
while PRIMAP-hist has higher fugitive emissions than the other datasets. However, in general these four
datasets are in relative agreement as to total historical global emissions of fossil $CO_2$.

#### 3.1.2 Recent period 1960-2021

Global fossil $CO_2$ emissions, $E_{FOS}$ (including the cement carbonation sink), have increased every decade from an
average of $3.0 \pm 0.2$ GtC $yr^{-1}$ for the decade of the 1960s to an average of $9.6 \pm 0.5$ GtC $yr^{-1}$ during 2012-2021
(Table 6, Figure 2 and Figure 5). The growth rate in these emissions decreased between the 1960s and the



1990s, from 4.3% $yr^{-1}$ in the 1960s (1960-1969), 3.2% $yr^{-1}$ in the 1970s (1970-1979), 1.6% $yr^{-1}$ in the 1980s
(1980-1989), to 0.9% $yr^{-1}$ in the 1990s (1990-1999). After this period, the growth rate began increasing again in
the 2000s at an average growth rate of 3.0% $yr^{-1}$, decreasing to 0.5% $yr^{-1}$ for the last decade (2012-2021).
China's emissions increased by +1.5% $yr^{-1}$ on average over the last 10 years dominating the global trend, and
India's emissions increased by +3.8% $yr^{-1}$, while emissions decreased in EU27 by –1.8% $yr^{-1}$, and in the USA
by –1.1% $yr^{-1}$. Figure 6 illustrates the spatial distribution of fossil fuel emissions for the 2012-2021 period.
$E_{FOS}$ includes the uptake of $CO_2$ by cement via carbonation which has increased with increasing stocks of
cement products, from an average of 20 MtC $yr^{-1}$ (0.02 GtC $yr^{-1}$) in the 1960s to an average of 200 MtC $yr^{-1}$ (0.2
GtC $yr^{-1}$) during 2012-2021 (Figure 5).

### 3.1.3    Final year 2021

Global fossil $CO_2$ emissions were 5.1% higher in 2021 than in 2020, because of the global rebound from the
worst of the COVID-19 pandemic, with an increase of 0.5 GtC to reach $10.1 \pm 0.5$ GtC ($9.9 \pm 0.5$ GtC when
including the cement carbonation sink) in 2021 (Figure 5), distributed among coal (41%), oil (32%), natural gas
(22%), cement (5%) and others (1%). Compared to the previous year, 2021 emissions from coal, oil and gas
increased by 5.7%, 5.8% and 4.8% respectively, while emissions from cement increased by 2.1%. All growth
rates presented are adjusted for the leap year, unless stated otherwise.
In 2021, the largest absolute contributions to global fossil $CO_2$ emissions were from China (31%), the USA
(14%), the EU27 (8%), and India (7%). These four regions account for 59% of global $CO_2$ emissions, while the
rest of the world contributed 41%, including international aviation and marine bunker fuels (2.8% of the total).
Growth rates for these countries from 2020 to 2021 were 3.5% (China), 6.2% (USA), 6.8% (EU27), and 11.1%
(India), with +4.5% for the rest of the world. The per-capita fossil $CO_2$ emissions in 2021 were 1.3 tC person$^{-1}$
$yr^{-1}$ for the globe, and were 4.0 (USA), 2.2 (China), 1.7 (EU27) and 0.5 (India) tC person$^{-1}$ $yr^{-1}$ for the four
highest emitting countries (Figure 5).
The post-COVID-19 rebound in emissions of 5.1% in 2021 is close to the projected increase of 4.8% published
in Friedlingstein et al. (2021) (Table 7). Of the regions, the projection for the 'rest of world' region was least
accurate, largely because of poorly projected emissions from international transport (bunker fuels), which were
subject to very large changes during this period.

### 3.1.4    Year 2022 Projection

Globally, we estimate that global fossil $CO_2$ emissions will grow by 1.1% in 2022 (0.0% to 1.7%) to 10.2 GtC
(37.3 $GtCO_2$), exceeding their 2019 emission levels of 10.0 GtC (36.7 $GtCO_2$). Global increase in 2022
emissions per fuel types are projected to be +0.8% (range 0.0% to 1.7%) for coal, +2.2% (range -0.7% to 2.9%)
for oil, +1.1% (range 0.0% to 2.2%) for natural gas, and -2.8% (range -5.5% to -0.2%) for cement.
For China, projected fossil emissions in 2022 are expected to decline by 1.5% (range -3.0% to +0.1%) compared
with 2021 emissions, bringing 2022 emissions for China around 3.0 GtC $yr^{-1}$ (11.1 $GtCO_2$ $yr^{-1}$). Changes in fuel
specific projections for China are -0.5% for coal, -2.3% for oil, -1.1% natural gas, and -9.2% for cement.





For the USA, the Energy Information Administration (EIA) emissions projection for 2022 combined with
cement clinker data from USGS gives an increase of 1.6% (range -0.9% to +4.1%) compared to 2021, bringing
USA 2022 emissions to around 1.4 GtC yr$^{-1}$ (5.1 GtCO$_2$ yr$^{-1}$). This is based on separate projections for coal -
2.8%, oil +1.9%, natural gas +4.1%, and cement +0.7%.
For the European Union, our projection for 2022 is for a decline of 1.0% (range -2.9% to +1.0%) over 2021,
with 2022 emissions around 0.8 GtC yr$^{-1}$ (2.8 GtCO$_2$ yr$^{-1}$). This is based on separate projections for coal of
+7.5%, oil +0.6%, natural gas -11.0%, and cement unchanged.
For India, our projection for 2022 is an increase of 5.6% (range of 3.5% to 7.7%) over 2021, with 2022
emissions around 0.8 GtC yr$^{-1}$ (2.9 GtCO$_2$ yr$^{-1}$). This is based on separate projections for coal of +5.0%, oil
+8.0%, natural gas -3.0%, and cement +10.0%.
For the rest of the world, the expected growth rate for 2022 is 2.5% (range 0.1% to 2.3%). The fuel-specific
projected 2022 growth rates for the rest of the world are: +1.4% (range -0.6% to +3.4%) for coal, +3.2% (1.6%
to +4.9%) for oil, +2.6% (1.1% to 4.1%) for natural gas, +2.8% (+0.6% to +5.1%) for cement.
**3.2    Emissions from Land Use Changes**
**3.2.1    Historical period 1850-2021**
Cumulative CO$_2$ emissions from land-use changes (E$_{LUC}$) for 1850-2021 were 205 ± 60 GtC (Table 8; Figure 3;
Figure 14). The cumulative emissions from E$_{LUC}$ are particularly uncertain, with large spread among individual
estimates of 140 GtC (updated H&N2017), 280 GtC (BLUE), and 190 GtC (OSCAR) for the three bookkeeping
models and a similar wide estimate of 185 ± 60 GtC for the DGVMs (all cumulative numbers are rounded to the
nearest 5GtC). These estimates are broadly consistent with indirect constraints from vegetation biomass
observations, giving a cumulative source of 155 ± 50 GtC over the 1901-2012 period (Li et al., 2017). However,
given the large spread, a best estimate is difficult to ascertain.
**3.2.2    Recent period 1960-2021**
In contrast to growing fossil emissions, CO$_2$ emissions from land-use, land-use change, and forestry have
remained relatively constant, over the 1960-1999 period, but showing a slight decrease of about 0.1 GtC per
decade since the 1990s, reaching 1.2 ± 0.7 GtC yr$^{-1}$ for the 2012-2021 period (Table 6), but with large spread
across estimates (Table 5, Figure 7). Different from the bookkeeping average, the DGVMs model average grows
slightly larger over the 1970-2021 period and shows no sign of decreasing emissions in the recent decades
(Table 5, Figure 7). This is, however, expected as DGVM-based estimates include the loss of additional sink
capacity, which grows with time, while the bookkeeping estimates do not (Appendix D4).
E$_{LUC}$ is a net term of various gross fluxes, which comprise emissions and removals. Gross emissions on average
over the 1850-2021 period are two (BLUE, OSCAR) to three (updated H&N2017) times larger than the net E$_{LUC}$
emissions, and remained largely constant over the last 60 years, with a moderate increase from an average of 3.2
± 0.9 GtC yr$^{-1}$ for the decade of the 1960s to an average of 3.8 ± 0.7 GtC yr$^{-1}$ during 2012-2021 (Figure 7),
showing the relevance of land management such as harvesting or rotational agriculture. Increases in gross
removals, from 1.8 ± 0.4 GtC yr$^{-1}$ for the 1960s to 2.6 ± 0.4 GtC yr$^{-1}$ for 2012-2021, were slightly larger than the



increase in gross emissions. Since the processes behind gross removals, foremost forest regrowth and soil
recovery, are all slow, while gross emissions include a large instantaneous component, short-term changes in
land-use dynamics, such as a temporary decrease in deforestation, influences gross emissions dynamics more
than gross removals dynamics. It is these relative changes to each other that explain the small decrease in net
$E_{LUC}$ emissions over the last two decades and the last few years. Gross fluxes often differ more across the three
bookkeeping estimates than net fluxes, which is expected due to different process representation; in particular,
treatment of shifting cultivation, which increases both gross emissions and removals, differs across models.
There is a smaller decrease in net $CO_2$ emissions from land-use change in the last few years (Figure 7) than in
our last year's estimate (Friedlingstein et al., 2021), which places our updated estimates between last year's
estimate and the estimate from the GCB2020 (Friedlingstein et al., 2020). This change is principally attributable
to changes in $E_{LUC}$ estimates from BLUE and OSCAR, which relate to changes in the underlying land-use
forcing (see Appendix C.2.2 for details). These changes address issues identified with last year's land-use
forcing (see Friedlingstein et al., 2022) and remove/attenuate several emission peaks in Brazil and the DR
Congo and lead to higher net emissions in Brazil in the last decades compared to last year's global carbon
budget. While we deem these changes in land-use forcing and emissions an improvement, the estimated
emissions based on the new land-use forcing still do not fully reflect the rise in Brazilian deforestation in the
recent few years (Silva Junior, 2021), and associated increasing emissions from deforestation would have been
missed here. Differences still exist, which highlight the need for accurate knowledge of land-use transitions and
their spatial and temporal variability. A further caveat is that global land-use change data for model input does
not capture forest degradation, which often occurs on small scale or without forest cover changes easily
detectable from remote sensing and poses a growing threat to forest area and carbon stocks that may surpass
deforestation effects (e.g., Matricardi et al., 2020, Qin et al., 2021).
We additionally separate the net $E_{LUC}$ into component fluxes to gain further insight into the drivers of gross
sources and sinks and how the bookkeeping models compare to each other (Figure 7; Sec. C.2.1). On average
over the 2012-2021 period and over the three bookkeeping estimates, emissions from deforestation amount to
$1.8 \pm 0.4$ GtC yr$^{-1}$ and carbon uptake in forests to $-0.9 \pm 0.3$ GtC yr$^{-1}$ (Table 5). Emissions from organic soils
caused by peat drainage or peat fires (with $0.2 \pm 0.1$ GtC yr$^{-1}$) and the net flux from other transitions (with $0.1 \pm$
$0.1$ GtC yr$^{-1}$) are substantially less important globally, but emissions from organic soils contribute over
proportionally to interannual variability (related in particular to peat fires in dry years in Southeast Asia).
Deforestation is thus the main driver of global gross sources. The relatively small deforestation flux in
comparison to the gross source estimate above is explained by the fact that emissions associated with wood
harvesting, while they do constitute a source of emissions to the atmosphere, are contained in the component
flux on forest, together with the associated carbon uptake in regrowth, because wood harvesting does not change
the land cover. For the same reason the flux on forest, being a net flux of sources from slash and product decay
following wood harvest and sinks due to regrowth after wood harvest or after abandonment, is smaller than the
gross sink estimates above. This split into component fluxes thus clarifies better the potentials for emission
reduction and carbon dioxide removal than the gross fluxes do: the emissions from deforestation could be halted
(largely) without compromising carbon uptake in other component fluxes and contribute to emissions reduction;
reforestation following agricultural abandonment does not directly depend on deforestation and can
independently provide carbon dioxide removal. By contrast, reducing wood harvesting to reduce emissions to

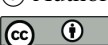



the atmosphere is associated with less forest regrowth; sinks and sources cannot be decoupled here. Last, we
compare our component flux estimates to NGHGI (Grassi et al., 2022b): With 1.1 GtC yr$^{-1}$ averaged over 2012-
2021, deforestation emissions are reported to be smaller by countries than the bookkeeping estimate. A reason
for this lies in the fact that country reports do not (fully) capture the carbon flux consequences of shifting
cultivation. With 0.3 GtC yr$^{-1}$ and 0.2 GtC yr$^{-1}$, emissions from organic soils and the net flux from other
transitions, respectively, are similar (slightly larger) than the estimates based on the bookkeeping approach and
the external peat drainage and burning datasets. With 1.75 GtC yr$^{-1}$, carbon uptake in forests is substantially
larger, owing to the inclusion of natural $CO_2$ fluxes on managed land in the NGHGI (see below).
Overall, highest land-use emissions occur in the tropical regions of all three continents. The top three emitters
(both cumulatively 1959-2021 and on average over 2012-2021) are Brazil (in particular the Amazon Arc of
Deforestation), Indonesia and the Democratic Republic of the Congo, with these 3 countries contributing 0.7
GtC yr$^{-1}$ or 58% of the global total land-use emissions (average over 2012-2021) (Figure 6b). This is related to
massive expansion of cropland, particularly in the last few decades in Latin America, Southeast Asia, and sub-
Saharan Africa Emissions (Hong et al., 2021), to a substantial part for export (Pendrill et al., 2019). Emission
intensity is high in many tropical countries, particularly of Southeast Asia, due to high rates of land conversion
in regions of carbon-dense and often still pristine, undegraded natural forests (Hong et al., 2021). Emissions are
further increased by peat fires in equatorial Asia (GFED4s, van der Werf et al., 2017). Uptake due to land-use
change occurs, particularly in Europe, partly related to expanding forest area as a consequence of the forest
transition in the 19$^{th}$ and 20$^{th}$ century and subsequent regrowth of forest (Figure 6b) (Mather 2001; McGrath et
al., 2015).
While the mentioned patterns are supported by independent literature and robust, we acknowledge that model
spread is substantially larger on regional than global level, as has been shown for bookkeeping models (Bastos
et al., 2021) as well as DGVMs (Obermeier et al., 2021). A detailed analysis of country-level or regional
uncertainties is beyond the scope of this study. Assessments for individual regions will be performed as part of
REgional Carbon Cycle Assessment and Processes (RECCAP2; Ciais et al., 2020) or already exist for selected
regions (e.g., for Europe by Petrescu et al., 2020, for Brazil by Rosan et al., 2021, for 8 selected
countries/regions in comparison to inventory data by Schwingshackl et al., subm.).
National GHG inventory data (NGHGI) under the LULUCF sector or data submitted by countries to FAOSTAT
differ from the global models' definition of $E_{LUC}$ we adopt here in that in the NGHGI reporting, the natural
fluxes ($S_{LAND}$) are counted towards $E_{LUC}$ when they occur on managed land (Grassi et al., 2018). In order to
compare our results to the NGHGI approach, we perform a re-mapping of our $E_{LUC}$ estimate by including the
$S_{LAND}$ over managed forest from the DGVMs simulations (following Grassi et al., 2021) to the bookkeeping
$E_{LUC}$ estimate (see Appendix C.2.3). For the 2012-2021 period, we estimate that 1.8 GtC yr$^{-1}$ of $S_{LAND}$ occurred
on managed forests and is then reallocated to $E_{LUC}$ here, as done in the NGHGI method. Doing so, our mean
estimate of $E_{LUC}$ is reduced from a source of 1.2 GtC to a sink of 0.6 GtC, very similar to the NGHGI estimate
of a 0.5 GtC sink (Table 9). The re-mapping approach has been shown to be generally applicable also on
country-level (Schwingshackl et al., subm.). Country-level analysis suggests, e.g., that the bookkeeping mean
estimates higher deforestation emissions than the national report in Indonesia, but estimates less $CO_2$ removal
by afforestation than the national report in China. The fraction of the natural $CO_2$ sinks that the NGHGI



estimates include differs substantially across countries, related to varying proportions of managed vs all forest
areas (Schwingshackl et al., subm.).
Though estimates between GHGI, FAOSTAT, individual process-based models and the mapped budget
estimates still differ in value and need further analysis, the approach taken here provides a possibility to relate
the global models' and NGHGI approach to each other routinely and thus link the anthropogenic carbon budget
estimates of land $CO_2$ fluxes directly to the Global Stocktake, as part of UNFCCC Paris Agreement.
**3.2.3    Final year 2021**
The global $CO_2$ emissions from land-use change are estimated as 1.1 ± 0.7 GtC in 2021, similar to the 2020
estimate. However, confidence in the annual change remains low.
Land-use change and related emissions may have been affected by the COVID-19 pandemic (e.g. Poulter et al.,
2021). During the period of the pandemic, environmental protection policies and their implementation may have
been weakened in Brazil (Vale et al., 2021). In other countries, too, monitoring capacities and legal enforcement
of measures to reduce tropical deforestation have been reduced due to budget restrictions of environmental
agencies or impairments to ground-based monitoring that prevents land grabs and tenure conflicts (Brancalion et
al., 2020, Amador-Jiménez et al., 2020). Effects of the pandemic on trends in fire activity or forest cover
changes are hard to separate from those of general political developments and environmental changes and the
long-term consequences of disruptions in agricultural and forestry economic activities (e.g., Gruère and Brooks,
2020; Golar et al., 2020; Beckman and Countryman, 2021) remain to be seen. Overall, there is limited evidence
so far that COVID-19 was a key driver of changes in LULUCF emissions at global scale. Impacts vary across
countries and deforestation-curbing and enhancing factors may partly compensate each other (Wunder et al.,

861 2021).

**3.2.4    Year 2022 Projection**
In Indonesia, peat fire emissions are very low, potentially related to a relatively wet dry season (GFED4.1s, van
der Werf et al., 2017). In South America, the trajectory of tropical deforestation and degradation fires resembles
the long-term average; global emissions from tropical deforestation and degradation fires were estimated to be
116 TgC by August 23 (GFED4.1s, van der Werf et al., 2017). Our preliminary estimate of $E_{LUC}$ for 2022 is
substantially lower than the 2012-2021 average, which saw years of anomalously dry conditions in Indonesia
and high deforestation fires in South America (Friedlingstein et al., 2022). Based on the fire emissions until
August 23, we expect $E_{LUC}$ emissions of around 1.0 GtC in 2022. Note that although our extrapolation is based
on tropical deforestation and degradation fires, degradation attributable to selective logging, edge-effects or
fragmentation will not be captured. Further, deforestation and fires in deforestation zones may become more
disconnected, partly due changes in legislation in some regions. For example, Van Wees et al. (2021) found that
the contribution from fires to forest loss decreased in the Amazon and in Indonesia over the period of 2003-
2018. More recent years, however, saw an uptick in the Amazon again (Tyukavina et al., 2022 with update) and
more work is needed to understand fire-deforestation relations.



The fires in Mediterranean Europe in summer 2022 and in the U.S. in spring 2022, though above average for
those regions, only contribute a small amount to global emissions. However, they were unrelated to land-use
change and are thus not attributed to $E_{LUC}$, but would be captured by the natural land sink.
Land use dynamics may be influenced by the disruption to the global food market associated with the war in
Ukraine, but scientific evidence so far is very limited. High food prices, which preceded but were exacerbated
by the war (Torero 2022), are generally linked to higher deforestation (Angelsen and Kaimowitz 1999), while
high prices on agricultural inputs such as fertilizers and fuel, which are also under pressure from embargoes,
may impair yields.
**3.3    Total anthropogenic emissions**
Cumulative anthropogenic $CO_2$ emissions for 1850-2021 totalled $670 \pm 65$ GtC ($2455 \pm 240$ GtCO$_2$), of which
70% (470 GtC) occurred since 1960 and 33% (220 GtC) since 2000 (Table 6 and 8). Total anthropogenic
emissions more than doubled over the last 60 years, from $4.5 \pm 0.7$ GtC yr$^{-1}$ for the decade of the 1960s to an
average of $10.9 \pm 0.8$ GtC yr$^{-1}$ during 2012-2021, and reaching $11.1 \pm 0.9$ GtC ($40.8 \pm 3.3$ GtCO$_2$) in 2021. For
2022, we project global total anthropogenic $CO_2$ emissions from fossil and land use changes to be also around
11.1 GtC (40.9 GtCO$_2$).
During the historical period 1850-2021, 30% of historical emissions were from land use change and 79% from
fossil emissions. However, fossil emissions have grown significantly since 1960 while land use changes have
not, and consequently the contributions of land use change to total anthropogenic emissions were smaller during
recent periods (18% during the period 1960-2021 and 11% during 2012-2021).
**3.4    Atmospheric $CO_2$**
**3.4.1    Historical period 1850-2021**
Atmospheric $CO_2$ concentration was approximately 277 parts per million (ppm) in 1750 (Joos and Spahni,
2008), reaching 300 ppm in the 1910s, 350 ppm in the late 1980s, and reaching $414.71 \pm 0.1$ ppm in 2021
(Dlugokencky and Tans, 2022); Figure 1). The mass of carbon in the atmosphere increased by 48% from 590
GtC in 1750 to 879 GtC in 2021. Current $CO_2$ concentrations in the atmosphere are unprecedented in the last 2
million years and the current rate of atmospheric $CO_2$ increase is at least 10 times faster than at any other time
during the last 800,000 years (Canadell et al., 2021).
**3.4.2    Recent period 1960-2021**
The growth rate in atmospheric $CO_2$ level increased from $1.7 \pm 0.07$ GtC yr$^{-1}$ in the 1960s to $5.2 \pm 0.02$ GtC yr$^{-1}$
during 2012-2022 with important decadal variations (Table 6, Figure 3 and Figure 4). During the last decade
(2012-2021), the growth rate in atmospheric $CO_2$ concentration continued to increase, albeit with large
interannual variability (Figure 4).
The airborne fraction (AF), defined as the ratio of atmospheric $CO_2$ growth rate to total anthropogenic
emissions:
$AF = G_{ATM} / (E_{FOS} + E_{LUC})$                    (2)



provides a diagnostic of the relative strength of the land and ocean carbon sinks in removing part of the
anthropogenic $CO_2$ perturbation. The evolution of AF over the last 60 years shows no significant trend,
remaining at around 44%, albeit showing a large interannual and decadal variability driven by the year-to-year
variability in $G_{ATM}$ (Figure 9). The observed stability of the airborne fraction over the 1960-2020 period
indicates that the ocean and land $CO_2$ sinks have been removing on average about 55% of the anthropogenic
emissions (see sections 3.5 and 3.6).

### 917   3.4.3   Final year 2021

The growth rate in atmospheric $CO_2$ concentration was $5.2 \pm 0.2$ GtC ($2.46 \pm 0.08$ ppm) in 2021 (Figure 4;
Dlugokencky and Tans, 2022), slightly above the 2020 growth rate (5.0 GtC) but similar to the 2011-2020
average (5.2 GtC).

### 921   3.4.4   Year 2022 Projection

The 2022 growth in atmospheric $CO_2$ concentration ($G_{ATM}$) is projected to be about 5.5 GtC (2.58 ppm) based
on GLO observations until August 2022, bringing the atmospheric $CO_2$ concentration to an expected level of
417.3 ppm averaged over the year, 51% over the pre-industrial level.

### 925   3.5   Ocean Sink

### 926   3.5.1   Historical period 1850-2021

Cumulated since 1850, the ocean sink adds up to $175 \pm 35$ GtC, with more than two thirds of this amount (120
GtC) being taken up by the global ocean since 1960. Over the historical period, the ocean sink increased in pace
with the anthropogenic emissions exponential increase (Figure 3b). Since 1850, the ocean has removed 26% of
total anthropogenic emissions.

### 931   3.5.2   Recent period 1960-2021

The ocean $CO_2$ sink increased from $1.1 \pm 0.4$ GtC yr$^{-1}$ in the 1960s to $2.9 \pm 0.4$ GtC yr$^{-1}$ during 2012-2021
(Table 6), with interannual variations of the order of a few tenths of GtC yr$^{-1}$ (Figure 10). The ocean-borne
fraction ($S_{OCEAN}/(E_{FOS}+E_{LUC})$) has been remarkably constant around 25% on average (Figure 9). Variations
around this mean illustrate decadal variability of the ocean carbon sink. So far, there is no indication of a
decrease in the ocean-borne fraction from 1960 to 2021. The increase of the ocean sink is primarily driven by
the increased atmospheric $CO_2$ concentration, with the strongest $CO_2$ induced signal in the North Atlantic and
the Southern Ocean (Figure 11a). The effect of climate change is much weaker, reducing the ocean sink globally
by $0.11 \pm 0.09$ GtC yr$^{-1}$ or 4.2% (2012-2021, nine models simulate a weakening of the ocean sink by climate
change, range -3.2 to -8.9% and one model a strengthening by 4.8%), and does not show clear spatial patterns
across the GOBMs ensemble (Figure 11b). This is the combined effect of change and variability in all
atmospheric forcing fields, previously attributed to wind and temperature changes in one model (LeQuéré et al.,

943   2010).

The global net air-sea $CO_2$ flux is a residual of large natural and anthropogenic $CO_2$ fluxes into and out of the
ocean with distinct regional and seasonal variations (Figure 6 and B1). Natural fluxes dominate on regional



scales, but largely cancel out when integrated globally (Gruber et al., 2009). Mid-latitudes in all basins and the
high-latitude North Atlantic dominate the ocean $CO_2$ uptake where low temperatures and high wind speeds
facilitate $CO_2$ uptake at the surface (Takahashi et al., 2009). In these regions, formation of mode, intermediate
and deep-water masses transport anthropogenic carbon into the ocean interior, thus allowing for continued $CO_2$
uptake at the surface. Outgassing of natural $CO_2$ occurs mostly in the tropics, especially in the equatorial
upwelling region, and to a lesser extent in the North Pacific and polar Southern Ocean, mirroring a well-
established understanding of regional patterns of air-sea $CO_2$ exchange (e.g., Takahashi et al., 2009, Gruber et
al., 2009). These patterns are also noticeable in the Surface Ocean CO2 Atlas (SOCAT) dataset, where an ocean
$fCO_2$ value above the atmospheric level indicates outgassing (Figure B1). This map further illustrates the data-
sparsity in the Indian Ocean and the southern hemisphere in general.
Interannual variability of the ocean carbon sink is driven by climate variability with a first-order effect from a
stronger ocean sink during large El Niño events (e.g., 1997-1998) (Figure 10; Rödenbeck et al., 2014, Hauck et
al., 2020). The GOBMs show the same patterns of decadal variability as the mean of the $fCO_2$-based data
products, with a stagnation of the ocean sink in the 1990s and a strengthening since the early 2000s (Figure 10,
Le Quéré et al., 2007; Landschützer et al., 2015, 2016; DeVries et al., 2017; Hauck et al., 2020; McKinley et al.,
2020). Different explanations have been proposed for this decadal variability, ranging from the ocean's response
to changes in atmospheric wind and pressure systems (e.g., Le Quéré et al., 2007, Keppler and Landschützer,
2019), including variations in upper ocean overturning circulation (DeVries et al., 2017) to the eruption of
Mount Pinatubo and its effects on sea surface temperature and slowed atmospheric $CO_2$ growth rate in the 1990s
(McKinley et al., 2020). The main origin of the decadal variability is a matter of debate with a number of studies
initially pointing to the Southern Ocean (see review in Canadell et al., 2021), but also contributions from the
North Atlantic and North Pacific (Landschützer et al., 2016, DeVries et al., 2019), or a global signal (McKinley
et al., 2020) were proposed.
Although all individual GOBMs and data-products fall within the observational constraint, the ensemble means
of GOBMs, and data-products adjusted for the riverine flux diverge over time with a mean offset increasing
from 0.28 GtC yr$^{-1}$ in the 1990s to 0.61 GtC yr$^{-1}$ in the decade 2012-2021 and reaching 0.79 GtC yr$^{-1}$ in 2021.
The $S_{OCEAN}$ positive trend over time diverges by a factor two since 2002 (GOBMs: $0.28 \pm 0.07$ GtC yr$^{-1}$ per
decade, data-products: $0.61 \pm 0.17$ GtC yr$^{-1}$ per decade, $S_{OCEAN}$: 0.45 GtC yr$^{-1}$ per decade) and by a factor of
three since 2010 (GOBMs: $0.21 \pm 0.14$ GtC yr$^{-1}$ per decade, data-products: $0.66 \pm 0.38$ GtC yr$^{-1}$ per decade,
$S_{OCEAN}$: 0.44 GtC yr$^{-1}$ per decade). The GOBMs estimate is slightly higher (<0.1 GtC yr$^{-1}$) than in the previous
global carbon budget (Friedlingstein et al., 2022), because one new model is included (CESM2) and four models
revised their estimate upwards (CESM-ETHZ, CNRM, FESOM2-REcoM, PlankTOM). The data-product
estimate is higher by about 0.1 GtC yr$^{-1}$ compared to Friedlingstein et al. (2022) as a result of an upward
correction in three products (Jena-MLS, MPI-SOMFFN, OS-ETHZ-Gracer), the submission of LDEO-HPD
which is above average, the non-availability of the CSIR product, and the small upward correction of the river
flux adjustment.
The discrepancy between the two types of estimates stems mostly from a larger Southern Ocean sink in the data-
products prior to 2001, and from a larger $S_{OCEAN}$ trend in the northern and southern extra-tropics since then
(Figure 13). Note that the location of the mean offset (but not its trend) depends strongly on the choice of



regional river flux adjustment and would occur in the tropics rather than in the Southern Ocean when using the
dataset of Lacroix et al. (2020) instead of Aumont et al. (2001). Other possible explanations for the discrepancy
in the Southern Ocean could be missing winter observations and data sparsity in general (Bushinsky et al., 2019,
Gloege et al., 2021), or model biases (as indicated by the large model spread in the South, Figure 13, and the
larger model-data mismatch, Figure B2).
In GCB releases until 2021, the ocean sink 1959-1989 was only estimated by GOBMs due to the absence of
$fCO_2$ observations. Now, the first data-based estimates extending back to 1957/58 are becoming available (Jena-
MLS, Rödenbeck et al., 2022, LDEO-HPD, Bennington et al., 2022; Gloege et al. 2022). These are based on a
multi-linear regression of $pCO_2$ with environmental predictors (Rödenbeck et al., 2022, included here) or on
model-data $pCO_2$ misfits and their relation to environmental predictors (Bennington et al., 2022). The Jena-MLS
estimate falls well within the range of GOBM estimates and has a correlation of 0.98 with $S_{OCEAN}$ (1959-2021 as
well as 1959-1989). It agrees well on the mean $S_{OCEAN}$ estimate since 1977 with a slightly higher amplitude of
variability (Figure 10). Until 1976, Jena-MLS is 0.2-0.3 GtCyr$^{-1}$ below the central $S_{OCEAN}$ estimate. The
agreement especially on phasing of variability is impressive, and the discrepancies in the mean flux 1959-1976
could be explained by an overestimated trend of Jena-MLS (Rödenbeck et al., 2022). Bennington et al. (2022)
report a larger flux into the pre-1990 ocean than in Jena-MLS.
The reported $S_{OCEAN}$ estimate from GOBMs and data-products is 2.1 ± 0.4 GtC yr$^{-1}$ over the period 1994 to
2007, which is in agreement with the ocean interior estimate of 2.2 ± 0.4 GtC yr$^{-1}$ which accounts for the
climate effect on the natural $CO_2$ flux of −0.4 ± 0.24 GtC yr$^{-1}$ (Gruber et al., 2019) to match the definition of
$S_{OCEAN}$ used here (Hauck et al., 2020). This comparison depends critically on the estimate of the climate effect
on the natural $CO_2$ flux, which is smaller from the GOBMs (-0.1 GtC yr$^{-1}$) than in Gruber et al. (2019).
Uncertainties of these two estimates would also overlap when using the GOBM estimate of the climate effect on
the natural $CO_2$ flux.
During 2010-2016, the ocean $CO_2$ sink appears to have intensified in line with the expected increase from
atmospheric $CO_2$ (McKinley et al., 2020). This effect is stronger in the $fCO_2$-based data products (Figure 10,
ocean sink 2016 minus 2010, GOBMs: +0.42 ± 0.09 GtC yr$^{-1}$, data-products: +0.52 ± 0.22 GtC yr$^{-1}$). The
reduction of -0.09 GtC yr$^{-1}$ (range: -0.39 to +0.01 GtC yr$^{-1}$) in the ocean $CO_2$ sink in 2017 is consistent with the
return to normal conditions after the El Niño in 2015/16, which caused an enhanced sink in previous years.
After 2017, the GOBMs ensemble mean suggests the ocean sink levelling off at about 2.6 GtC yr$^{-1}$, whereas the
data-products' estimate increases by 0.24 ± 0.17 GtC yr$^{-1}$ over the same period.
**3.5.3    Final year 2021**
The estimated ocean $CO_2$ sink was 2.9 ± 0.4 GtC in 2021. This is a decrease of 0.12 GtC compared to 2020, in
line with the expected sink weakening from persistent La Niña conditions. GOBM and data-product estimates
consistently result in a stagnation of $S_{OCEAN}$ (GOBMs: -0.09 ±0.15 GtC, data-products: -0.15 ±0.24 GtC).  Seven
models and six data products show a decrease in $S_{OCEAN}$ (GOBMs down to -0.31 GtC, data-products down to -
0.58 GtC), while three models and two data products show an increase in $S_{OCEAN}$ (GOBMs up to 0.15 GtC, data-
products up to 0.12 GtC; Figure 10). The data-products have a larger uncertainty at the tails of the reconstructed
time series (e.g., Watson et al., 2020). Specifically, the data-products' estimate of the last year is regularly





adjusted in the following release owing to the tail effect and an incrementally increasing data availability with 1-
5 years lag (Figure 10 inset).
**3.5.4    Year 2022 Projection**
Using a feed-forward neural network method (see section 2.4) we project an ocean sink of 2.9 GtC for 2022.
This is similar to the year 2021 as the La Niña conditions persist in 2022.
**3.5.5    Model Evaluation**
The additional simulation D allows to separate the anthropogenic carbon component (steady state and non-
steady state, sim D - sim A) and to compare the model flux and DIC inventory change directly to the interior
ocean estimate of Gruber et al. (2019) without further assumptions. The GOBMs ensemble average of
anthropogenic carbon inventory changes 1994-2007 amounts to 2.2 GtC yr$^{-1}$ and is thus lower than the 2.6 ± 0.3
GtC yr$^{-1}$ estimated by Gruber et al (2019). Only four models with the highest sink estimate fall within the range
reported by Gruber et al. (2019). This suggests that most of the GOBMs underestimate anthropogenic carbon
uptake by the ocean. Analysis of Earth System Models indicate that this may be due to biases in ocean carbon
transport and mixing from the surface mixed layer to the ocean interior (Goris et al., 2018, Terhaar et al., 2021,
Bourgeois et al., 2022, Terhaar et al., 2022,), biases in the chemical buffer capacity (Revelle factor) of the ocean
(Vaittinada Ayar et al., 2022; Terhaar et al., 2022) and partly due to a late starting date of the simulations
(mirrored in atmospheric $CO_2$ chosen for the preindustrial control simulation, Table A2, Bronselaer et al., 2017,
Terhaar et al., 2022). Interestingly, and in contrast to the uncertainties in the surface $CO_2$ flux, we find the
largest mismatch in interior ocean carbon accumulation in the tropics (93% of the mismatch), with minor
contribution from the north (1%) and the south (6%). This highlights the role of interior ocean carbon
redistribution for those inventories (Khatiwala et al., 2009).
The evaluation of the ocean estimates (Figure B2) shows an RMSE from annually detrended data of 0.4 to 2.6
µatm for the seven fCO$_2$-based data products over the globe, relative to the fCO$_2$ observations from the SOCAT
v2022 dataset for the period 1990-2021. The GOBMs RMSEs are larger and range from 3.0 to 4.8 µatm. The
RMSEs are generally larger at high latitudes compared to the tropics, for both the data products and the
GOBMs. The data products have RMSEs of 0.4 to 3.2 µatm in the tropics, 0.8 to 2.8 µatm in the north, and 0.8
to 3.6 µatm in the south. Note that the data products are based on the SOCAT v2022 database, hence the
SOCAT is not an independent dataset for the evaluation of the data products. The GOBMs RMSEs are more
spread across regions, ranging from 2.5 to 3.9 µatm in the tropics, 3.1 to 6.5 µatm in the North, and 5.4 to 7.9
µatm in the South. The higher RMSEs occur in regions with stronger climate variability, such as the northern
and southern high latitudes (poleward of the subtropical gyres). The upper range of the model RMSEs have
decreased somewhat relative to Friedlingstein et al. (2022).





### 3.6 Land Sink

#### 3.6.1 Historical period 1850-2021

Cumulated since 1850, the terrestrial $CO_2$ sink amounts to $210 \pm 45$ GtC, 31% of total anthropogenic emissions. Over the historical period, the sink increased in pace with the anthropogenic emissions exponential increase (Figure 3b).

#### 3.6.2 Recent period 1960-2021

The terrestrial $CO_2$ sink increased from $1.2 \pm 0.4$ GtC yr$^{-1}$ in the 1960s to $3.1 \pm 0.6$ GtC yr$^{-1}$ during 2012-2021, with important interannual variations of up to 2 GtC yr$^{-1}$ generally showing a decreased land sink during El Niño events (Figure 8), responsible for the corresponding enhanced growth rate in atmospheric $CO_2$ concentration. The larger land $CO_2$ sink during 2012-2021 compared to the 1960s is reproduced by all the DGVMs in response to the increase in both atmospheric $CO_2$ and nitrogen deposition, and the changes in climate, and is consistent with constraints from the other budget terms (Table 5).

Over the period 1960 to present the increase in the global terrestrial $CO_2$ sink is largely attributed to the $CO_2$ fertilisation effect (Prentice et al., 2001, Piao et al., 2009), directly stimulating plant photosynthesis and increased plant water use in water limited systems, with a small negative contribution of climate change (Figure 11). There is a range of evidence to support a positive terrestrial carbon sink in response to increasing atmospheric $CO_2$, albeit with uncertain magnitude (Walker et al., 2021). As expected from theory, the greatest $CO_2$ effect is simulated in the tropical forest regions, associated with warm temperatures and long growing seasons (Hickler et al., 2008) (Figure 11a). However, evidence from tropical intact forest plots indicate an overall decline in the land sink across Amazonia (1985-2011), attributed to enhanced mortality offsetting productivity gains (Brienen et al., 2005, Hubau et al., 2020). During 2012-2021 the land sink is positive in all regions (Figure 6) with the exception of eastern Brazil, Southwest USA, Southeast Europe and Central Asia, North and South Africa, and eastern Australia, where the negative effects of climate variability and change (i.e. reduced rainfall) counterbalance $CO_2$ effects. This is clearly visible on Figure 11 where the effects of $CO_2$ (Figure 11a) and climate (Figure 11b) as simulated by the DGVMs are isolated. The negative effect of climate is the strongest in most of South America, Central America, Southwest US, Central Europe, western Sahel, southern Africa, Southeast Asia and southern China, and eastern Australia (Figure 11b). Globally, climate change reduces the land sink by $0.63 \pm 0.52$ GtC yr$^{-1}$ or 17% (2012-2021).

Since 2020 the globe has experienced La Niña conditions which would be expected to lead to an increased land carbon sink. A clear peak in the global land sink is not evident in $S_{LAND}$, and we find that a La Niña-driven increase in tropical land sink is offset by a reduced high latitude extra-tropical land sink, which may be linked to the land response to recent climate extremes. In the past years several regions experienced record-setting fire events. While global burned area has declined over the past decades mostly due to declining fire activity in savannas (Andela et al., 2017), forest fire emissions are rising and have the potential to counter the negative fire trend in savannas (Zheng et al., 2021). Noteworthy events include the 2019-2020 Black Summer event in Australia (emissions of roughly 0.2 GtC; van der Velde et al., 2021) and Siberia in 2021 where emissions approached 0.4 GtC or three times the 1997-2020 average according to GFED4s. While other regions, including



Western US and Mediterranean Europe, also experienced intense fire seasons in 2021 their emissions are
substantially lower.
Despite these regional negative effects of climate change on $S_{LAND}$, the efficiency of land to remove
anthropogenic $CO_2$ emissions has remained broadly constant over the last six decades, with a land-borne
fraction ($S_{LAND}/(E_{FOS}+E_{LUC})$) of ~30% (Figure 9).

### 3.6.3    Final year 2021

The terrestrial $CO_2$ sink from the DGVMs ensemble was 3.5 ± 0.9 GtC in 2021, slightly above the decadal
average of 3.1 ± 0.6GtC yr$^{-1}$ (Figure 4, Table 6). We note that the DGVMs estimate for 2021 is larger, but
within the uncertainty, than the 2.8 ± 0.9 GtC yr$^{-1}$ estimate from the residual sink from the global budget
($E_{FOS}+E_{LUC}-G_{ATM}-S_{OCEAN}$) (Table 5).

### 3.6.4    Year 2022 Projection

Using a feed-forward neural network method we project a land sink of 3.4 GtC for 2022, very similar to the
2021 estimate. As for the ocean sink, we attribute this to the persistence of La Niña conditions in 2022.

### 3.6.5    Model Evaluation

The evaluation of the DGVMs (Figure B3) shows generally high skill scores across models for runoff, and to a
lesser extent for vegetation biomass, GPP, and ecosystem respiration (Figure B3, left panel). Skill score was
lowest for leaf area index and net ecosystem exchange, with a widest disparity among models for soil carbon.
These conclusions are supported by a more comprehensive analysis of DGVM performance in comparison with
benchmark data (Seiler et al., 2022). Furthermore, results show how DGVM differences are often of similar
magnitude compared with the range across observational datasets.

### 3.7    Partitioning the carbon sinks

### 3.7.1    Global sinks and spread of estimates

In the period 2012-2021, the bottom-up view of total global carbon sinks provided by the GCB, $S_{OCEAN}$ for the
ocean and $S_{LAND}-E_{LUC}$ for the land (to be comparable to inversions), agrees closely with the top-down global
carbon sinks delivered by the atmospheric inversions. Figure 12 shows both total sink estimates of the last
decade split by ocean and land (including $E_{LUC}$), which match the difference between $G_{ATM}$ and $E_{FOS}$ to within
0.01-0.12 GtC yr$^{-1}$ for inverse systems, and to 0.34 GtC yr$^{-1}$ for the GCB mean. The latter represents the $B_{IM}$
discussed in Section 3.8, which by design is minimal for the inverse systems.
The distributions based on the individual models and data products reveal substantial spread but converge near
the decadal means quoted in Tables 5 and 6. Sink estimates for $S_{OCEAN}$ and from inverse systems are mostly
non-Gaussian, while the ensemble of DGVMs appears more normally distributed justifying the use of a multi-
model mean and standard deviation for their errors in the budget. Noteworthy is that the tails of the distributions
provided by the land and ocean bottom-up estimates would not agree with the global constraint provided by the





fossil fuel emissions and the observed atmospheric $CO_2$ growth rate ($E_{FOS} - G_{ATM}$). This illustrates the power of
the atmospheric joint constraint from $G_{ATM}$ and the global $CO_2$ observation network it derives from.

### 3.7.2    Total atmosphere-to-land fluxes

The total atmosphere-to-land fluxes ($S_{LAND} - E_{LUC}$), calculated here as the difference between $S_{LAND}$ from the
DGVMs and $E_{LUC}$ from the bookkeeping models, amounts to a $1.9 \pm 0.9$ GtC yr$^{-1}$ sink during 2012-2021 (Table
5). Estimates of total atmosphere-to-land fluxes ($S_{LAND} - E_{LUC}$) from the DGVMs alone ($1.5 \pm 0.5$ GtC yr$^{-1}$) are
consistent with this estimate and also with the global carbon budget constraint ($E_{FOS} - G_{ATM} - S_{OCEAN}$, $1.5 \pm 0.6$
GtC yr$^{-1}$ Table 5). For the last decade (2012-2021), the inversions estimate the net atmosphere-to-land uptake to
lie within a range of 1.1 to 1.7 GtC yr$^{-1}$, consistent with the GCB and DGVMs estimates of $S_{LAND} - E_{LUC}$ (Figure
13 top row).

### 3.7.3    Total atmosphere-to-ocean fluxes

For the 2012-2021 period, the GOBMs ($2.6 \pm 0.5$ GtC yr$^{-1}$) produce a lower estimate for the ocean sink than the
$fCO_2$-based data products ($3.2 \pm 0.6$ GtC yr$^{-1}$), which shows up in Figure 12 as a separate peak in the
distribution from the GOBMs (triangle symbols pointing right) and from the $fCO_2$-based products (triangle
symbols pointing left). Atmospheric inversions (2.7 to 3.3 GtC yr$^{-1}$) also suggest higher ocean uptake in the
recent decade (Figure 13 top row). In interpreting these differences, we caution that the riverine transport of
carbon taken up on land and outgassing from the ocean is a substantial (0.65 GtC yr$^{-1}$) and uncertain term that
separates the various methods. A recent estimate of decadal ocean uptake from observed $O_2/N_2$ ratios (Tohjima
et al., 2019) also points towards a larger ocean sink, albeit with large uncertainty (2012-2016: $3.1 \pm 1.5$ GtC yr$^{-}$
$^{1}$).

### 3.7.4    Regional breakdown and interannual variability

Figure 13 also shows the latitudinal partitioning of the total atmosphere-to-surface fluxes excluding fossil $CO_2$
emissions ($S_{OCEAN} + S_{LAND} - E_{LUC}$) according to the multi-model average estimates from GOBMs and ocean
$fCO_2$-based products ($S_{OCEAN}$) and DGVMs ($S_{LAND} - E_{LUC}$), and from atmospheric inversions ($S_{OCEAN}$ and $S_{LAND}$
$- E_{LUC}$).

#### 3.7.4.1    North

Despite being one of the most densely observed and studied regions of our globe, annual mean carbon sink
estimates in the northern extra-tropics (north of 30°N) continue to differ. The atmospheric inversions suggest an
atmosphere-to-surface sink ($S_{OCEAN} + S_{LAND} - E_{LUC}$) for 2012-2021 of 2.0 to 3.2 GtC yr$^{-1}$, which is higher than
the process models' estimate of $2.2 \pm 0.4$ GtC yr$^{-1}$ (Figure 13). The GOBMs ($1.2 \pm 0.2$ GtC yr$^{-1}$), $fCO_2$-based
data products ($1.4 \pm 0.1$ GtC yr$^{-1}$), and inversion systems (0.9 to 1.4 GtC yr$^{-1}$) produce consistent estimates of
the ocean sink. Thus, the difference mainly arises from the total land flux ($S_{LAND} - E_{LUC}$) estimate, which is 1.0
$\pm 0.4$ GtC yr$^{-1}$ in the DGVMs compared to 0.6 to 2.0 GtC yr$^{-1}$ in the atmospheric inversions (Figure 13, second
row).
Discrepancies in the northern land fluxes conforms with persistent issues surrounding the quantification of the
drivers of the global net land $CO_2$ flux (Arneth et al., 2017; Huntzinger et al., 2017; O'Sullivan et al., 2022) and



the distribution of atmosphere-to-land fluxes between the tropics and high northern latitudes (Baccini et al.,
2017; Schimel et al., 2015; Stephens et al., 2007; Ciais et al. 2019; Gaubert et al., 2019).
In the northern extratropics, the process models, inversions, and $fCO_2$-based data products consistently suggest
that most of the variability stems from the land (Figure 13). Inversions generally estimate similar interannual
variations (IAV) over land to DGVMs (0.30 – 0.37 vs 0.17 – 0.69 GtC yr$^{-1}$, averaged over 1990-2021), and they
have higher IAV in ocean fluxes (0.05 – 0.09 GtC yr$^{-1}$) relative to GOBMs (0.02 – 0.06 GtC yr$^{-1}$, Figure B2),
and $fCO_2$-based data products (0.03 – 0.09 GtC yr$^{-1}$).

### 3.7.4.2    Tropics

In the tropics (30°S-30°N), both the atmospheric inversions and process models estimate a total carbon balance
($S_{OCEAN}$+$S_{LAND}$-$E_{LUC}$) that is close to neutral over the past decade. The GOBMs (0.06 ± 0.34 GtC yr$^{-1}$), $fCO_2$-
based data products (0.00 ± 0.06 GtC yr$^{-1}$), and inversion systems (-0.2 to 0.5 GtC yr$^{-1}$) all indicate an
approximately neutral tropical ocean flux (see Figure B1 for spatial patterns). DGVMs indicate a net land sink
($S_{LAND}$-$E_{LUC}$) of 0.5 ± 0.3 GtC yr$^{-1}$, whereas the inversion systems indicate a net land flux between -0.9 and 0.7
GtC yr$^{-1}$, though with high uncertainty (Figure 13, third row).
The tropical lands are the origin of most of the atmospheric $CO_2$ interannual variability (Ahlström et al., 2015),
consistently among the process models and inversions (Figure 13). The interannual variability in the tropics is
similar among the ocean data products (0.07 – 0.16 GtC yr−1) and the GOBMs (0.07 – 0.16 GtC yr$^{-1}$, Figure
B2), which is the highest ocean sink variability of all regions. The DGVMs and inversions indicate that
atmosphere-to-land $CO_2$ fluxes are more variable than atmosphere-to-ocean $CO_2$ fluxes in the tropics, with
interannual variability of 0.5 to 1.1 and 0.8 to 1.0 GtC yr$^{-1}$ for DGVMs and inversions, respectively.

### 3.7.4.3    South

In the southern extra-tropics (south of 30°S), the atmospheric inversions suggest a total atmosphere-to-surface
sink ($S_{OCEAN}$+$S_{LAND}$-$E_{LUC}$) for 2012-2021 of 1.6 to 1.9 GtC yr$^{-1}$, slightly higher than the process models'
estimate of 1.4 ± 0.3 GtC yr$^{-1}$ (Figure 13). An approximately neutral total land flux ($S_{LAND}$-$E_{LUC}$) for the
southern extra-tropics is estimated by both the DGVMs (0.02 ± 0.06 GtC yr$^{-1}$) and the inversion systems (sink of
-0.2 to 0.2 GtC yr$^{-1}$). This means nearly all carbon uptake is due to oceanic sinks south of 30°S.  The Southern
Ocean flux in the $fCO_2$-based data products (1.8 ± 0.1 GtC yr$^{-1}$) and inversion estimates (1.6 to 1.9 GtCyr-1) is
higher than in the GOBMs (1.4 ± 0.3 GtC yr$^{-1}$) (Figure 13, bottom row). This discrepancy in the mean flux is
likely explained by the uncertainty in the regional distribution of the river flux adjustment (Aumont et al., 2001,
Lacroix et al., 2020) applied to $fCO_2$-based data products and inverse systems to isolate the anthropogenic
$S_{OCEAN}$ flux. Other possibly contributing factors are that the data-products potentially underestimate the winter
$CO_2$ outgassing south of the Polar Front (Bushinsky et al., 2019) and model biases. $CO_2$ fluxes from this region
are more sparsely sampled by all methods, especially in wintertime (Figure B1).
The interannual variability in the southern extra-tropics is low because of the dominance of ocean areas with
low variability compared to land areas. The split between land ($S_{LAND}$-$E_{LUC}$) and ocean ($S_{OCEAN}$) shows a
substantial contribution to variability in the south coming from the land, with no consistency between the
DGVMs and the inversions or among inversions. This is expected due to the difficulty of separating exactly the



land and oceanic fluxes when viewed from atmospheric observations alone. The $S_{OCEAN}$ interannual variability
was found to be higher in the $fCO_2$-based data products (0.09 to 0.19 GtC yr−1) compared to GOBMs (0.03 to
0.06 GtC yr−1) in 1990-2021 (Figure B2). Model subsampling experiments recently illustrated that observation-
based products may overestimate decadal variability in the Southern Ocean carbon sink by 30% due to data
sparsity, based on one data product with the highest decadal variability (Gloege et al., 2021).
**3.7.4.4 Tropical vs northern land uptake**
A continuing conundrum is the partitioning of the global atmosphere-land flux between the northern hemisphere
land and the tropical land (Stephens et al., 2017; Pan et al., 2011; Gaubert et al., 2019). It is of importance
because each region has its own history of land-use change, climate drivers, and impact of increasing
atmospheric $CO_2$ and nitrogen deposition. Quantifying the magnitude of each sink is a prerequisite to
understanding how each individual driver impacts the tropical and mid/high-latitude carbon balance.
We define the North-South (N-S) difference as net atmosphere-land flux north of 30°N minus the net
atmosphere-land flux south of 30°N. For the inversions, the N-S difference ranges from 0.1 GtC yr$^{-1}$ to 2.9 GtC
yr$^{-1}$ across this year's inversion ensemble with a preference across models for either a smaller Northern land
sink with a near neutral tropical land flux (medium N-S difference), or a large Northern land sink and a tropical
land source (large N-S difference).
In the ensemble of DGVMs the N-S difference is $0.6 \pm 0.5$ GtC yr$^{-1}$, a much narrower range than the one from
inversions. Only two DGVMs have a N-S difference larger than 1.0 GtC yr$^{-1}$. The larger agreement across
DGVMs than across inversions is to be expected as there is no correlation between Northern and Tropical land
sinks in the DGVMs as opposed to the inversions where the sum of the two regions being well-constrained leads
to an anti-correlation between these two regions. The much smaller spread in the N-S difference between the
DGVMs could help to scrutinise the inverse systems further. For example, a large northern land sink and a
tropical land source in an inversion would suggest a large sensitivity to $CO_2$ fertilisation (the dominant factor
driving the land sinks) for Northern ecosystems, which would be not mirrored by tropical ecosystems. Such a
combination could be hard to reconcile with the process understanding gained from the DGVMs ensembles and
independent measurements (e.g. Free Air $CO_2$ Enrichment experiments). Such investigations will be further
pursued in the upcoming assessment from REgional Carbon Cycle Assessment and Processes (RECCAP2; Ciais
et al., 2020).
**3.8    Closing the Global Carbon Cycle**
**3.8.1    Partitioning of Cumulative Emissions and Sink Fluxes**
The global carbon budget over the historical period (1850-2021) is shown in Figure 3.
Emissions during the period 1850-2021 amounted to $670 \pm 65$ GtC and were partitioned among the atmosphere
($275 \pm 5$ GtC; 41%), ocean ($175 \pm 35$ GtC; 26%), and the land ($210 \pm 45$ GtC; 31%). The cumulative land sink
is almost equal to the cumulative land-use emissions ($200 \pm 60$ GtC), making the global land nearly neutral over
the whole 1850-2021 period.



The use of nearly independent estimates for the individual terms of the global carbon budget shows a cumulative
budget imbalance of 15 GtC (2% of total emissions) during 1850-2021 (Figure 3, Table 8), which, if correct,
suggests that emissions could be slightly too high by the same proportion (2%) or that the combined land and
ocean sinks are slightly underestimated (by about 3%), although these are well within the uncertainty range of
each component of the budget. Nevertheless, part of the imbalance could originate from the estimation of
significant increase in $E_{FOS}$ and $E_{LUC}$ between the mid 1920s and the mid 1960s which is unmatched by a similar
growth in atmospheric $CO_2$ concentration as recorded in ice cores (Figure 3). However, the known loss of
additional sink capacity of 30-40 GtC (over the 1850-2020 period) due to reduced forest cover has not been
accounted for in our method and would exacerbate the budget imbalance (Section 2.7.4).
For the more recent 1960-2021 period where direct atmospheric $CO_2$ measurements are available, total
emissions ($E_{FOS}$ + $E_{LUC}$) amounted to 470 ± 50 GtC, of which 385 ± 20 GtC (82%) were caused by fossil $CO_2$
emissions, and 85 ± 45 GtC (18%) by land-use change (Table 8). The total emissions were partitioned among
the atmosphere (210 ± 5 GtC; 45%), ocean (120 ± 25 GtC; 26%), and the land (145 ± 30 GtC; 30%), with a near
zero (-5 GtC) unattributed budget imbalance. All components except land-use change emissions have
significantly grown since 1960, with important interannual variability in the growth rate in atmospheric $CO_2$
concentration and in the land $CO_2$ sink (Figure 4), and some decadal variability in all terms (Table 6).
Differences with previous budget releases are documented in Figure B5.
The global carbon budget averaged over the last decade (2012-2021) is shown in Figure 2, Figure 14 (right
panel) and Table 6. For this period, 89% of the total emissions ($E_{FOS}$ + $E_{LUC}$) were from fossil $CO_2$ emissions
($E_{FOS}$), and 11% from land-use change ($E_{LUC}$). The total emissions were partitioned among the atmosphere
(48%), ocean (26%) and land (29%), with a near-zero unattributed budget imbalance (~3%). For single years,
the budget imbalance can be larger (Figure 4). For 2021, the combination of our estimated sources (10.9 ± 0.9
GtC yr$^{-1}$) and sinks (11.6 ± 1.0 GtC yr$^{-1}$) leads to a $B_{IM}$ of -0.6 GtC, suggesting a slight underestimation of the
anthropogenic sources, and/or an overestimation of the combined land and ocean sinks
**3.8.2    Carbon Budget Imbalance trend and variability**
The carbon budget imbalance ($B_{IM}$; Eq. 1, Figure 4) quantifies the mismatch between the estimated total
emissions and the estimated changes in the atmosphere, land, and ocean reservoirs. The mean budget imbalance
from 1960 to 2021 is very small (4.6 GtC over the period, i.e. average of 0.07 GtC yr$^{-1}$) and shows no trend over
the full time series (Figure 4). The process models (GOBMs and DGVMs) and data-products have been selected
to match observational constraints in the 1990s, but no further constraints have been applied to their
representation of trend and variability. Therefore, the near-zero mean and trend in the budget imbalance is seen
as evidence of a coherent community understanding of the emissions and their partitioning on those time scales
(Figure 4). However, the budget imbalance shows substantial variability of the order of ±1 GtC yr$^{-1}$, particularly
over semi-decadal time scales, although most of the variability is within the uncertainty of the estimates. The
positive carbon imbalance during the 1960s, and early 1990s, indicates that either the emissions were
overestimated, or the sinks were underestimated during these periods. The reverse is true for the 1970s, and to a
lower extent for the 1980s and 2012-2021 period (Figure 4, Table 6).
We cannot attribute the cause of the variability in the budget imbalance with our analysis, we only note that the
budget imbalance is unlikely to be explained by errors or biases in the emissions alone because of its large semi-
decadal variability component, a variability that is untypical of emissions and has not changed in the past 60
years despite a near tripling in emissions (Figure 4). Errors in $S_{LAND}$ and $S_{OCEAN}$ are more likely to be the main
cause for the budget imbalance, especially on interannual to semi-decadal timescales. For example,
underestimation of the $S_{LAND}$ by DGVMs has been reported following the eruption of Mount Pinatubo in 1991
possibly due to missing responses to changes in diffuse radiation (Mercado et al., 2009). Although since
GCB2021 we accounted for aerosol effects on solar radiation quantity and quality (diffuse vs direct), most
DGVMs only used the former as input (i.e., total solar radiation) (Table A1). Thus, the ensemble mean may not
capture the full effects of volcanic eruptions, i.e. associated with high light scattering sulphate aerosols, on the
land carbon sink (O'Sullivan et al., 2021). DGVMs are suspected to overestimate the land sink in response to
the wet decade of the 1970s (Sitch et al., 2008). Quasi-decadal variability in the ocean sink has also been
reported, with all methods agreeing on a smaller than expected ocean $CO_2$ sink in the 1990s and a larger than
expected sink in the 2000s (Figure 10; Landschützer et al., 2016, DeVries et al., 2019, Hauck et al., 2020,
McKinley et al., 2020). Errors in sink estimates could also be driven by errors in the climatic forcing data,
particularly precipitation for $S_{LAND}$ and wind for $S_{OCEAN}$.  Also, the $B_{IM}$ shows substantial departure from zero on
yearly time scales (Figure 4e), highlighting unresolved variability of the carbon cycle, likely in the land sink
($S_{LAND}$), given its large year to year variability (Figure 4d and 8).
Both the budget imbalance ($B_{IM}$, Table 6) and the residual land sink from the global budget ($E_{FOS}+E_{LUC}-G_{ATM}-$
$S_{OCEAN}$, Table 5) include an error term due to the inconsistencies that arises from using $E_{LUC}$ from bookkeeping
models, and $S_{LAND}$ from DGVMs, most notably the loss of additional sink capacity (see section 2.7). Other
differences include a better accounting of land use changes practices and processes in bookkeeping models than
in DGVMs, or the bookkeeping models error of having present-day observed carbon densities fixed in the past.
That the budget imbalance shows no clear trend towards larger values over time is an indication that these
inconsistencies probably play a minor role compared to other errors in $S_{LAND}$ or $S_{OCEAN}$.
Although the budget imbalance is near zero for the recent decades, it could be due to compensation of errors.
We cannot exclude an overestimation of $CO_2$ emissions, particularly from land-use change, given their large
uncertainty, as has been suggested elsewhere (Piao et al., 2018), combined with an underestimate of the sinks. A
larger DGVM ($S_{LAND}-E_{LUC}$) over the extra-tropics would reconcile model results with inversion estimates for
fluxes in the total land during the past decade (Figure 13; Table 5). Likewise, a larger $S_{OCEAN}$ is also possible
given the higher estimates from the data-products (see section 3.1.2, Figure 10 and Figure 13), the
underestimation of interior ocean anthropogenic carbon accumulation in the GOBMs (section 3.5.5), and the
recently suggested upward adjustments of the ocean carbon sink in Earth System Models (Terhaar et al., 2022),
and in data-products, here related to a potential temperature bias and skin effects (Watson et al., 2020, Dong et
al., 2022, Figure 10). If $S_{OCEAN}$ were to be based on data-products alone, with all data-products including this
adjustment, this would result in a 2012-2021  $S_{OCEAN}$ of 3.8 GtC yr$^{-1}$ (Dong et al., 2022) or >4 GtC yr$^{-1}$ (Watson
et al., 2020), i.e., outside of the range supported by the atmospheric inversions and with an implied negative $B_{IM}$
of more than -1 GtC yr$^{-1}$ indicating that a closure of the budget could only be achieved with either anthropogenic
emissions being significantly larger and/or the net land sink being substantially smaller than estimated here.
More integrated use of observations in the Global Carbon Budget, either on their own or for further constraining
model results, should help resolve some of the budget imbalance (Peters et al., 2017).



## 4    Tracking progress towards mitigation targets

The average growth in global fossil $CO_2$ emissions peaked at +3% per year during the 2000s, driven by the rapid
growth in emissions in China. In the last decade, however, the global growth rate has slowly declined, reaching
a low +0.5% per year over 2012-2021 (including the 2020 global decline and the 2021 emissions rebound).
While this slowdown in global fossil $CO_2$ emissions growth is welcome, it is far from the emission decrease
needed to be consistent with the temperature goals of the Paris Agreement.
Since the 1990s, the average growth rate of fossil $CO_2$ emissions has continuously declined across the group of
developed countries of the Organisation for Economic Co-operation and Development (OECD), with emissions
peaking in around 2005 and now declining at around 1% $yr^{-1}$ (Le Quéré et al., 2021). In the decade 2012-2021,
territorial fossil $CO_2$ emissions decreased significantly (at the 95% confidence level) in 24 countries whose
economies grew significantly (also at the 95% confidence level): Belgium, Croatia, Czech Republic, Denmark,
Estonia, Finland, France, Germany, Hong Kong, Israel, Italy, Japan, Luxembourg, Malta, Mexico, Netherlands,
Norway, Singapore, Slovenia, Sweden, Switzerland, United Kingdom, USA, and Uruguay (updated from Le
Quéré et al., 2019). Altogether, these 24 countries emitted 2.4 GtC $yr^{-1}$ (8.8 GtCO$_2$ $yr^{-1}$) on average over the last
decade, about one quarter of world $CO_2$ fossil emissions. Consumption-based emissions also fell significantly
during the final decade for which estimates are available (2011-2020) in 15 of these countries: Belgium,
Denmark, Estonia, Finland, France, Germany, Hong Kong, Israel, Japan, Luxembourg, Mexico, Netherlands,
Singapore, Sweden, United Kingdom, and Uruguay. Figure 15 shows that the emission declines in the USA and
the EU27 are primarily driven by increased decarbonisation ($CO_2$ emissions per unit energy) in the last decade
compared to the previous, with smaller contributions in the EU27 from slightly weaker economic growth and
slightly larger declines in energy per GDP. These countries have stable or declining energy use and so
decarbonisation policies replace existing fossil fuel infrastructure (Le Quéré et al. 2019).
In contrast, fossil $CO_2$ emissions continue to grow in non-OECD countries, although the growth rate has slowed
from almost 6% $yr^{-1}$ during the 2000s to less than 2% $yr^{-1}$ in the last decade. Representing 47% of non-OECD
emissions in 2021, a large part of this slowdown is due to China, which has seen emissions growth decline from
nearly 10% $yr^{-1}$ in the 2000s to 1.5% $yr^{-1}$ in the last decade. Excluding China, non-OECD emissions grew at
3.3% $yr^{-1}$ in the 2000s compared to 1.6% $yr^{-1}$ in the last decade. Figure 15 shows that, compared to the previous
decade, China has had weaker economic growth in the last decade and a higher decarbonisation rate, with more
rapid declines in energy per GDP that are now back to levels seen during the 1990s. India and the rest of the
world have strong economic growth that is not offset by decarbonisation or declines in energy per GDP, driving
up fossil $CO_2$ emissions. Despite the high deployment of renewables in some countries (e.g., India), fossil
energy sources continue to grow to meet growing energy demand (Le Quéré et al. 2019).
Globally, fossil CO2 emissions growth is slowing, and this is due to the emergence of climate policy (Eskander
and Fankhauser 2020; Le Quere et al 2019) and technological change, which is leading to a shift from coal to
gas and growth in renewable energies, and reduced expansion of coal capacity. At the aggregated global level,
decarbonisation shows a strong and growing signal in the last decade, with smaller contributions from lower
economic growth and declines in energy per GDP. Despite the slowing growth in global fossil $CO_2$ emissions,
emissions are still growing, far from the reductions needed to meet the ambitious climate goals of the UNFCCC
Paris agreement.





We update the remaining carbon budget assessed by the IPCC AR6 (Canadell et al., 2021), accounting for the
2020 to 2022 estimated emissions from fossil fuel combustion ($E_{FOS}$) and land use changes ($E_{LUC}$). From
January 2023, the remaining carbon (50% likelihood) for limiting global warming to 1.5°C, 1.7°C and 2°C is
estimated to amount to 105, 200, and 335 GtC (380, 730, 1230 $GtCO_2$). These numbers include an uncertainty
based on model spread (as in IPCC AR6), which is reflected through the percent likelihood of exceeding the
given temperature threshold. These remaining amounts correspond respectively to about 9, 18 and 30 years from
the beginning of 2023, at the 2022 level of total $CO_2$ emissions. Reaching net zero $CO_2$ emissions by 2050
entails cutting total anthropogenic $CO_2$ emissions by about 0.4 GtC (1.4 $GtCO_2$) each year on average,
comparable to the decrease observed in 2020 during the COVID-19 pandemic.

**5    Discussion**
Each year when the global carbon budget is published, each flux component is updated for all previous years to
consider corrections that are the result of further scrutiny and verification of the underlying data in the primary
input data sets. Annual estimates may be updated with improvements in data quality and timeliness (e.g., to
eliminate the need for extrapolation of forcing data such as land-use). Of all terms in the global budget, only the
fossil $CO_2$ emissions and the growth rate in atmospheric $CO_2$ concentration are based primarily on empirical
inputs supporting annual estimates in this carbon budget. The carbon budget imbalance, yet an imperfect
measure, provides a strong indication of the limitations in observations in understanding and representing
processes in models, and/or in the integration of the carbon budget components.
The persistent unexplained variability in the carbon budget imbalance limits our ability to verify reported
emissions (Peters et al., 2017) and suggests we do not yet have a complete understanding of the underlying
carbon cycle dynamics on annual to decadal timescales. Resolving most of this unexplained variability should
be possible through different and complementary approaches. First, as intended with our annual updates, the
imbalance as an error term is reduced by improvements of individual components of the global carbon budget
that follow from improving the underlying data and statistics and by improving the models through the
resolution of some of the key uncertainties detailed in Table 10. Second, additional clues to the origin and
processes responsible for the variability in the budget imbalance could be obtained through a closer scrutiny of
carbon variability in light of other Earth system data (e.g., heat balance, water balance), and the use of a wider
range of biogeochemical observations to better understand the land-ocean partitioning of the carbon imbalance
(e.g. oxygen, carbon isotopes). Finally, additional information could also be obtained through higher resolution
and process knowledge at the regional level, and through the introduction of inferred fluxes such as those based
on satellite $CO_2$ retrievals. The limit of the resolution of the carbon budget imbalance is yet unclear, but most
certainly not yet reached given the possibilities for improvements that lie ahead.
Estimates of global fossil $CO_2$ emissions from different datasets are in relatively good agreement when the
different system boundaries of these datasets are considered (Andrew, 2020a). But while estimates of $E_{FOS}$ are
derived from reported activity data requiring much fewer complex transformations than some other components
of the budget, uncertainties remain, and one reason for the apparently low variation between datasets is
precisely the reliance on the same underlying reported energy data. The budget excludes some sources of fossil



$CO_2$ emissions, which available evidence suggests are relatively small (<1%). We have added emissions from
lime production in China and the US, but these are still absent in most other non-Annex I countries, and before
1990 in other Annex I countries.
Estimates of $E_{LUC}$ suffer from a range of intertwined issues, including the poor quality of historical land-cover
and land-use change maps, the rudimentary representation of management processes in most models, and the
confusion in methodologies and boundary conditions used across methods (e.g., Arneth et al., 2017; Pongratz et
al., 2014, see also Section 2.7.4 on the loss of sink capacity; Bastos et al., 2021). Uncertainties in current and
historical carbon stocks in soils and vegetation also add uncertainty in the $E_{LUC}$ estimates. Unless a major effort
to resolve these issues is made, little progress is expected in the resolution of $E_{LUC}$. This is particularly
concerning given the growing importance of $E_{LUC}$ for climate mitigation strategies, and the large issues in the
quantification of the cumulative emissions over the historical period that arise from large uncertainties in $E_{LUC}$.
By adding the DGVMs estimates of $CO_2$ fluxes due to environmental change from countries' managed forest
areas (part of $S_{LAND}$ in this budget) to the budget $E_{LUC}$ estimate, we successfully reconciled the large gap
between our $E_{LUC}$ estimate and the land use flux from NGHGIs using the approach described in Grassi et al.
(2021) for future scenario and in Grassi et al. (2022b) using data from the Global Carbon Budget 2021. The
updated data presented here can be used as potential adjustment in the policy context, e.g., to help assessing the
collective countries' progress towards the goal of the Paris Agreement and avoiding double-accounting for the
sink in managed forests. In the absence of this adjustment, collective progress would hence appear better than it
is (Grassi et al. 2021). The need of such adjustment whenever a comparison between LULUCF fluxes reported
by countries and the global emission estimates of the IPCC is attempted is recommended also in the recent
UNFCCC Synthesis report for the first Global Stocktake (UNFCCC, 2022). However, this adjustment should be
seen as a short-term and pragmatic fix based on existing data, rather than a definitive solution to bridge the
differences between global models and national inventories. Additional steps are needed to understand and
reconcile the remaining differences, some of which are relevant at the country level (Grassi, et al. 2022b,
Schwingshackl, et al., subm.).
The comparison of GOBMs, data products and inversions highlights substantial discrepancy in the Southern
Ocean (Figure 13, Hauck et al., 2020). A large part of the uncertainty in the mean fluxes stems from the regional
distribution of the river flux adjustment term. The current distribution (Aumont et al., 2001) is based on one
model study yielding the largest riverine outgassing flux south of 20°S, whereas a recent study, also based on
one model, simulates the largest share of the outgassing to occur in the tropics (Lacroix et al., 2020). The long-
standing sparse data coverage of $fCO_2$ observations in the Southern compared to the Northern Hemisphere (e.g.,
Takahashi et al., 2009) continues to exist (Bakker et al., 2016, 2022, Figure B1) and to lead to substantially
higher uncertainty in the $S_{OCEAN}$ estimate for the Southern Hemisphere (Watson et al., 2020, Gloege et al.,
2021). This discrepancy, which also hampers model improvement, points to the need for increased high-quality
$fCO_2$ observations especially in the Southern Ocean. At the same time, model uncertainty is illustrated by the
large spread of individual GOBM estimates (indicated by shading in Figure 13) and highlights the need for
model improvement. The diverging trends in $S_{OCEAN}$ from different methods is a matter of concern, which is
unresolved. The assessment of the net land-atmosphere exchange from DGVMs and atmospheric inversions also
shows substantial discrepancy, particularly for the estimate of the total land flux over the northern extra-tropic.



This discrepancy highlights the difficulty to quantify complex processes ($CO_2$ fertilisation, nitrogen deposition
and fertilisers, climate change and variability, land management, etc.) that collectively determine the net land
$CO_2$ flux. Resolving the differences in the Northern Hemisphere land sink will require the consideration and
inclusion of larger volumes of observations.
We provide metrics for the evaluation of the ocean and land models and the atmospheric inversions (Figs. B2 to
B4). These metrics expand the use of observations in the global carbon budget, helping 1) to support
improvements in the ocean and land carbon models that produce the sink estimates, and 2) to constrain the
representation of key underlying processes in the models and to allocate the regional partitioning of the $CO_2$
fluxes. However, GOBMs skills have changed little since the introduction of the ocean model evaluation. The
additional simulation allows for direct comparison with interior ocean anthropogenic carbon estimates and
suggests that the models underestimate anthropogenic carbon uptake and storage. This is an initial step towards
the introduction of a broader range of observations that we hope will support continued improvements in the
annual estimates of the global carbon budget.
We assessed before that a sustained decrease of –1% in global emissions could be detected at the 66%
likelihood level after a decade only (Peters et al., 2017). Similarly, a change in behaviour of the land and/or
ocean carbon sink would take as long to detect, and much longer if it emerges more slowly. To continue
reducing the carbon imbalance on annual to decadal time scales, regionalising the carbon budget, and integrating
multiple variables are powerful ways to shorten the detection limit and ensure the research community can
rapidly identify issues of concern in the evolution of the global carbon cycle under the current rapid and
unprecedented changing environmental conditions.

**6      Conclusions**
The estimation of global $CO_2$ emissions and sinks is a major effort by the carbon cycle research community that
requires a careful compilation and synthesis of measurements, statistical estimates, and model results. The
delivery of an annual carbon budget serves two purposes. First, there is a large demand for up-to-date
information on the state of the anthropogenic perturbation of the climate system and its underpinning causes. A
broad stakeholder community relies on the data sets associated with the annual carbon budget including
scientists, policy makers, businesses, journalists, and non-governmental organisations engaged in adapting to
and mitigating human-driven climate change. Second, over the last decades we have seen unprecedented
changes in the human and biophysical environments (e.g., changes in the growth of fossil fuel emissions, impact
of COVID-19 pandemic, Earth's warming, and strength of the carbon sinks), which call for frequent
assessments of the state of the planet, a better quantification of the causes of changes in the contemporary global
carbon cycle, and an improved capacity to anticipate its evolution in the future. Building this scientific
understanding to meet the extraordinary climate mitigation challenge requires frequent, robust, transparent, and
traceable data sets and methods that can be scrutinised and replicated. This paper via 'living data' helps to keep
track of new budget updates.



## 7 Data availability

The data presented here are made available in the belief that their wide dissemination will lead to greater understanding and new scientific insights of how the carbon cycle works, how humans are altering it, and how we can mitigate the resulting human-driven climate change. Full contact details and information on how to cite the data shown here are given at the top of each page in the accompanying database and summarised in Table 2.

The accompanying database includes two Excel files organised in the following spreadsheets:

File Global_Carbon_Budget_2022v0.1.xlsx includes the following:

1. Summary
2. The global carbon budget (1959-2021);
3. The historical global carbon budget (1750-2021);
4. Global $CO_2$ emissions from fossil fuels and cement production by fuel type, and the per-capita emissions (1850-2021);
5. $CO_2$ emissions from land-use change from the individual methods and models (1959-2021);
6. Ocean $CO_2$ sink from the individual ocean models and $fCO_2$-based products (1959-2021);
7. Terrestrial $CO_2$ sink from the individual DGVMs (1959-2021);
8. Cement carbonation CO2 sink (1959-2021).

File National_Carbon_Emissions_2022v0.1.xlsx includes the following:

1. Summary
2. Territorial country $CO_2$ emissions from fossil $CO_2$ emissions (1850-2021);
3. Consumption country $CO_2$ emissions from fossil $CO_2$ emissions and emissions transfer from the international trade of goods and services (1990-2020) using CDIAC/UNFCCC data as reference;
4. Emissions transfers (Consumption minus territorial emissions; 1990-2020);
5. Country definitions.

Both spreadsheets are published by the Integrated Carbon Observation System (ICOS) Carbon Portal and are available at https://doi.org/10.18160/GCP-2022 (Friedlingstein et al., 2022b). National emissions data are also available from the Global Carbon Atlas (http://www.globalcarbonatlas.org/, last access: 25 September 2022) and from Our World in Data (https://ourworldindata.org/co2-emissions, last access: 25 September 2022).

## 8 Author contributions

PF, MOS, MWJ, RMA, LGr, JH, CLQ, ITL, AO, GPP, WP, JP, ClS, and SS designed the study, conducted the analysis, and wrote the paper with input from JGC, PC and RBJ. RMA, GPP and JIK produced the fossil fuel emissions and their uncertainties and analysed the emissions data. MH and GM provided fossil fuel emission data. JP, TGa, ClS and RAH provided the bookkeeping land-use change emissions with synthesis by JP and



ClS. JH, LB, ÖG, NG, TI, KL, NMa, LR, JS, RS, HiT, and ReW provided an update of the global ocean

biogeochemical models, MG, LGl, LGr, YI, AJ, ChR, JDS, and JZ provided an update of the ocean $fCO_2$ data

products, with synthesis on both streams by JH, LGr and NMa. SRA, NRB, MB, HCB, MC, WE, RAF, TGk,

KK, NL, NMe, NMM, DRM, SN, TO, DP, KP, ChR, IS, TS, AJS, CoS, ST, TT, BT, RiW, CW, AW provided

ocean $fCO_2$ measurements for the year 2021, with synthesis by AO and KO. AA, VKA, SF, AKJ, EK, DK, JK,

MJM, MOS, BP, QS, HaT, APW, WY, XY, and SZ provided an update of the Dynamic Global Vegetation

Models, with synthesis by SS and MOS. WP, ITL, FC, JL, YN, PIP, ChR, XT, and BZ provided an updated

atmospheric inversion, WP, FC, and ITL developed the protocol and produced the evaluation. RMA provided

predictions of the 2022 emissions and atmospheric $CO_2$ growth rate. PL provided the predictions of the 2022

ocean and land sinks. LPC, GCH, KKG, TMR and GRvdW provided forcing data for land-use change. RA, GG,

FT, and CY provided data for the land-use change NGHGI mapping. PPT provided key atmospheric $CO_2$ data.

MWJ produced the model atmospheric $CO_2$ forcing and the atmospheric $CO_2$ growth rate. MOS and NB

produced the aerosol diffuse radiative forcing for the DGVMs. IH provided the climate forcing data for the

DGVMs. ER provided the evaluation of the DGVMs. MWJ provided the emissions prior for use in the inversion

systems. ZL provided seasonal emissions data for most recent years for the emission prior. MWJ and MOS

developed the new data management pipeline which automates many aspects of the data collation, analysis,

plotting and synthesis. PF, MOS and MMJ coordinated the effort, revised all figures, tables, text and/or numbers

to ensure the update was clear from the 2021 edition and in line with the globalcarbonatlas.org.

**Competing interests.** The authors declare that they have no conflict of interest.

## 9      Acknowledgements

We thank all people and institutions who provided the data used in this global carbon budget 2022 and the Global

Carbon Project members for their input throughout the development of this publication. We thank Nigel Hawtin

for producing Figure 2 and Figure 14. We thank Thomas Hawes for technical support with the data management

pipeline. We thank Ed Dlugokencky for providing atmospheric $CO_2$ measurements. We thank Ian G. C. Ashton,

Fatemeh Cheginig, Trang T. Chau, Sam Ditkovsky, Christian Ethé, Amanda R. Fay, Lonneke Goddijn-Murphy,

T. Holding, Fabrice Lacroix, Enhui Liao, Galen A. McKinley, Shijie Shu, Richard Sims, Jade Skye, Andrew J.

Watson, David Willis, and David K. Woolf for their involvement in the development, use and analysis of the

models and data-products used here. Daniel Kennedy thanks all the scientists, software engineers, and





administrators who contributed to the development of CESM2. We thank Joe Salisbury, Doug Vandemark,
Christopher W. Hunt, and Peter Landschützer who contributed to the provision of surface ocean $CO_2$ observations
for the year 2021 (see Table A5). We also thank Benjamin Pfeil, Rocío Castaño-Primo, and Stephen D. Jones of
the Ocean Thematic Centre of the EU Integrated Carbon Observation System (ICOS) Research Infrastructure,
Eugene Burger of NOAA's Pacific Marine Environmental Laboratory and Alex Kozyr of NOAA's National
Centers for Environmental Information, for their contribution to surface ocean $CO_2$ data and metadata
management. This is PMEL contribution 5434. We thank the scientists, institutions, and funding agencies
responsible for the collection and quality control of the data in SOCAT as well as the International Ocean Carbon
Coordination Project (IOCCP), the Surface Ocean Lower Atmosphere Study (SOLAS) and the Integrated Marine
Biosphere Research (IMBeR) program for their support. We thank data providers ObsPack GLOBALVIEWplus
v7.0 and NRT v7.2 for atmospheric $CO_2$ observations. We thank the individuals and institutions that provided the
databases used for the model evaluations used here. We thank Fortunat Joos, Samar Khatiwala and Timothy
DeVries for providing historical data. Matthew J. McGrath thanks the whole ORCHIDEE group. Ian Harris thanks
the Japan Meteorological Agency (JMA) for producing the Japanese 55-year Reanalysis (JRA-55). Anthony P.
Walker thanks ORNL which is managed by UT-Battelle, LLC, for the DOE under contract DE-AC05-1008
00OR22725. Yosuke Niwa thanks CSIRO, EC, EMPA, FMI, IPEN, JMA, LSCE, NCAR, NIES, NILU, NIWA,
NOAA, SIO, and TU/NIPR for providing data for NISMON-CO2. Xiangjun Tian thanks Zhe Jin, Yilong Wang,
Tao Wang and Shilong Piao for their contributions to the GONGGA inversion system. Bo Zheng thanks the
comments and suggestions from Philippe Ciais and Frédéric Chevallier. Frédéric Chevallier thanks Marine
Remaud who maintained the atmospheric transport model for the CAMS inversion. Paul I. Palmer thanks Liang
Feng and acknowledges ongoing support from the National Centre for Earth Observation. Junjie Liu thanks the
Jet Propulsion Laboratory, California Institute of Technology. Wiley Evans thanks the Tula Foundation for
funding support. Australian ocean $CO_2$ data were sourced from Australia's Integrated Marine Observing System
(IMOS); IMOS is enabled by the National Collaborative Research Infrastructure Strategy (NCRIS). Margot
Cronin thanks Anthony English, Clynt Gregory and Gordon Furey (P&O Maritime Services) for their support.
Nathalie Lefèvre thanks the crew of the Cap San Lorenzo and the US IMAGO of IRD Brest for technical support.
Henry C. Bittig is grateful for the skillful technical support of M. Glockzin and B. Sadkowiak. Meike Becker and
Are Olsen thank Sparebanken Vest/Agenda Vestlandet for their support for the observations on the Statsraad
Lehmkuhl. Thanos Gkritzalis thanks the personnel and crew of Simon Stevin. Matthew W. Jones thanks Anthony
J. De-Gol for his technical and conceptual assistance with the development of GCP-GridFED. FAOSTAT is



funded by FAO member states through their contributions to the FAO Regular Programme, data contributions by
national experts are greatly acknowledged. The views expressed in this paper are the authors' only and do not
necessarily reflect those of FAO. Finally, we thank all funders who have supported the individual and joint
contributions to this work (see Table A9), as well as the reviewers of this manuscript and previous versions, and
the many researchers who have provided feedback.

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



**Tables**

| Table 1. Factors used to convert carbon in various units (by convention, Unit 1 = Unit 2 × conversion). | | | |
|---|---|---|---|
| Unit 1 | Unit 2 | Conversion | Source |
| GtC (gigatonnes of carbon) | ppm (parts per million) (a) | 2.124 (b) | Ballantyne et al. (2012) |
| GtC (gigatonnes of carbon) | PgC (petagrams of carbon) | 1 | SI unit conversion |
| GtCO2 (gigatonnes of carbon dioxide) | GtC (gigatonnes of carbon) | 3.664 | 44.01/12.011 in mass equivalent |
| GtC (gigatonnes of carbon) | MtC (megatonnes of carbon) | 1000 | SI unit conversion |
| (a) Measurements of atmospheric CO2 concentration have units of dry-air mole fraction. 'ppm' is an abbreviation for micromole/mol, dry air. | | | |
| (b) The use of a factor of 2.124 assumes that all the atmosphere is well mixed within one year. In reality, only the troposphere is well mixed and the growth rate of CO2 concentration in the less well-mixed stratosphere is not measured by sites from the NOAA network. Using a factor of 2.124 makes the approximation that the growth rate of CO2 concentration in the stratosphere equals that of the troposphere on a yearly basis. | | | |







| Table 2. How to cite the individual components of the global carbon budget presented here. | |
|---|---|
| **Component** | **Primary reference** |
| Global fossil CO2 emissions (EFOS), total and by fuel type | Updated from Andrew and Peters (2021) |
| National territorial fossil CO2 emissions (EFOS) | Gilfillan and Marland (2022), UNFCCC (2022) |
| National consumption-based fossil CO2 emissions (EFOS) by country (consumption) | Peters et al. (2011b) updated as described in this paper |
| Net land-use change flux (ELUC) | This paper (see Table 4 for individual model references). |
| Growth rate in atmospheric CO2 concentration (GATM) | Dlugokencky and Tans (2022) |
| Ocean and land CO2 sinks (SOCEAN and SLAND) | This paper (see Table 4 for individual model and data products references). |






**Table 3.** Main methodological changes in the global carbon budget since 2018. Methodological changes introduced in one year are kept for the following years unless noted. Empty cells mean there were no methodological changes introduced that year. Table A7 lists methodological changes from the first global carbon budget publication up to 2017.

| Publication year | Fossil fuel emissions | | LUC emissions | Reservoirs | | | Uncertainty & other changes |
|---|---|---|---|---|---|---|---|
| | Global | Country (territorial) | | Atmosphere | Ocean | Land | |
| 2018<br><br>Le Quéré et al. (2018b) GCB2018 | Revision in cement emissions; Projection includes EU-specific data | Aggregation of overseas territories into governing nations for total of 213 countries a | Average of two bookkeeping models; use of 16 DGVMs | Use of four atmospheric inversions | Based on seven models | Based on 16 models; revised atmospheric forcing from CRUNCEP to CRUJRA | Introduction of metrics for evaluation of individual models using observations |
| 2019<br><br>Friedlingstein et al. (2019) GCB2019 | Global emissions calculated as sum of all countries plus bunkers, rather than taken directly from CDIAC. | | Average of two bookkeeping models; use of 15 DGVMs | Use of three atmospheric inversions | Based on nine models | Based on 16 models | |
| 2020<br><br>Friedlingstein et al. (2020) GCB2020 | Cement carbonation now included in the EFOS estimate, reducing EFOS by about 0.2GtC yr-1 for the last decade | India's emissions from Andrew (2020: India); Corrections to Netherland Antilles and Aruba and Soviet emissions before 1950 as per Andrew (2020: CO2); China's coal emissions in 2019 derived from official statistics, emissions now shown for EU27 instead of EU28.Projection for 2020 based on assessment of four | Average of three bookkeeping models; use of 17 DGVMs. Estimate of gross land use sources and sinks provided | Use of six atmospheric inversions | Based on nine models. River flux revised and partitioned NH, Tropics, SH | Based on 17 models | |



| | | approaches. | | | | | |
|---|---|---|---|---|---|---|---|
| 2021<br><br>Friedlingstein et al. (2022a) GCB2021 | Projections are no longer an assessment of four approaches. | Official data included for a number of additional countries, new estimates for South Korea, added emissions from lime production in China. | ELUC estimate compared to the estimates adopted in national GHG inventories (NGHGI) | | Average of means of eight models and means of seven data-products. Current year prediction of SOCEAN using a feed-forward neural network method | Current year prediction of SLAND using a feed-forward neural network method | |
| 2022<br><br>This study | | | ELUC provided at country level. Decomposition into fluxes from deforestation, organic soils, uptake in forests, and other transitions. Change in the methodology to derive LUC maps for Brazil to capture recent upturn in deforestation | Use of nine atmospheric inversions | Average of means of ten models and means of seven data-products | Based on 16 models. Change in the methodology to derive LUC maps for Brazil to capture recent upturn in deforestation | |

**Table 4. References for the process models, bookkeeping models, ocean data products, and atmospheric inversions. All models and products are updated with new data to the end of year 2021, and the atmospheric forcing for the DGVMs has been updated as described in Section C.2.2.**

| Model/data name | Reference | Change from Global Carbon Budget 2021 (Friedlingstein et al., 2022a) |
|---|---|---|
| *Bookkeeping models for land-use change emissions* | | |
| BLUE | Hansis et al. (2015) | No change to model, but simulations performed with updated LUH2 forcing. Update in added peat drainage emissions (based on three spatially explicit datasets). |
| updated H&N2017 | Houghton and Nassikas (2017) | Minor bug fix in the fuel harvest estimates, that was causing an overestimation of fuel sink. Update in added peat drainage emissions (based on three spatially explicit datasets). |
| OSCAR | Gasser et al. (2020) | No change to model, but land use forcing changed to LUH2-GCB2022 and FRA2020 (as used by H&N and extrapolated to 2021), both prescribed at higher spatial resolution (210 instead of 96 regions/countries). Constraining based on last year's budget data for SLAND over 1960-2021. Update in added peat drainage emissions (based on three spatially explicit datasets). |
| *Dynamic global vegetation models* | | |
| CABLE-POP | Haverd et al. (2018) | changes in parameterisation. Diffuse fraction of incoming radiation read in as forcing. |
| CLASSIC | Melton et al. (2020) (a) | Minor bug fixes. |
| CLM5.0 | Lawrence et al. (2019) | No change. |
| DLEM | Tian et al. (2015) (b) | No change. |
| IBIS | Yuan et al. (2014) (c) | No change. |
| ISAM | Meiyappan et al. (2015) (d) | No change. |
| JSBACH | Reick et al. (2021) (f) | No change. |
| JULES-ES | Wiltshire et al. (2021) (g) | Minor bug fixes. (Using JULES v6.3, suite u-co002) |
| LPJ-GUESS | Smith et al. (2014) (h) | No change. |
| LPJ | Poulter et al. (2011) (i) | No change. |
| LPX-Bern | Lienert and Joos (2018) | Following the results of Joos et al. (2018), we use modified parameter values which yield a more reasonable (lower) BNF, termed LPX v1.5. This parameter version has increased N immobilization and a stronger N limitation, than the previous version. The N2O Emissions were adjusted accordingly. The parameters |



| | | were obtained by running an ensemble simulation and imposing various observational constraints and subsequently adjusting N immobilization.<br>For the methodology see Lienert et. al. (2018). |
|---|---|---|
| OCN | Zaehle and Friend (2010) (j) | No change (uses r294). |
| ORCHIDEEv3 | Vuichard et al. (2019) (k) | No change (ORCHIDEE - V3; revision 7267) |
| SDGVM | Walker et al. (2017) (l) | No change. |
| VISIT | Kato et al. (2013) (m) | No change. |
| YIBs | Yue and Unger (2015) | No change. |
| *Global ocean biogeochemistry models* | | |
| NEMO-PlankTOM12 | Wright et al. (2021) | Minor bug fixes |
| MICOM-HAMOCC (NorESM-OCv1.2) | Schwinger et al. (2016) | No change. |
| MPIOM-HAMOCC6 | Lacroix et al. (2021) | No change. |
| NEMO3.6-PISCESv2-gas (CNRM) | Berthet et al. (2019) (n) | No change. |
| FESOM-2.1-REcoM2 | Hauck et al. (2020) (o) | Extended spin-up, minor bug fixes |
| MOM6-COBALT (Princeton) | Liao et al. (2020) | No change |
| CESM-ETHZ | Doney et al. (2009) | Changed salinity restoring in the surface ocean from 700 days to 300 days, except for the Southern Ocean south of 45S, where the restoring timescale was set to 60 days. |
| NEMO-PISCES (IPSL) | Aumont et al. (2015) | No change. |
| MRI-ESM2-1 | Nakano et al. (2011), Urakawa et al. (2020) | New this year. |
| CESM2 | Long et al. (2021) (p) | New this year. |
| *ocean data products* | | |
| MPI-SOMFFN | Landschützer et al. (2016) | update to SOCATv2022 measurements and timeperiod 1982-2021; The estimate now covers the full ocean domain as well as the Arctic Ocean extension described in: Landschützer, P., Laruelle, G. G., Roobaert, A., and Regnier, P.: A uniform pCO2 |





| | | |
|---|---|---|
| | | climatology combining open and coastal oceans, Earth Syst. Sci. Data, 12, 2537–2553, https://doi.org/10.5194/essd-12-2537-2020, 2020. |
| Jena-MLS | Rödenbeck et al. (2022) | update to SOCATv2022 measurements, time period extended to 1957-2021 |
| CMEMS-LSCE-FFNNv2 | Chau et al. (2022) | Update to SOCATv2022 measurements and time period 1985-2021. The CMEMS-LSCE-FFNNv2 product now covers both the open ocean and coastal regions. |
| LDEO-HPD | Gloege et al. (2022) (q) | New this year |
| UOEx-Watson | Watson et al. (2020) | Updated to SOCAT v2022 and OISSTv2.1, as recalculated by Holding et al. |
| NIES-NN | Zeng et al. (2014) | Updated to SOCAT v2022. Small changes in method (gas-exchange coefficient a= 0.271; trend calculation 1990-2020, predictors include lon and lat) |
| JMA-MLR | Iida et al. (2021) | Updated to SOCATv2022 SST fields (MGDSST) updated |
| OS-ETHZ-GRaCER | Gregor and Gruber (2021) | No change |
| *Atmospheric inversions* | | |
| CAMS | Chevallier et al. (2005) (r) | Updated to WMOX2019 scale. Extension to year 2021, revision of the station list, update of the prior fluxes |
| CarbonTracker Europe (CTE) | van der Laan-Luijkx et al. (2017) | Updated to WMOX2019 scale. Biosphere prior fluxes from the SiB4 model instead of SiBCASA model. Extension to 2021. |
| Jena CarboScope | Rödenbeck et al. (2018) (s) | Updated to WMOX2019 scale. Extension to 2021. |
| UoE in-situ | Feng et al., (2016) (t) | Updated to WMOX2019 scale. Updated station list, and refined land-ocean map. Extension to 2021. |
| NISMON-CO2 | Niwa et al., (2022) (u) | Updated to WMOX2019 scale. Positive definite flux parameters and updated station list. Extension to 2021. |
| CMS-Flux | Liu et al., (2021) | Updated to WMOX2019 scale. Extension to 2021. |
| GONGGA | Jin et al. (2022 in review) (v) | New this year. |
| THU | Kong et al. (2022) | New this year. |
| CAMS-Satellite | Chevallier et al. (2005) (r) | New this year. |
| (a) see also Asaadi et al. (2018). | | |
| (b) see also Tian et al. (2011) | | |
| (c) the dynamic carbon allocation scheme was presented by Xia et al. (2015) | | |
| (d) see also Jain et al. (2013). Soil biogeochemistry is updated based on Shu et al. (2020) | | |





| |
|---|
| (e) see also Decharme et al. (2019) and Seferian et al. (2019) |
| (f) see also Mauritsen et al. (2019) |
| (g) see also Sellar et al. (2019) and Burton et al., (2019). JULES-ES is the Earth System configuration of the Joint UK Land Environment Simulator as used in the UK Earth System Model (UKESM). |
| (h) to account for the differences between the derivation of shortwave radiation from CRU cloudiness and DSWRF from CRUJRA, the photosynthesis scaling parameter $\alpha a$ was modified (-15%) to yield similar results. |
| (i) compared to published version, decreased LPJ wood harvest efficiency so that 50 % of biomass was removed off-site compared to 85 % used in the 2012 budget. Residue management of managed grasslands increased so that 100 % of harvested grass enters the litter pool. |
| (j) see also Zaehle et al. (2011). |
| (k) see also Zaehle and Friend (2010) and Krinner et al. (2005) |
| (l) see also Woodward and Lomas (2004) |
| (m) see also Ito and Inatomi (2012). |
| (n) see also Séférian et al. (2019) |
| (o) see also Schourup-Kristensen et al (2014) |
| (p) see also Yeager et al. (2022) |
| (q) see also Bennington et al. (2022) |
| (r) see also Remaud (2018) |
| (s) see also Rödenbeck et al. (2003) |
| (t) see also Feng et al. (2009) and Palmer et al. (2019) |
| (u) see also Niwa et al. (2020) |
| (v) see also Tian et al. (2014) |



**Table 5. Comparison of results from the bookkeeping method and budget residuals with results from the DGVMs and inverse estimates for different periods, the last decade, and the last year available. All values are in GtCyr–1. See Fig. 7 for explanation of the bookkeeping component fluxes. The DGVM uncertainties represent ±1σ of the decadal or annual (for 2021) estimates from the individual DGVMs: for the inverse systems the range of available results is given. All values are rounded to the nearest 0.1 GtC and therefore columns do not necessarily add to zero.**

### Mean (GtC/yr)

|  |  | 1960s | 1970s | 1980s | 1990s | 2000s | 2012-2021 | 2021 |
|---|---|---|---|---|---|---|---|---|
| **Land-use change emissions (ELUC)** | Bookkeeping (BK) Net flux (1a) | 1.5±0.7 | 1.2±0.7 | 1.3±0.7 | 1.5±0.7 | 1.4±0.7 | 1.2±0.7 | 1.1±0.7 |
|  | BK - deforestation | 1.6±0.4 | 1.5±0.4 | 1.6±0.4 | 1.8±0.3 | 1.9±0.4 | 1.8±0.4 | 1.8±0.4 |
|  | BK - organic soils | 0.1±0.1 | 0.1±0.1 | 0.2±0.1 | 0.2±0.1 | 0.2±0.1 | 0.2±0.1 | 0.2±0.1 |
|  | BK - re-/afforestation and forestry | -0.6±0.1 | -0.6±0.1 | -0.6±0.2 | -0.7±0.1 | -0.8±0.2 | -0.9±0.3 | -1.0±0.3 |
|  | BK - other transitions | 0.4±0.0 | 0.2±0.1 | 0.2±0.1 | 0.1±0.1 | 0.1±0.1 | 0.1±0.1 | 0.1±0.1 |
|  | DGVMs-net flux (1b) | 1.4±0.5 | 1.3±0.5 | 1.5±0.5 | 1.5±0.6 | 1.6±0.6 | 1.6±0.5 | 1.6±0.5 |
| **Terrestrial sink (SLAND)** | Residual sink from global budget (EFOS+ELUC(1a)-GATM-SOCEAN) (2a) | 1.7±0.8 | 1.8±0.8 | 1.6±0.9 | 2.6±0.9 | 2.8±0.9 | 2.8±0.9 | 2.8±1 |
|  | DGVMs (2b) | 1.2±0.4 | 2.2±0.5 | 1.9±0.7 | 2.5±0.4 | 2.7±0.5 | 3.1±0.6 | 3.5±0.9 |
| **Total land fluxes (SLAND-ELUC)** | GCB2022 Budget (2b-1a) | -0.2±0.8 | 1±0.9 | 0.5±1 | 1±0.8 | 1.4±0.9 | 1.9±0.9 | 2.4±1.1 |
|  | Budget constraint (2a-1a) | 0.2±0.4 | 0.6±0.5 | 0.3±0.5 | 1.1±0.5 | 1.5±0.6 | 1.5±0.6 | 1.7±0.7 |
|  | DGVMs-net (2b-1b) | -0.1±0.4 | 0.9±0.5 | 0.4±0.5 | 0.9±0.4 | 1.2±0.3 | 1.5±0.5 | 1.9±0.7 |
|  | Inversions[*] | --- | --- | 0.3-0.6 (2) | 0.7-1.1 (3) | 1.2-1.6 (3) | 1.1-1.7 (7) | 1.5-2.1 (9) |

[*]Estimates are adjusted for the pre-industrial influence of river fluxes, for the cement carbonation sink, and adjusted to common EFOS (Sect. 2.6). The ranges given include varying numbers (in parentheses) of inversions in each decade (Table A4)





**Table 6. Decadal mean in the five components of the anthropogenic CO2 budget for different periods, and last year available. All values are in GtC yr-1, and uncertainties are reported as ±1σ. Fossil CO$_2$ emissions include cement carbonation. The table also shows the budget imbalance (B$_{IM}$), which provides a measure of the discrepancies among the nearly independent estimates. A positive imbalance means the emissions are overestimated and/or the sinks are too small. All values are rounded to the nearest 0.1 GtC and therefore columns do not necessarily add to zero.**

Mean (GtC/yr)

| | | 1960s | 1970s | 1980s | 1990s | 2000s | 2012-2021 | 2021 | 2022 (Projection) |
|---|---|---|---|---|---|---|---|---|---|
| **Total emissions (EFOS + ELUC)** | Fossil CO2 emissions (EFOS)[*] | 3±0.2 | 4.7±0.2 | 5.5±0.3 | 6.3±0.3 | 7.7±0.4 | 9.6±0.5 | 9.9±0.5 | 10.2±0.5 |
| | Land-use change emissions (ELUC) | 1.5±0.7 | 1.2±0.7 | 1.3±0.7 | 1.5±0.7 | 1.4±0.7 | 1.2±0.7 | 1.1±0.7 | 1±0.7 |
| | Total emissions | 4.5±0.7 | 5.9±0.7 | 6.8±0.8 | 7.8±0.8 | 9.1±0.8 | 10.8±0.8 | 10.9±0.9 | 11.1±0.9 |
| **Partitioning** | Growth rate in atmos CO2 (GATM) | 1.7±0.07 | 2.8±0.07 | 3.4±0.02 | 3.1±0.02 | 4±0.02 | 5.2±0.02 | 5.2±0.2 | 5.5±0.4 |
| | Ocean sink (SOCEAN) | 1.1±0.4 | 1.4±0.4 | 1.8±0.4 | 2.1±0.4 | 2.3±0.4 | 2.9±0.4 | 2.9±0.4 | 2.9±0.4 |
| | Terrestrial sink (SLAND) | 1.2±0.4 | 2.2±0.5 | 1.9±0.7 | 2.5±0.4 | 2.7±0.5 | 3.1±0.6 | 3.5±0.9 | 3.4±0.9 |
| **Budget Imbalance** | BIM=EFOS+ELUC-(GATM+SOCEAN+SLAND) | 0.4 | -0.4 | -0.3 | 0.1 | 0.1 | -0.3 | -0.6 | -0.6 |

[*]Fossil emissions excluding the cement carbonation sink amount to 3.1±0.2 GtC/yr, 4.7±0.2 GtC/yr, 5.5±0.3 GtC/yr, 6.4±0.3 GtC/yr, 7.9±0.4 GtC/yr, and 9.8±0.5 GtC/yr for the decades 1960s to 2010s respectively and to 10.1±0.5 GtC/yr for 2021.



**Table 7.** Comparison of the projection with realised fossil CO2 emissions (EFOS). The 'Actual' values are first the estimate available using actual data, and the 'Projected' values refers to estimates made before the end of the year for each publication. Projections based on a different method from that described here during 2008-2014 are available in Le Quéré et al., (2016). All values are adjusted for leap years.

| | World | | China | | USA | | EU28 / EU27 (i) | | India | | Rest of World | |
|---|---|---|---|---|---|---|---|---|---|---|---|---|
| | Projected | Actual | Projected | Actual | Projected | Actual | Projected | Actual | Projected | Actual | Projected | Actual |
| 2015 (a) | −0.6% (−1.6 to 0.5) | 0.06% | −3.9% (−4.6 to −1.1) | −0.7% | −1.5% (−5.5 to 0.3) | −2.5% | – | – | – | – | 1.2% (−0.2 to 2.6) | 1.2% |
| 2016 (b) | −0.2% (−1.0 to +1.8) | 0.20% | −0.5% (−3.8 to +1.3) | −0.3% | −1.7% (−4.0 to +0.6) | −2.1% | – | – | – | – | 1.0% (−0.4 to +2.5) | 1.3% |
| 2017 (c) | 2.0% (+0.8 to +3.0) | 1.6% | 3.5% (+0.7 to +5.4) | 1.5% | −0.4% (−2.7 to +1.0) | −0.5% | – | – | 2.00% (+0.2 to +3.8) | 3.9% | 1.6% (0.0 to +3.2) | 1.9% |
| 2018 (d) | 2.7% (+1.8 to +3.7) | 2.1% | 4.7% (+2.0 to +7.4) | 2.3% | 2.5% (+0.5 to +4.5) | 2.8% | -0.7% (-2.6 to +1.3) | -2.1% | 6.3% (+4.3 to +8.3) | 8.0% | 1.8% (+0.5 to +3.0) | 1.7% |
| 2019 (e) | 0.5% (-0.3 to +1.4) | 0.1% | 2.6% (+0.7 to +4.4) | 2.2% | -2.4% (-4.7 to -0.1) | -2.6% | -1.7% (-5.1% to +1.8%) | -4.3% | 1.8% (-0.7 to +3.7) | 1.0% | 0.5% (-0.8 to +1.8) | 0.5% |
| 2020 (f) | -6.7% | -5.4% | -1.7% | 1.4% | -12.2% | -10.6% | -11.3% (EU27) | -10.9% | -9.1% | -7.3% | -7.4% | -7.0% |
| 2021 (g) | 4.8% (4.2% to 5.4%) | 5.1% | 4.3% (3.0% to 5.4%) | 3.5% | 6.8% (6.6% to 7.0%) | 6.2% | 6.3% (4.3% to 8.3%) | 6.8% | 11.2% (10.7% to 11.7%) | 11.1% | 3.2% (2.0% to 4.3%) | 4.5% |
| 2022 (h) | 1.1% (0% to 1.7%) | | -1.5% (-3.0% to 0.1%) | | 1.6% (-0.9% to 4.1%) | | -1.0% (-2.9% to 1.0%) | | 5.6% (3.5% to 7.7%) | | 2.5% (0.1% to 2.3%) | |

(a) Jackson et al. (2016) and Le Quéré et al. (2015a). (b) Le Quéré et al. (2016). (c) Le Quéré et al. (2018a). (d) Le Quéré et al. (2018b). (e) Friedlingstein et al., (2019), (f) Friedlingstein et al., (2020), (g) Friedlingstein et al., (2022a), (h) This study

(i) EU28 until 2019, EU27 from 2020





**Table 8. Cumulative CO₂ for different time periods in gigatonnes of carbon (GtC). Fossil CO₂ emissions include cement carbonation. The budget imbalance (B$_{IM}$) provides a measure of the discrepancies among the nearly independent estimates. All values are rounded to the nearest 5 GtC and therefore columns do not necessarily add to zero. Uncertainties are reported as follows: E$_{FOS}$ is 5% of cumulative emissions; E$_{LUC}$ prior to 1959 is 1σ spread from the DGVMs, E$_{LUC}$ post-1959 is 0.7*number of years (where 0.7 GtC/yr is the uncertainty on the annual ELUC flux estimate); G$_{ATM}$ uncertainty is held constant at 5 GtC for all time periods; S$_{OCEAN}$ uncertainty is 20% of the cumulative sink (20% relates to the annual uncertainty of 0.4 GtC/yr, which is ~20% of the current ocean sink); and S$_{LAND}$ is the 1σ spread from the DGVMs estimates.**

| | | 1750-2021 | 1850-2014 | 1850-2021 | 1960-2021 | 1850-2022 |
|---|---|---|---|---|---|---|
| Emissions | Fossil CO2 emissions (EFOS) | 470±25 | 400±20 | 465±25 | 385±20 | 475±25 |
| | Land-use change emissions (ELUC) | 235±70 | 195±60 | 205±60 | 85±45 | 205±60 |
| | Total emissions | 700±75 | 595±60 | 670±65 | 470±50 | 680±65 |
| Partitioning | Growth rate in atmos CO2 (GATM) | 295±5 | 235±5 | 275±5 | 210±5 | 280±5 |
| | Ocean sink (SOCEAN) | 185±35 | 155±30 | 175±35 | 120±25 | 180±35 |
| | Terrestrial sink (SLAND) | 230±50 | 185±40 | 210±45 | 145±30 | 210±45 |
| Budget imbalance | BIM=EFOS+ELUC-(GATM+SOCEAN+SLAND) | -5 | 15 | 15 | -5 | 10 |

**Table 9: Mapping of global carbon cycle models' land flux definitions to the definition of the LULUCF net flux used in national Greenhouse Gas Inventories reported to UNFCCC. See Sec. C.2.3 and Tab. A8 for detail on methodology and comparison to other datasets.**

| | 2002-2011 | 2012-2021 |
|---|---|---|
| ELUC from bookkeeping estimates (from Table 5) | 1.4 | 1.2 |
| SLAND on non-intact forest from DGVMs | -1.7 | -1.8 |
| ELUC plus SLAND on non-intact forests | -0.3 | -0.6 |
| National Greenhouse Gas Inventories | -0.4 | -0.5 |



**Table 10.** Major known sources of uncertainties in each component of the Global Carbon Budget, defined as input data or processes that have a demonstrated effect of at least ±0.3 GtC yr-1.

| Source of uncertainty | Time scale (years) | Location | Status | Evidence |
|---|---|---|---|---|
| **Fossil CO2 emissions (EFOS; Section 2.1)** | | | | |
| energy statistics | annual to decadal | global, but mainly China & major developing countries | see Sect. 2.1 | (Korsbakken et al., 2016, Guan et al., 2012) |
| carbon content of coal | annual to decadal | global, but mainly China & major developing countries | see Sect. 2.1 | (Liu et al., 2015) |
| system boundary | annual to decadal | all countries | see Sect. 2.1 | (Andrew, 2020) |
| **Net land-use change flux (ELUC; section 2.2)** | | | | |
| land-cover and land-use change statistics | continuous | global; in particular tropics | see Sect. 2.4 | (Houghton et al., 2012, Gasser et al., 2020, Ganzenmüller et al., 2022, Yu et al. 2022) |
| sub-grid-scale transitions | annual to decadal | global | see Sect. 2.4, Table A1 | (Wilkenskjeld et al., 2014) |
| vegetation biomass | annual to decadal | global; in particular tropics | see Sect. 2.4 | (Houghton et al., 2012, Bastos et al., 2021) |
| forest degradation (fire, selective logging) | annual to decadal | tropics | see Sec. 3.2.2, Table A1 | (Aragão et al., 2018, Qin et al., 2020) |
| wood and crop harvest | annual to decadal | global; SE Asia | see Table A1 | (Arneth et al., 2017, Erb et al., 2018) |
| peat burning (a) | multi-decadal trend | global | see Table A1 | (van der Werf et al., 2010, 2017) |
| loss of additional sink capacity | multi-decadal trend | global | not included; see Appendix D4 | (Pongratz et al, 2014, Gasser et al, 2020; Obermeier et al., 2021) |
| **Atmospheric growth rate (GATM; section 2.3) no demonstrated uncertainties larger than ±0.3 GtC yr-1 (b)** | | | | |
| **Ocean sink (SOCEAN; section 2.4)** | | | | |
| sparsity in surface fCO2 observations | mean, decadal variability and trend | global, in particular southern hemisphere | see Sect 3.5.2 | (Gloege et al., 2021, Denvil-Sommer et al., 2021, Bushinsky et al., 2019) |
| riverine carbon outgassing and its anthropogenic perturbation | annual to decadal | global, in particular partitioning between Tropics and South | see Sect. 2.4 (anthropogenic perturbations not included) | (Aumont et al., 2001, Resplandy et al., 2018, Lacroix et al., 2020) |
| Models underestimate interior ocean | annual to decadal | global | see Sect 3.5.5 | (Friedlingstein et al., 2021, this study, see also Terhaar et al., 2022) |



| anthropogenic carbon storage | | | | |
|---|---|---|---|---|
| near-surface temperature and salinity gradients | mean on all time-scales | global | see Sect. 3.8.2 | (Watson et al., 2020, Dong et al., 2022) |
| **Land sink (SLAND; section 2.5)** | | | | |
| strength of CO2 fertilisation | multi-decadal trend | global | see Sect. 2.5 | (Wenzel et al., 2016; Walker et al., 2021) |
| response to variability in temperature and rainfall | annual to decadal | global; in particular tropics | see Sect. 2.5 | (Cox et al., 2013; Jung et al., 2017; Humphrey et al., 2018; 2021) |
| nutrient limitation and supply | annual to decadal | global | | (Zaehle et al., 2014) |
| carbon allocation and tissue turnover rates | annual to decadal | global | | (De Kauwe et al., 2014; O'Sullivan et al., 2022) |
| tree mortality | annual | global in particular tropics | see Sect. 2.5 | (Hubau et al., 2021; Brienen et al., 2020) |
| response to diffuse radiation | annual | global | see Sect. 2.5 | (Mercado et al., 2009; O'Sullivan et al., 2021) |

| (a) As result of interactions between land-use and climate |
|---|

| (b) The uncertainties in GATM have been estimated as ±0.2 GtC yr-1, although the conversion of the growth rate into a global annual flux assuming instantaneous mixing throughout the atmosphere introduces additional errors that have not yet been quantified. |
|---|

**Figures and Captions**

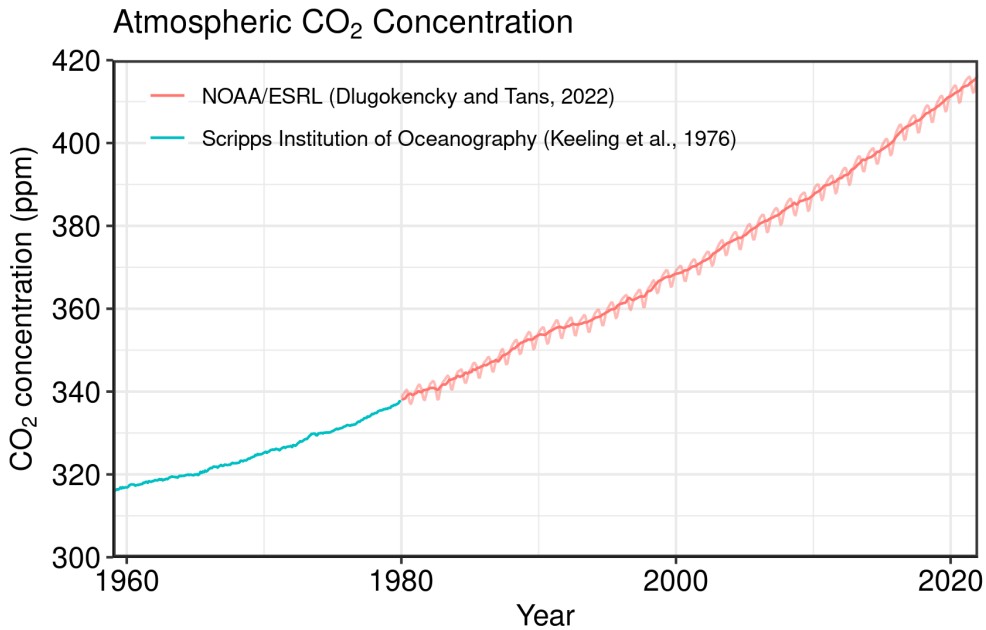

**Figure 1.** Surface average atmospheric $CO_2$ concentration (ppm). Since 1980, monthly data are from NOAA/ESRL (Dlugokencky and Tans, 2022) and are based on an average of direct atmospheric $CO_2$ measurements from multiple stations in the marine boundary layer (Masarie and Tans, 1995). The 1958-1979 monthly data are from the Scripps Institution of Oceanography, based on an average of direct atmospheric $CO_2$ measurements from the Mauna Loa and South Pole stations (Keeling et al., 1976). To account for the difference of mean $CO_2$ and seasonality between the NOAA/ESRL and the Scripps station networks used here, the Scripps surface average (from two stations) was de-seasonalised and adjusted to match the NOAA/ESRL surface average (from multiple stations) by adding the mean difference of 0.667 ppm, calculated here from overlapping data during 1980-2012.



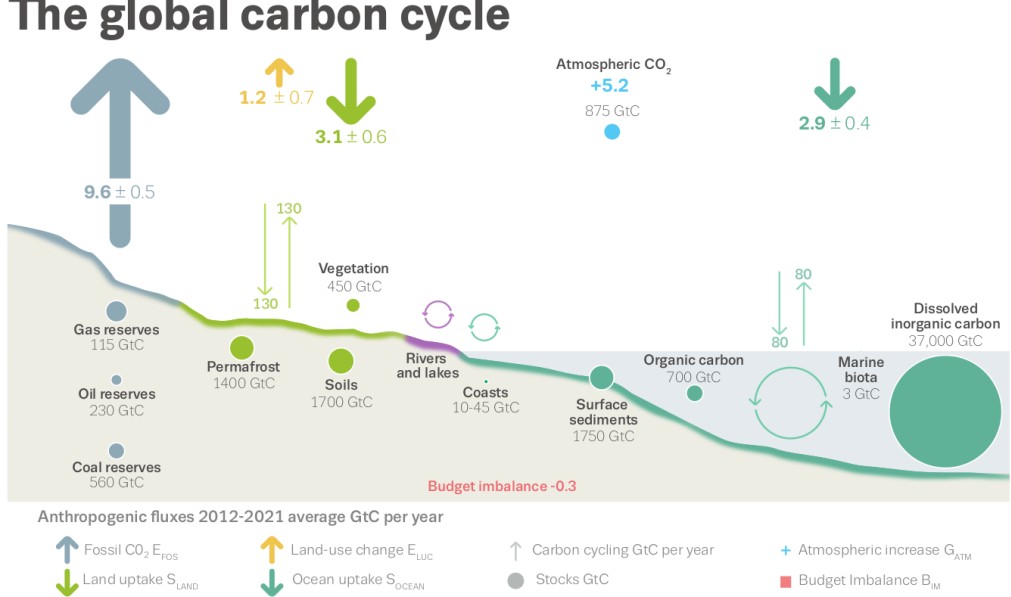

**Figure 2. Schematic representation of the overall perturbation of the global carbon cycle caused by anthropogenic activities, averaged globally for the decade 2012-2021. See legends for the corresponding arrows and units. The uncertainty in the atmospheric CO₂ growth rate is very small (±0.02 GtC yr⁻¹) and is neglected for the figure. The anthropogenic perturbation occurs on top of an active carbon cycle, with fluxes and stocks represented in the background and taken from Canadell et al. (2021) for all numbers, except for the carbon stocks in coasts which is from a literature review of coastal marine sediments (Price and Warren, 2016).**





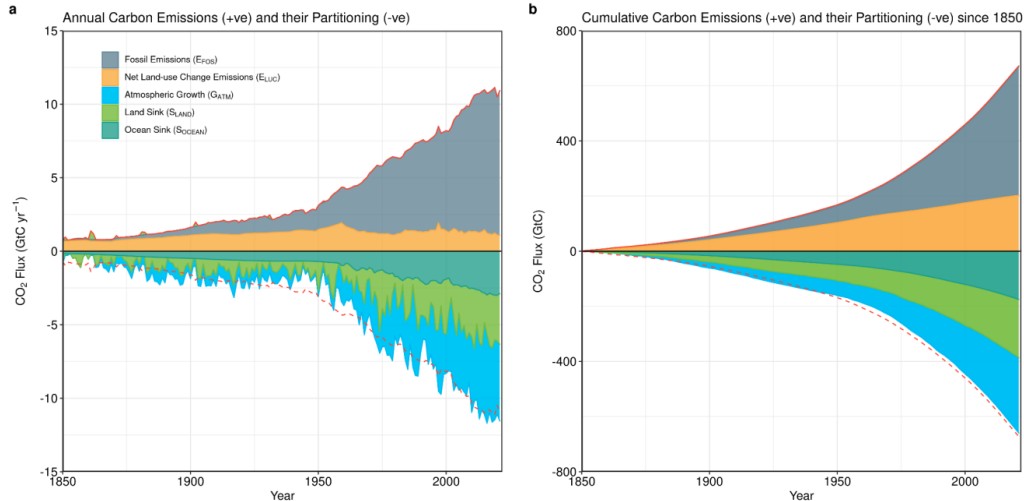

**Figure 3. Combined components of the global carbon budget illustrated in Figure 2 as a function of time, for fossil CO₂ emissions (E_FOS, including a small sink from cement carbonation; grey) and emissions from land-use change (E_LUC; brown), as well as their partitioning among the atmosphere (G_ATM; cyan), ocean (S_OCEAN; blue), and land (S_LAND; green). Panel (a) shows annual estimates of each flux and panel (b) the cumulative flux (the sum of all prior annual fluxes) since the year 1850. The partitioning is based on nearly independent estimates from observations (for G_ATM) and from process model ensembles constrained by data (for S_OCEAN and S_LAND) and does not exactly add up to the sum of the emissions, resulting in a budget imbalance (BI_M) which is represented by the difference between the bottom red line (mirroring total emissions) and the sum of carbon fluxes in the ocean, land, and atmosphere reservoirs. All data are in GtC yr⁻¹ (panel a) and GtC (panel b). The E_FOS estimate is based on a mosaic of different datasets, and has an uncertainty of ±5% (±1σ). The E_LUC estimate is from three bookkeeping models (Table 4) with uncertainty of ±0.7 GtC yr⁻¹. The G_ATM estimates prior to 1959 are from Joos and Spahni (2008) with uncertainties equivalent to about ±0.1-0.15 GtC yr⁻¹ and from Dlugokencky and Tans (2022) since 1959 with uncertainties of about +-0.07 GtC yr⁻¹ during 1959-1979 and ±0.02 GtC yr⁻¹ since 1980. The S_OCEAN estimate is the average from Khatiwala et al. (2013) and DeVries (2014) with uncertainty of about ±30% prior to 1959, and the average of an ensemble of models and an ensemble of fCO₂ data products (Table 4) with uncertainties of about ±0.4 GtC yr⁻¹ since 1959. The S_LAND estimate is the average of an ensemble of models (Table 4) with uncertainties of about ±1 GtC yr⁻¹. See the text for more details of each component and their uncertainties.**





**Figure 4. Components of the global carbon budget and their uncertainties as a function of time, presented individually for (a) fossil CO₂ and cement carbonation emissions (E_FOS), (b) growth rate in atmospheric CO₂ concentration (G_ATM), (c) emissions from land-use change (E_LUC), (d) the land CO₂ sink (S_LAND), (e) the ocean CO₂ sink (S_OCEAN), (f) the budget imbalance that is not accounted for by the other terms. Positive values of S_LAND and S_OCEAN represent a flux from the atmosphere to land or the ocean. All data are in GtC yr⁻¹ with the uncertainty bounds representing ±1 standard deviation in shaded colour. Data sources are as in Figure 3. The red dots indicate our projections for the year 2022 and the red error bars the uncertainty in the projections (see methods).**

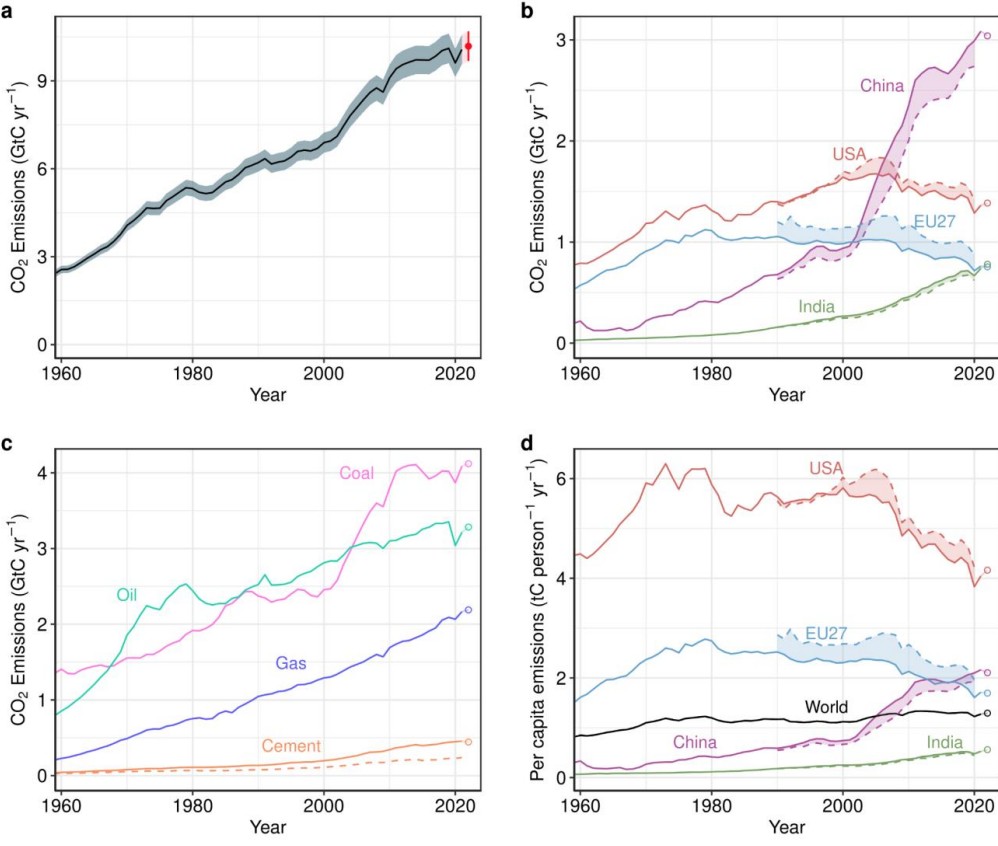



**Figure 5. Fossil CO₂ emissions for (a) the globe, including an uncertainty of ± 5% (grey shading) and a projection through the year 2022 (red dot and uncertainty range), (b) territorial (solid lines) and consumption (dashed lines) emissions for the top three country emitters (USA, China, India) and for the European Union (EU27), (c) global emissions by fuel type, including coal, oil, gas, and cement, and cement minus cement carbonation (dashed), and (d) per-capita emissions the world and for the large emitters as in panel (b). Territorial emissions are primarily from a draft update of Gilfillan and Marland (2021) except for national data for Annex I countries for 1990-2020, which are reported to the UNFCCC as detailed in the text, as well as some improvements in individual countries, and extrapolated forward to 2021 using BP Energy Statistics. Consumption-based emissions are updated from Peters et al. (2011b). See Section 2.1 and Appendix C.1 for details of the calculations and data sources.**

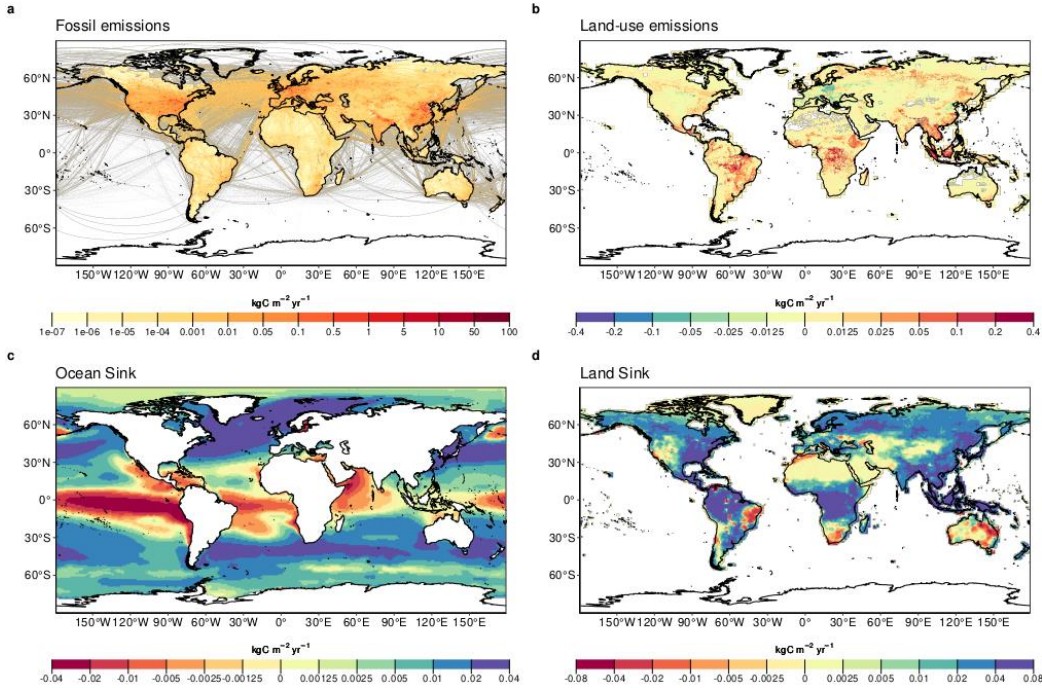

**Figure 6. The 2012-2021 decadal mean components of the global carbon budget, presented for (a) fossil CO₂ emissions (E_FOS), (b) land-use change emissions (E_LUC), (c) the ocean CO₂ sink (S_OCEAN), and (d) the land CO₂ sink (S_LAND). Positive values for E_FOS and E_LUC represent a flux to the atmosphere, whereas positive values of S_OCEAN and S_LAND represent a flux from the atmosphere to the ocean or the land. In all panels, yellow/red (green/blue) colours represent a flux from (into) the land/ocean to (from) the atmosphere. All units are in kgC m⁻² yr⁻¹. Note the different scales in each panel. E_FOS data shown is from GCP-GridFEDv2022.2. E_LUC data shown is only from BLUE as the updated H&N2017 and OSCAR do not resolve gridded fluxes. S_OCEAN data shown is the average of GOBMs and data-products means, using GOBMs simulation A, no adjustment for bias and drift applied to the gridded fields (see Section 2.4). S_LAND data shown is the average of DGVMs for simulation S2 (see Section 2.5).**



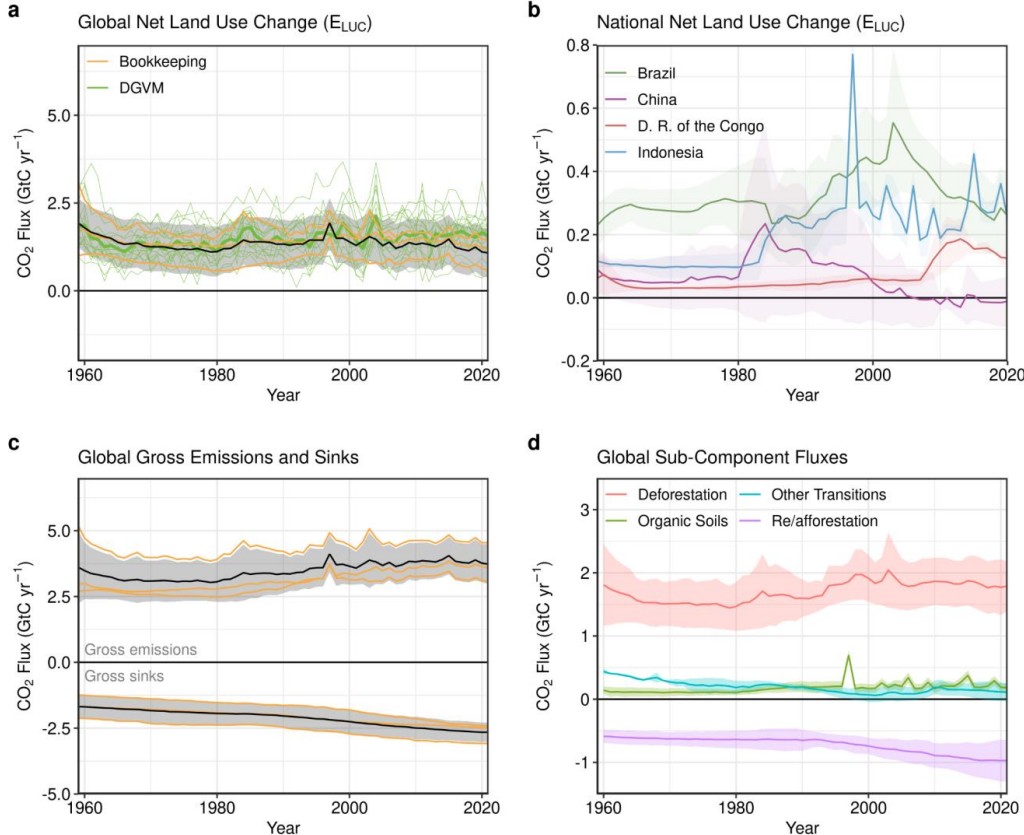

**Figure 7. Net CO$_2$ exchanges between the atmosphere and the terrestrial biosphere related to land use change. (a) Net CO$_2$ emissions from land-use change (E$_{LUC}$) with estimates from the three bookkeeping models (yellow lines) and the budget estimate (black with ±1σ uncertainty), which is the average of the three bookkeeping models. Estimates from individual DGVMs (narrow green lines) and the DGVM ensemble mean (thick green line) are also shown. (b) Net CO$_2$ emissions from land-use change from the four countries with largest cumulative emissions since 1959. Values shown are the average of the three bookkeeping models. (c) CO$_2$ gross sinks (negative, from regrowth after agricultural abandonment and wood harvesting) and gross sources (positive, from decaying material left dead on site, products after clearing of natural vegetation for agricultural purposes, wood harvesting, and, for BLUE, degradation from primary to secondary land through usage of natural vegetation as rangeland, and also from emissions from peat drainage and peat burning). Values are shown for the three bookkeeping models (yellow lines) and for their average (black with ±1σ uncertainty). The sum of the gross sinks and sources is E$_{LUC}$ shown in panel (a). (d) Sources and sinks aggregated into four components that contribute to the net fluxes of CO$_2$, including: (i) gross sources from deforestation; (ii) net flux on forest lands (slash and product decay following wood harvest; sinks due to regrowth after wood harvest or after abandonment, including reforestation and in shifting cultivation cycles; afforestation), (iii) emissions from organic soils (peat drainage and pear fire, and (iv) sources and sinks related to other land use transitions. The scale of the fluxes shown is smaller than in panel (c) because the substantial gross sources and sinks from wood harvesting are accounted for as net flux under (ii) . The sum of the component fluxes is E$_{LUC}$ shown in panel (a).**

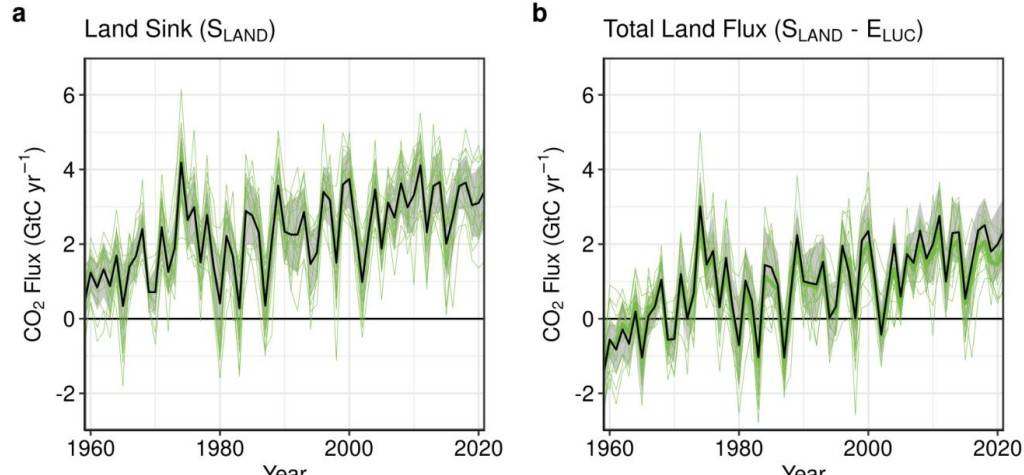

**Figure 8: (a) The land CO₂ sink (S$_{LAND}$) estimated by individual DGVMs estimates (green), as well as the budget estimate (black with ±1σ uncertainty), which is the average of all DGVMs. (b) Total atmosphere-land CO₂ fluxes (S$_{LAND}$ − E$_{LUC}$). The budget estimate of the total land flux (black with ±1σ uncertainty) combines the DGVM estimate of S$_{LAND}$ from panel (a) with the bookkeeping estimate of E$_{LUC}$ from Figure 7(a). Uncertainties are similarly propagated in quadrature from the budget estimates of S$_{LAND}$ from panel (a) and E$_{LUC}$ from Figure 7(a). DGVMs also provide estimates of E$_{LUC}$ (see Figure 7(a)), which can be combined with their own estimates of the land sink. Hence panel (b) also includes an estimate for the total land flux for individual DGVMs (thin green lines) and their multi-model mean (thick green line).**



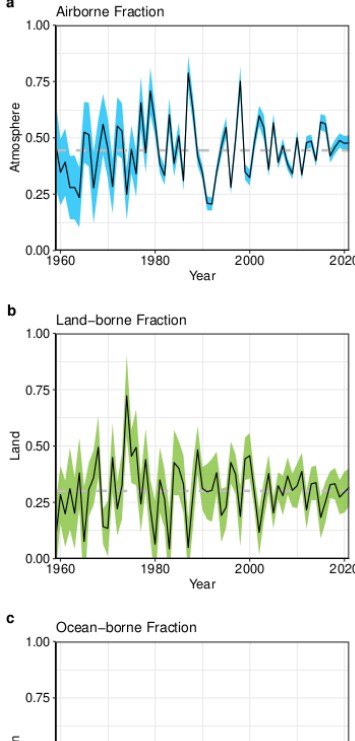

**Figure 9.** The partitioning of total anthropogenic $CO_2$ emissions ($E_{FOS}$ + $E_{LUC}$) across (a) the atmosphere (airborne fraction), (b) land (land-borne fraction), and (c) ocean (ocean-borne fraction). Black lines represent the central estimate, and the coloured shading represents the uncertainty. The grey dashed lines represent the long-term average of the airborne (44%), land-borne (30%) and ocean-borne (25%) fractions during 1960-2021.

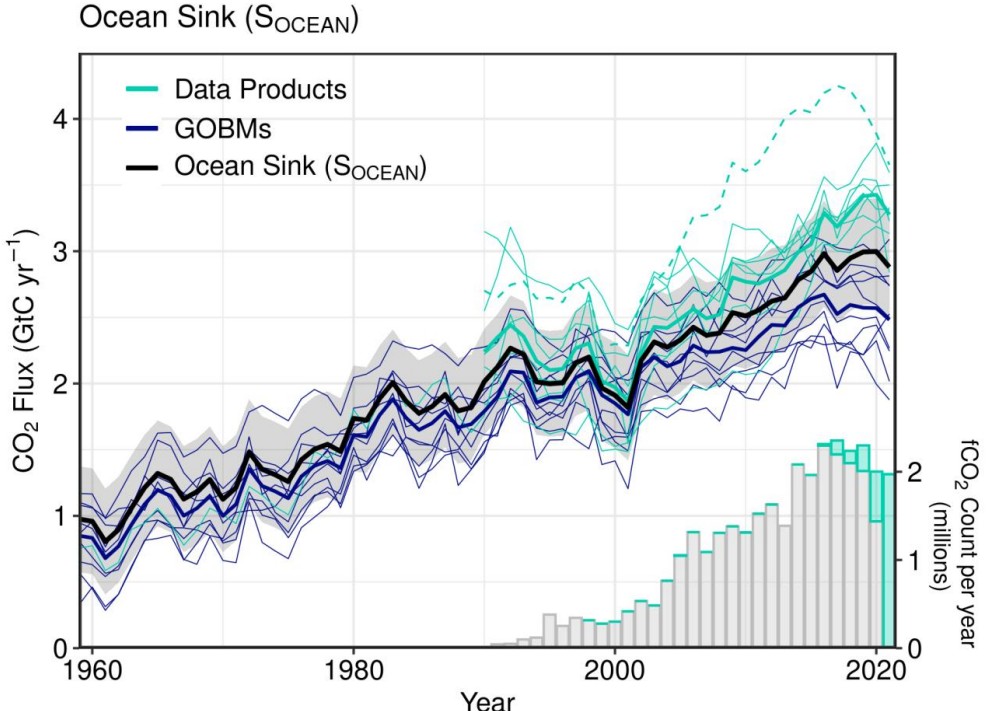

**Figure 10. Comparison of the anthropogenic atmosphere-ocean $CO_2$ flux showing the budget values of $S_{OCEAN}$ (black; with the uncertainty in grey shading), individual ocean models (royal blue), and the ocean $fCO_2$-based data products (cyan; with Watson et al. (2020) in dashed line as not used for ensemble mean). Only one data product (Jena-MLS) extends back to 1959 (Rödenbeck et al., 2022). The $fCO_2$-based data products were adjusted for the pre-industrial ocean source of $CO_2$ from river input to the ocean, by subtracting a source of 0.65 $GtC\ yr^{-1}$ to make them comparable to $S_{OCEAN}$ (see Section 2.4). Bar-plot in the lower right illustrates the number of $fCO_2$ observations in the SOCAT v2022 database (Bakker et al., 2022). Grey bars indicate the number of data points in SOCAT v2021, and coloured bars the newly added observations in v2022.**

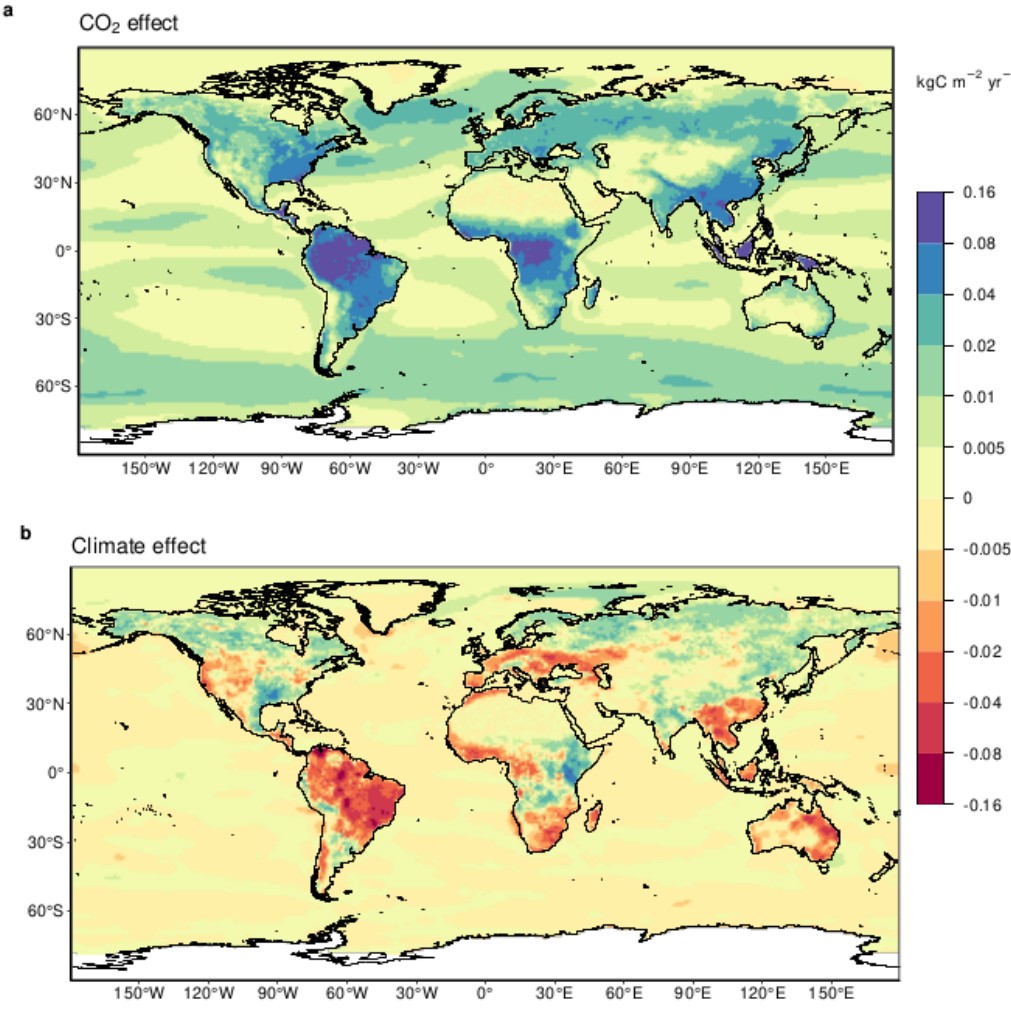

**Figure 11. Attribution of the atmosphere-ocean (S$_{OCEAN}$) and atmosphere-land (S$_{LAND}$) CO$_2$ fluxes to (a) increasing atmospheric CO$_2$ concentrations and (b) changes in climate, averaged over the previous decade 2012-2021. All data shown is from the processed-based GOBMs and DGVMs. The sum of ocean CO$_2$ and climate effects will not equal the ocean sink shown in Figure 6 which includes the fCO$_2$-based data products. See Appendix C.3.2 and C.4.1 for attribution methodology. Units are in kgC m$^{-2}$ yr$^{-1}$ (note the non-linear colour scale).**



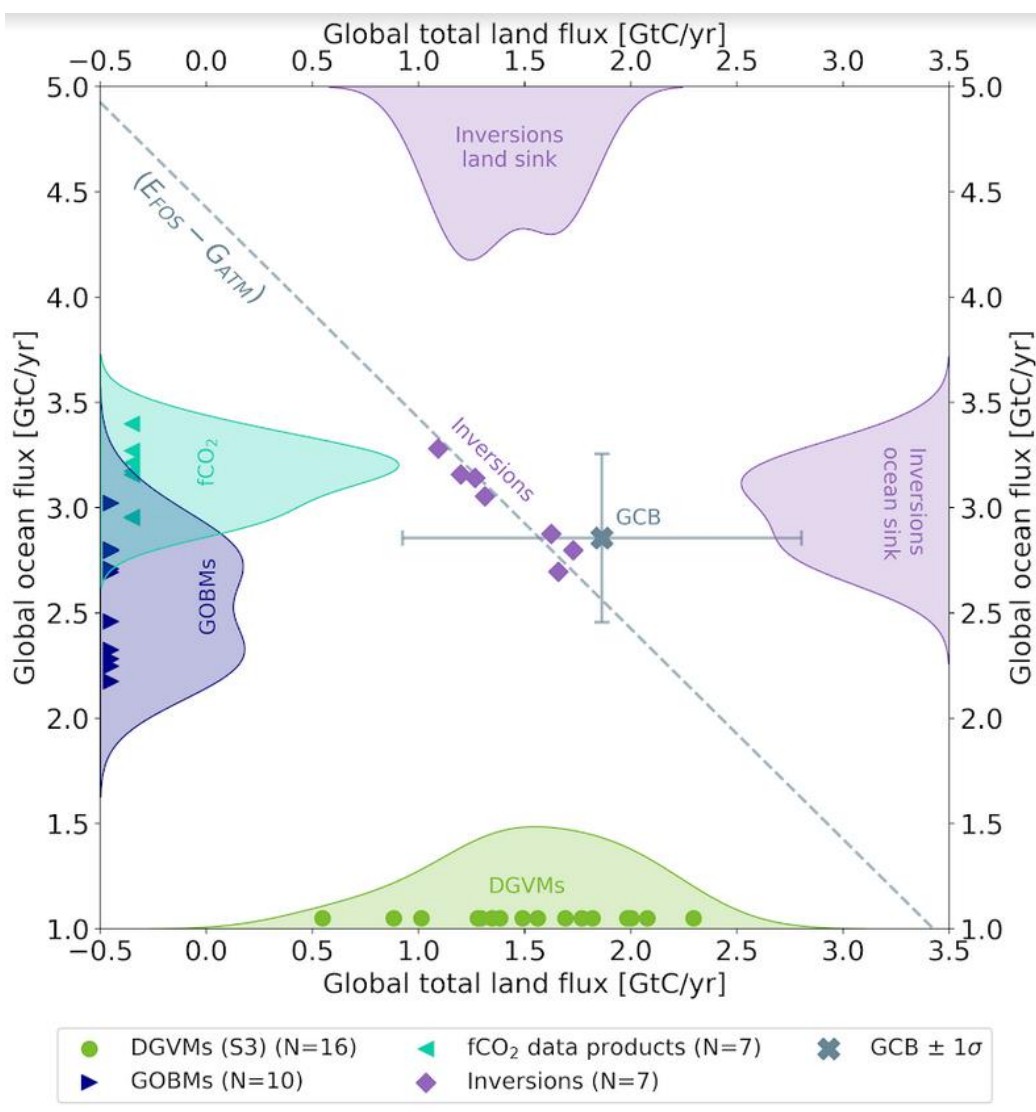

**Figure 12. The 2012-2021 decadal mean net atmosphere-ocean and atmosphere-land fluxes derived from the ocean models and fCO2 products (y-axis, right and left pointing blue triangles respectively), and from the DGVMs (x-axis, green symbols), and the same fluxes estimated from the six inversions (purple symbols on secondary x- and y-axis). The grey central point is the mean (±1σ) of $S_{OCEAN}$ and ($S_{LAND} - E_{LUC}$) as assessed in this budget. The shaded distributions show the density of the ensemble of individual estimates. The grey diagonal band represents the fossil fuel emissions minus the atmospheric growth rate from this budget ($E_{FOS} - G_{ATM}$). Note that positive values are $CO_2$ sinks.**

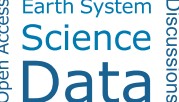

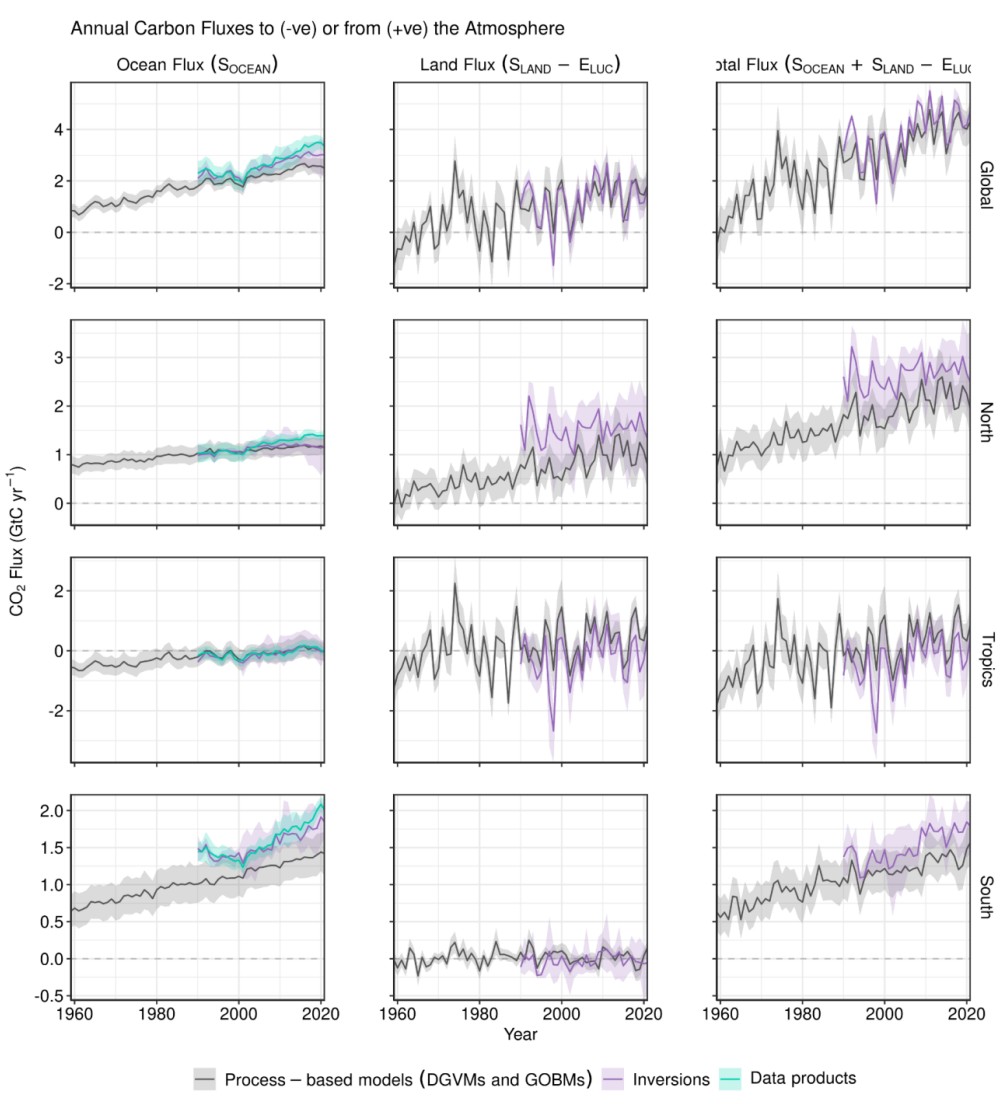



**Figure 13. CO₂ fluxes between the atmosphere and the Earth's surface separated between land and oceans, globally and in three latitude bands. The ocean flux is S$_{OCEAN}$ and the land flux is the net atmosphere-land fluxes from the DGVMs. The latitude bands are (top row) global, (2$^{nd}$ row) north (>30°N), (3$^{rd}$ row) tropics (30°S-30°N), and (bottom row) south (<30°S), and over ocean (left column), land (middle column), and total (right column). Estimates are shown for: process-based models (DGVMs for land, GOBMs for oceans); inversion systems (land and ocean); and fCO₂-based data products (ocean only). Positive values indicate a flux from the atmosphere to the land or the ocean. Mean estimates from the combination of the process models for the land and oceans are shown (black line) with ±1 standard deviation (1σ) of the model ensemble (grey shading). For the total uncertainty in the process-based estimate of the total sink, uncertainties are summed in quadrature. Mean estimates from the atmospheric inversions are shown (purple lines) with their full spread (purple shading). Mean estimates from the fCO₂-based data products are shown for the ocean domain (light blue lines) with their ±1σ spread (light blue shading). The global S$_{OCEAN}$ (upper left) and the sum of S$_{OCEAN}$ in all three regions represents the anthropogenic atmosphere-to-ocean flux based on the assumption that the preindustrial ocean sink was 0 GtC yr$^{-1}$ when riverine fluxes are not considered. This assumption does not hold at the regional level, where preindustrial fluxes can be significantly different from zero. Hence, the regional panels for S$_{OCEAN}$ represent a combination of natural and anthropogenic fluxes. Bias-correction and area-weighting were only applied to global S$_{OCEAN}$; hence the sum of the regions is slightly different from the global estimate (<0.05 GtC yr$^{-1}$).**





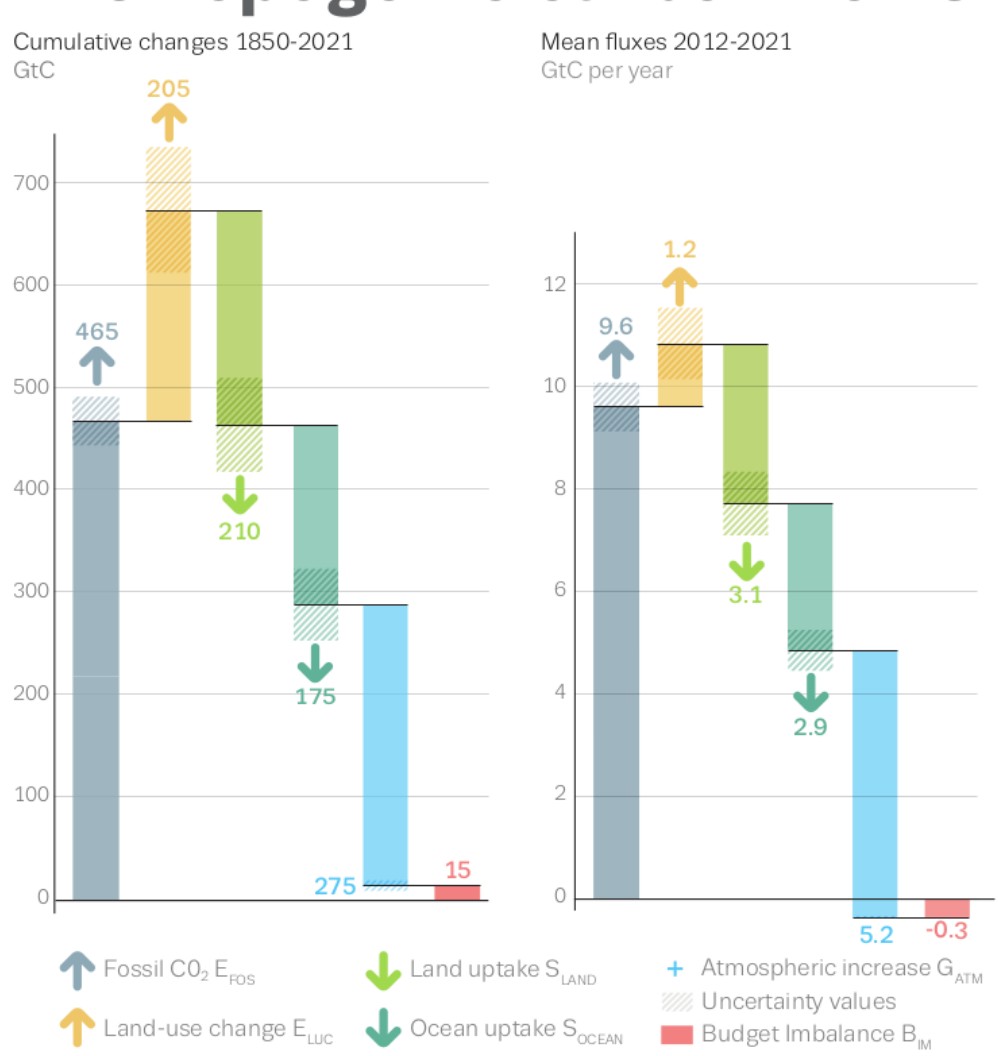

**Figure 14. Cumulative changes over the 1850-2021 period (left) and average fluxes over the 2012-2021 period (right) for the anthropogenic perturbation of the global carbon cycle. See the caption of Figure 3 for key information and the methods in text for full details.**



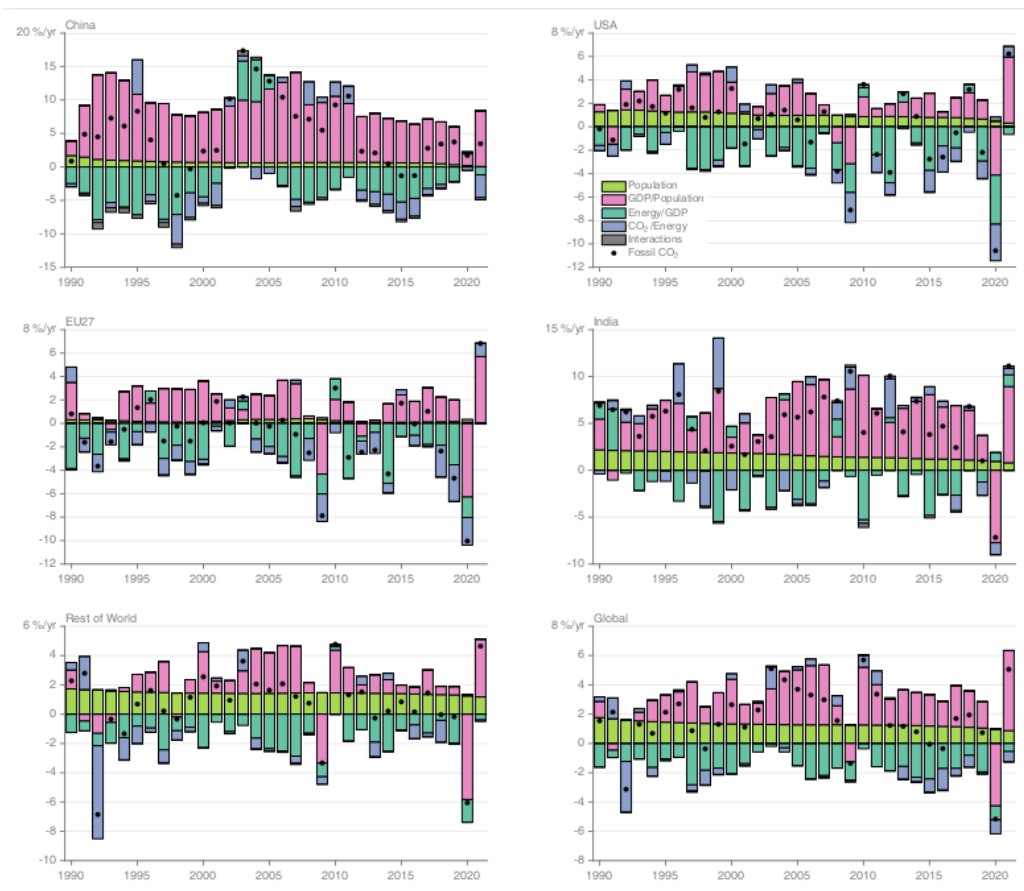

**Figure 15. Kaya decomposition of the main drivers of fossil CO₂ emissions, considering population, GDP per person, Energy per GDP, and CO₂ emissions per energy, for China (top left), USA (top right), EU27 (middle left), India (middle right), Rest of the World (bottom left), and World (bottom right). Black dots are the annual fossil CO₂ emissions growth rate, coloured bars are the contributions from the different drivers. A general trend is that population and GDP growth put upward pressure on emissions, while energy per GDP and more recently CO₂ emissions per energy put downward pressure on emissions. Both the COVID-19 induced changes during 2020 and the recovery in 2021 led to a stark contrast to previous years, with different drivers in each region.**





# Appendix A. Supplementary Tables

Table A1. Comparison of the processes included in the bookkeeping method and DGVMs in their estimates of ELUC and SLAND. See Table 4 for model references. All models include deforestation and forest regrowth after abandonment of agriculture (or from afforestation activities on agricultural land). Processes relevant for ELUC are only described for the DGVMs used with land-cover change in this study.

| | Bookkeeping Models | | | DGVMs | | | | | | | | | | | | | | | |
|---|---|---|---|---|---|---|---|---|---|---|---|---|---|---|---|---|---|---|---|
| | H&N | BLUE | OSCAR | CABLE-POP | CLASSIC | CLM5.0 | DLEM | IBIS | ISAM | JSBACH | JULES-ES | LPJ-GUESS | LPJ | LPX-Bern | OCNv2 | ORCHIDEEv3 | SDGVM | VISIT | YIBs |
| **Processes relevant for ELUC** | | | | | | | | | | | | | | | | | | | |
| Wood harvest and forest degradation (a) | yes | yes | yes | yes | no | yes | yes | yes | yes | yes | no | yes | yes | no (d) | yes | yes | no | yes | no |
| Shifting cultivation / Subgrid scale transitions | yes (b) | yes | yes | yes | no | yes | no | yes | no | yes | no | yes | yes | no (d) | no | no | no | yes | no |
| Cropland harvest (removed, R, or added to litter, L) | yes (R) (j) | yes (R) (j) | yes (R) | yes (R) | yes (L) | yes (R) | yes | yes (R) | yes | yes (R+L) | yes (R) | yes (R) | yes (L) | yes (R) | yes (R+L) | yes (R) | yes (R) | yse (R) | yes (L) |
| Peat fires | yes | yes | yes | no | no | yes | no | no | no | no | no | no | no | no | no | no | no | no | no |
| fire as a management tool | yes (j) | yes (j) | yes (h) | no | no | no | no | no | no | no | no | no | no | no | no | no | no | no | no |
| N fertilisation | yes (j) | yes (j) | yes (h) | no | no | yes | no | yes | no | yes(i) | yes | no | yes | yes | yes | no | no | no | no |
| tillage | yes (j) | yes (j) | yes (h) | no | yes (g) | no | no | no | no | no | no | yes | no | no | no | yes (g) | no | no | no |
| irrigation | yes (j) | yes (j) | yes (h) | no | no | yes | yes | no | yes | no | no | yes | no | no | no | no | no | no | no |
| wetland drainage | yes (j) | yes (j) | yes (h) | no | no | no | no | no | no | no | no | no | no | no | no | no | no | no | no |
| erosion | yes (j) | yes (j) | yes (h) | no | no | no | yes | no | no | no | no | no | no | no | no | no | no | yes | no |
| peat drainage | yes | yes | yes | no | no | no | no | no | no | no | no | no | no | no | no | no | no | no | no |
| Grazing and mowing Harvest (removed, r, or added to litter, l) | yes (r) (j) | yes (r) (j) | yes (r) | yes (r) | no | no | no | yes (r, l) | yes (l) | no | yes (r) | yes (l) | no | yes (r+l) | no | no | no | no | no |
| **Processes also relevant for SLAND (in addition to CO2 fertilisation and climate)** | | | | | | | | | | | | | | | | | | | |
| Fire simulation and/or suppression | N.A. | N.A. | N.A. | no | yes | yes | no | yes | no | yes | yes | yes | yes | yes | no | no | yes | yes | no |
| Carbon-nitrogen interactions, including N deposition | N.A. | N.A. | N.A. | yes | no (f) | yes | yes | no | yes | yes | yes | yes | no | yes | yes | yes (c) | no | no | no (f) |
| Separate treatment of direct and diffuse solar radiation | N.A. | N.A | N.A | yes | no | yes | no | no | no | no | yes | no | no | no | no | no | no | no | yes |

(a) Refers to the routine harvest of established managed forests rather than pools of harvested products.

(b) No back- and forth-transitions between vegetation types at the country-level, but if forest loss based on FRA exceeded agricultural expansion based on FAO, then this amount of area was cleared for cropland and the same amount of area of old croplands abandoned.

(c) Limited. Nitrogen uptake is simulated as a function of soil C, and Vcmax is an empirical function of canopy N. Does not consider N deposition.

(d) Available but not active.

(e) Simple parameterization of nitrogen limitation based on Yin (2002; assessed on FACE experiments)

(f) Although C-N cycle interactions are not represented, the model includes a parameterization of down-regulation of photosynthesis as CO2 increases to emulate nutrient constraints (Arora et al., 2009)

(g) Tillage is represented over croplands by increased soil carbon decomposition rate and reduced humification of litter to soil carbon.

(h) as far as the DGVMs that OSCAR is calibrated to include it

(i) perfect fertilisation assumed, i.e. crops are not nitrogen limited and the implied fertiliser diagnosed

(j) Process captured implicitly by use of observed carbon densities.



**Table A2. Comparison of the processes and model set up for the Global Ocean Biogeochemistry Models for their estimates of SOCEAN. See Table 4 for model references.**

| | NEMO-PlankTOM 12 | NEMO-PISCES (IPSL) | MICOM-HAMOCC (NorESM1-OCv1.2) | MPIOM-HAMOCC 6 | FESOM-2.1-REcoM2 | NEMO3.6-PISCESv2-gas (CNRM) | MOM6-COBALT (Princeton) | CESM-ETHZ | MRI-ESM2-1 | CESM2 |
|---|---|---|---|---|---|---|---|---|---|---|
| **Model specifics** | | | | | | | | | | |
| Physical ocean model | NEMOv3.6-ORCA2 | NEMOv3.6-eORCA1L75 | MICOM (NorESM1-OCv1.2) | MPIOM | FESOM-2.1 | NEMOv3.6-GELATOv6-eORCA1L75 | MOM6-SIS2 | CESMv1.3 (ocean model based on POP2) | MRI.COMv4 | CESM2-POP2 |
| Biogeochemistry model | PlankTOM 12 | PISCESv2 | HAMOCC (NorESM1-OCv1.2) | HAMOCC6 | REcoM-2-M | PISCESv2-gas | COBALTv2 | BEC (modified & extended) | NPZD | MARBL |
| Horizontal resolution | 2° lon, 0.3 to 1.5° lat | 1° lon, 0.3 to 1° lat | 1° lon, 0.17 to 0.25 lat | 1.5° | unstructured mesh, 20-120 km resolution (CORE mesh) | 1° lon, 0.3 to 1° lat | 0.5° lon, 0.25 to 0.5° lat | 1.125° lon, 0.53° to 0.27° lat | 1° lon, 0.3 to 0.5° lat | 1.125° lon, 0.53° to 0.27° lat |
| Vertical resolution | 31 levels | 75 levels, 1m at the surface | 51 isopycnic layers + 2 layers representing a bulk mixed layer | 40 levels | 46 levels, 10 m spacing in the top 100 m | 75 levels, 1m at surface | 75 levels hybrid coordinates, 2m at surface | 60 levels | 60 levels with 1-level bottom boundary layer | 60 levels |
| Total ocean area on native grid (km2) | 3.6080E+08 | 3.6270E+08 | 3.6006E+08 | 3.6598E+08 | 3.6435E+08 | 3.6270E+14 | 3.6111E+08 | 3.5926E+08 | 3.6141E+08 | 3.61E+08 |
| Gas-exchange parameterization | Wanninkhof et al. 1992 | Orr et al., 2017 | Orr et al., 2017, but with a=0.337 | Orr et al., 2017 | Orr et al., 2017 | Orr et al., 2017 | Orr et al., 2017 | Wanninkhof (1992, coefficient a scaled down to 0.31) | Orr et al., 2017 | Orr et al., 2017 |
| CO2 chemistry routines | Following Broecker et al. (1982) | mocsy | Following Dickson et al. 2007 | Ilyina et al. (2013) adapted to comply with OMIP protocol (Orr et al., 2017) | mocsy | mocsy | mocsy | OCMIP2 (Orr et al.) | mocsy | OCMIP2 (Orr et al. 2017) |
| River input (PgC/yr) (organic/inorganic DIC) | 0.723 / - | 0.61 / - | 0 | 0.77 / - | 0 / 0 | ~0.611 / - | ~0.07 / ~0.15 | 0.33 / - | 0 / 0 | 0.173/0.263 |
| Net flux to sediment (PgC/yr) (organic/other) | 0.723 / - | 0.59 / - | around 0.54 / - | - / 0.44 | 0 / 0 | ~0.656 / - | ~0.11 / ~0.07 (CaCO3) | 0.21 / - | 0 / 0 | 0.345/0.110 (CaCO3) |
| **SPIN-UP procedure** | | | | | | | | | | |
| Initialisation of carbon chemistry | GLODAPv1 (preindustrial DIC) | GLODAPv2 (preindustrial DIC) | GLODAPv1 (preindustrial DIC) | initialization from previous simulation | GLODAPv2 (preindustrial DIC) | GLODAPv2 | GLODAPv2 (Alkalinity, DIC). DIC | GLODAPv2 (preindustrial DIC) | GLODAPv2 (preindustrial | GLODAPv2 (preindustrial |



| | | | | | | | | | | |
|---|---|---|---|---|---|---|---|---|---|---|
| | | | | | | | corrected to 1959 level (simulation A and C) and to pre-industrial level (simulation B and D) using Khatiwala et al 2009 | | DIC) | DIC) |
| Preindustrial spin-up prior to 1850 | spin-up 1750-1947 | spin-up starting in 1836 with 3 loops of JRA55 | 1000 year spin up | ~2000 years | 189 years | long spin-up (> 1000 years) | Other bgc tracers initialized from a GFDL-ESM2M spin-up (> 1000 years) | spinup 1655-1849 | 1661 years with xCO2 = 284.32 | spinup 1653-1850, xCO2=278 |
| **Atmospheric forcing fields and CO2** | | | | | | | | | | |
| Atmospheric forcing for (i) pre-industrial spin-up, (ii) spin-up 1850-1958 for simulation B, (iii) simulation B | looping NCEP year 1990 (i, ii, iii) | looping full JRA55 reanalysis | CORE-I (normal year) forcing (i, ii, iii) | OMIP climatology (i), NCEP year 1957 (ii,iii) | JRA55-do v.1.5.0 repeated year 1961 (i, ii, iii) | JRA55-do-v1.5.0 full reanaylsis (i) cycling year 1958 (ii,iii) | GFDL-ESM2M internal forcing (i), JRA55-do-v1.5.0 repeat year 1959 (ii,iii) | COREv2 until 1835 , from 1835-1850: JRA (i), normal year forcing created from JRA55-do version 1.3 (ii,iii) | JRA55-do v1.5.0 repeat year 1990/91 (i, ii, iii) | (i) repeating JRA 1958-2018 for spinup for A & D, repeating JRA 1990/1991 repeat year forcing for spinup for B & C, (ii) & (iii) JRA 1990/1991 repeat year forcing |
| Atmospheric CO2 for control spin-up 1850-1958 for simulation B, and for simulation B | constant 278ppm; converted to pCO2 temperature formulation (Sarmiento et al., 1992) | xCO2 of 286.46ppm, converted to pCO2 with constant sea-level pressure and water vapour pressure | xCO2 of 278ppm, converted to pCO2 with sea-level pressure and water vapour pressure | xCO2 of 278ppm, no conversion to pCO2 | xCO2 of 278ppm, converted to pCO2 with sea-level pressure and water vapour pressure | xCO2 of 286.46ppm, converted to pCO2 with constant sea-level pressure and water vapour pressure | xCO2 of 278ppm, converted to pCO2 with sea-level pressure and water vapour pressure | xCO2 = 287.4ppm, converted to pCO2 with atmospheric pressure, and water vapour pressure | xCO2 of 284.32ppm (CMIP6 piControl), converted to pCO2 with water vapour and sea-level pressure (JRA55- | xCO2=278 |

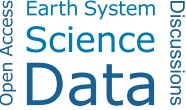

| | | | | | | | | do repeat year 1990/91) | |
|---|---|---|---|---|---|---|---|---|---|
| Atmospheric forcing for historical spin-up 1850-1958 for simulation A (i) and for simulation A (ii) | 1750-1947: looping NCEP year 1990; 1948-2021: NCEP | 1836-1958: looping full JRA55 reanalysis (i), JRA55-do-v1.4 then 1.5 for 2020-21 (ii) | CORE-I (normal year) forcing; from 1948 onwards NCEP-R1 with CORE-II corrections | NCEP 6 hourly cyclic forcing (10 years starting from 1948, i), 1948-2021: transient NCEP forcing | JRA55-do-v1.5.0 repeated year 1961 (i), transient JRA55-do-v1.5.0 (ii) | JRA55-do cycling year 1958 (i), JRA55-do-v1.5.0 (ii) | JRA55-do-v1.5 repeat year 1959 (i), v1.5.0 (1959-2019, v1.5.0.1b (2020), v1.5.0.1 (2021; ii) | JRA55 version 1.3, repeat cycle between 1958-2018 (i), v1.3 (1959-2018), v.1.5.0.1 (2020-2021) | 1653-1957: repeated cycle JRA55-do v1.5.0 1958-2018 (i), v1.5.0 (1958-2018), v1.5.0.1 (2019-2021; ii) | (i) repeating JRA 1958-2018, (ii) JRA 1958-2021 |
| Atmospheric CO2 for historical spin-up 1850-1958 for simulation A (i) and simulation A (ii) | xCO2 provided by the GCB; converted to pCO2 temperature formulation (Sarmiento et al., 1992), monthly resolution (i, ii) | xCO2 as provided by the GCB, global mean, annual resolution, converted to pCO2 with sea-level pressure and water vapour pressure (i, ii) | xCO2 as provided by the GCB, converted to pCO2 with sea level pressure (taken from the atmopheric forcing) and water vapor correction (i, ii) | transient monthly xCO2 provided by GCB, no conversion (i, ii) | xCO2 as provided by the GCB, converted to pCO2 with sea-level pressure and water vapour pressure, global mean, monthly resolution (i, ii) | xCO2 as provided by the GCB, converted to pCO2 with constant sea-level pressure and water vapour pressure, global mean, yearly resolution (i, ii) | xCO2 at year 1959 level (315 ppm, i) and as provided by GCB (ii), both converted to pCO2 with sea-level pressure and water vapour pressure, global mean, yearly resolution | xCO2 as provided by the GCB, converted to pCO2 with locally determined atm. pressure, and water vapour pressure (i, ii) | xCO2 as provided for CMIP6 historical simulations, annual resolution (i), and as provided by GCB (ii), both converted to pCO2 with water vapour and sea-level pressure | annual global xCO2 provided by GCB, converted to equilibrium CO2* using atmospheric pressure and Weiss and Price (1980) |




| | Jena-MLS | MPI-SOMFFN | CMEMS-LSCE-FFNN | Watson et al | NIES-NN | JMA-MLR | OS-ETHZ-GRaCER | LDEO HPD |
|---|---|---|---|---|---|---|---|---|
| **Method** | Spatio-temporal interpolation (version oc_v2022). Spatio-temporal field of ocean-internal carbon sources/sinks is fit to the SOCATv2022 pCO2 data. Includes a multi-linear regression against environmental drivers to bridge data gaps, | A feed-forward neural network (FFN) determines non-linear relationship between SOCAT pCO2 measurements and environmental predictor data for 16 biogeochemical provinces (defined through a self-organizing map, SOM) and is used to fill the existing data gaps. | An ensemble of neural network models trained on 100 subsampled datasets from SOCAT and environmental predictors. The models are used to reconstruct sea surface fugacity of CO2 and convert to air-sea CO2 fluxes | Modified MPI-SOMFFN with SOCATv2022 pCO2 database. Corrected to the subskin temperature of the ocean as measured by satellite (Goddijn-Murphy et al, 2015). Flux calculation corrected for the cool and salty surface skin. Monthly climatology for skin temperature correction derived from ESA CCI product for the period 2003 to 2011 (Merchant et al, 2019). | A feed forward neural network model trained on SOCAT 2021 fCO2 and environmental predictor data. The fCO2 was normalized to the reference year 2000 by a global fCO2 trend: We fitted the dependence of fCO2 on year by linear regression. We subtracted the trend from fCO2 and used the neural network to model the nonlinear dependence of the residual on predictors. The trend was added to model predictions to reconstruct fCO2. | Fields of total alkalinity (TA) were estimated by using a multiple linear regressions (MLR) method based on GLODAPv2.2021 and satellite observation data. SOCATv2022 fCO2 data were converted to dissolved inorganic carbon (DIC) with the TA. Fields of DIC were estimated by using a MLR method based on the DIC and satellite observation data | Geospatial Random Cluster Ensemble Regression is a two-step cluster-regression approach, where multiple clustering instances with slight variations are run to create an ensemble of estimates. We use K-means clustering and a combination of Gradient boosted trees and Feed-forward neural-networks to estimate SOCAT v2022 fCO2. | Based on fCO2-misfit between observed fCO2 and eight of the ocean biogeochemical models used in this assessment. The eXtreme Gradient Boosting method links this misfit to environmental observations to reconstruct the model misfit across all space and time., which is then added back to model-based fCO2 estimate. The final reconstrucion of surface fCO2 is the average across the eight reconstructions. |
| **Gas-exchange parameterizatio n** | Wanninkhof 1992. Transfer coefficient k scaled to match a global mean transfer rate of 16.5 cm/hr by (Naegler, 2009) | Wanninkhof 1992. Transfer coefficient k scaled to match a global mean transfer rate of 16.5 cm/hr | Wanninkhof 2014. Transfer coefficient k scaled to match a global mean transfer rate of 16.5 cm/hr (Naegler, 2009) | Nightingale et al 2000 | Wanninkhof, 2014. Transfer coefficient k scaled to match a global mean transfer rate of 16.5 cm/hr (Naegler, 2009) | Wanninkhof., 2014. Transfer coefficient k scaled to match a global mean transfer rate of 16.5 cm/hr (Naegler, 2009) | Wanninkhof 1992, averaged and scaled for three reanalysis wind data, to a global mean 16.5 cm/hr (after Naegler 2009; Fay & Gregor et al. 2021) | Wanninkhof 1992, averaged and scaled for three reanalysis wind data, to a global mean 16.5 cm/hr (after Naegler 2009; Fay & Gregor et al. 2021) |
| **Wind product** | JMA55-do reanalysis | ERA 5 | ERA5 | Mean and mean square winds monthly 1x1° from CCMP, 0.25x0.25° x 6-hourly, | ERA5 | JRA55 | JRA55, ERA5, NCEP1 | JRA55, ERA5, CCMP2 |
| **Spatial resolution** | 2.5 degrees longitude x 2 degrees latitude | 1x1 degree | 1x1 degree | 1x1 degree | 1x1 degree | 1x1 degree | 1x1 degree | 1x1 degree |
| **Temporal resolution** | daily | monthly | monthly | monthly | monthly | monthly | monthly | monthly |

Table A3: Description of ocean data-products used for assessment of SOCEAN. See Table 4 for references.




| Atmospheric CO2 | Spatially and temporally varying field based on atmospheric CO2 data from 169 stations (Jena CarboScope atmospheric inversion sEXTALL_v2021) | Spatially varying 1x1 degree atmospheric pCO2_wet calculated from the NOAA ESRL marine boundary layer xCO2 and NCEP sea level pressure with the moisture correction by Dickson et al 2007. | Spatially and monthly varying fields of atmospheric pCO2 computed from CO2 mole fraction (CO2 atmospheric inversion from the Copernicus Atmosphere Monitoring Service), and atmospheric dry-air pressure which is derived from monthly surface pressure (ERA5) and water vapour pressure fitted by Weiss and Price 1980 | Atmospheric pCO2 (wet) calculated from NOAA marine boundary layer XCO2 and NCEP sea level pressure, with pH2O calculated from Cooper et al, 1998. 2021 XCO2 marine boundary values were not available at submission so we used preliminary values, estimated from 2020 values and increase at Mauna Loa. | NOAA Greenhouse Gas Marine Boundary Layer Reference. https://gml.noaa.gov/ccgg/mbl/mbl.html | Atmospheric xCO2 fields of JMA-GSAM inversion model (Maki et al. 2010; Nakamura et al. 2015) were used. They were converted to pCO2 by using JRA55 sea level pressure. 2021 xCO2 fields were not available at this stage, and we used global xCO2 increments from 2020 to 2021. | NOAA's marine boundary layer product for xCO2 is linearly interpolated onto a 1x1 degree grid and resampled from weekly to monthly. xCO2 is multiplied by ERA5 mean sea level pressure, where the latter corrected for water vapour pressure using Dickson et al. (2007). This results in monthly 1x1 degree pCO2atm. | NOAA's marine boundary layer product for xCO2 is linearly interpolated onto a 1x1 degree grid and resampled from weekly to monthly. xCO2 is multiplied by ERA5 mean sea level pressure, where the latter corrected for water vapour pressure using Dickson et al. (2007). This results in monthly 1x1 degree pCO2atm. |
|---|---|---|---|---|---|---|---|---|
| Total ocean area on native grid (km2) | 3.63E+08 | 3.63E+08 | 3.50E+08 | 3.52E+08 | 3.49E+08 | 3.10E+08 (2.98E+08 to 3.16E+08, depending on ice cover) | 3.55E+08 | 3.61E+08 |
| method to extend product to full global ocean coverage | | Arctic and marginal seas added following Landschützer et al. (2020). No coastal cut. | | | | Fay & Gregor et al. 2021 | Method has near full coverage | Fay & Gregor et al. 2021. Gaps were filled with monthly climatology. Interannual variability was added to the climatology based on the temporal evolution of 5 products for years 1985 through 2020 and then only using this product for year 2021. |

**Table A4.** Comparison of the inversion set up and input fields for the atmospheric inversions. Atmospheric inversions see the full CO2 fluxes, including the anthropogenic and pre-industrial fluxes. Hence they need to be adjusted for the pre-industrial flux of CO2 from the land to the ocean that is part of the natural carbon cycle before they can be compared with SOCEAN and SLAND from process models. See Table 4 for references.

| | Copernicus Atmosphere Monitoring Service (CAMS) | Carbon-Tracker Europe (CTE) | Jena CarboScope | UoE | NISMON-CO2 | CMS-Flux | GONGGA | THU | Copernicus Atmosphere Monitoring Service (CAMS) Satellite |
|---|---|---|---|---|---|---|---|---|---|
| **Version number** | v21r1 | v2022 | v2022 | UoE v6.1b | v2022.1 | v2022 | v2022 | v2022 | FT21r2 |
| **Observations** | | | | | | | | | |
| **Atmospheric observations** | Hourly resolution (well-mixed conditions) obspack GLOBALVIEWplus v7.0 (a) and NRT_v7.2(b), WDCGG, RAMCES and ICOS ATC | Hourly resolution (well-mixed conditions) obspack GLOBALVIEWplus v7.0 (a) and NRT_v7.2(b) | Flasks and hourly from various institutions (outliers removed by 2σ criterion) | Hourly resolution (well-mixed conditions) obspack GLOBAL VIEWplus v7.0(a) and NRT_v7.2 (b) | Hourly resolution (well-mixed conditions) obspack GLOBALVIEWplus v7.0(a) and NRT_v7.2(b) | ACOS-GOSAT v9r, OCO-2 v10 scaled to WMO 2019 standard and remote flask observations from ObsPack, GLOBALVIEW puls, v7.0(a) and NRT_v 7.2(b) | OCO-2 v10r data that scaled to WMO 2019 standard | OCO-2 v10r data that scaled to WMO 2019 standard | bias-corrected ACOS GOSAT v9 over land until August 2024 + bias-corrected ACOS OCO-2 v10 over land, both rescaled to X2019 |
| **Period covered** | 1979-2021 | 2001-2021 | 1957-2021 | 2001-2021 | 1990-2021 | 2010-2021 | 2015-2021 | 2015-2021 | 2010-2021 |
| **Prior fluxes** | | | | | | | | | |
| **Biosphere and fires** | ORCHIDEE, GFEDv4.1s | SiB4 and GFAS | Zero | CASA v1.0, climatology after 2016 and GFED4.0 | VISIT and GFEDv4.1s | CARDAMOM | CASA and GFEDv4.1s | SiB4.2 and GFEDv4.1s | ORCHIDEE, GFEDv4.1s |
| **Ocean** | CMEMS-LSCE-FFNN 2021 | CarboScope v2021 | CarboScope v2022 | Takahashi climatology | JMA global ocean mapping (Iida et al., 2015) | MOM6 | Takahashi climatology | Takahashi climatology | CMEMS-LSCE-FFNN 2021 |
| **Fossil fuels** | GridFED 2021.2(c) with an extrapolation to 2021 based on Carbonmonitor and NO2 | GridFED 2021.3 + GridFED 2022.2 for 2021 (c) | GridFED v2022.2 (c) | GridFED 2022.1 (c) | GridFED v2022.2 (c) | GridFED2022.2 (c) | GridFED 2021.3 (c) with an extrapolation to 2021 based on Carbon-monitor | GridFED v2022.1 (c) | GridFED 2021.2 (c) with an extrapolation to 2021 based on Carbonmonitor and NO2 |
| **Transport and optimization** | | | | | | | | | |
| **Transport model** | LMDZ v6 | TM5 | TM3 | GEOS-CHEM | NICAM-TM | GEOS-CHEM | GEOS-Chem v12.9.3 | GEOS-CHEM | LMDZ v6 |





| Weather forcing | ECMWF | ECMWF | NCEP | MERRA | JRA55 | MERRA | MERRA2 | GEOS-FP | ECMWF |
|---|---|---|---|---|---|---|---|---|---|
| Horizontal Resolution | Global 3.75°x1.875° | Global 3°x2°, Europe 1°x1°, North America 1°x1° | Global 3.83°x5° | Global 4°x5° | Isocahedral grid: ~225km | Global 4°x5° | Global 2°x2.5° | Global 4°x5° | Global 3.75°x1.875° |
| Optimization | Variational | Ensemble Kalman filter | Conjugate gradient (re-ortho-normalization) (d) | Ensemble Kalman filter | Variational | Variational | Nonlinear least squares four-dimensional variation (NLS-4DVar) | Ensemble Kalman filter | Variational |

| |
|---|
| (a) https://doi.org/10.25925/20210801. Schuldt et al. Multi-laboratory compilation of atmospheric carbon dioxide data for the period 1957-2020; obspack_co2_1_GLOBALVIEWplus_v7.0_2021-08-18; NOAA Earth System Research Laboratory, Global Monitoring Laboratory. http://doi.org/10.25925/20210801. |
| (b) http://doi.org/10.25925/20220624. Schuldt et al. Multi-laboratory compilation of atmospheric carbon dioxide data for the period 2021-2022; obspack_co2_1_NRT_v7.2_2022-06-28; NOAA Earth System Research Laboratory, Global Monitoring Laboratory. http://doi.org/10.25925/20220624. |
| (c) GCP-GridFED v2021.2, v2021.3, v2022.1 and v2022.2 (Jones et al., 2022) are updates through the year 2021 of the GCP-GridFED dataset presented by Jones et al. (2021). |
| (d) ocean prior not optimised |



**Table A5 Attribution of fCO2 measurements for the year 2021 included in SOCATv2022 (Bakker et al., 2016, 2022) to inform ocean fCO2-based data products.**

| Platform Name | Regions | No. of measurements | Principal Investigators | No. of datasets | Platform Type |
|---|---|---|---|---|---|
| 1 degree | North Atlantic, coastal | 71,863 | Tanhua, T. | 1 | Ship |
| Alawai_158W_21N | Tropical Pacific | 387 | Sutton, A.; De Carlo, E. H.; Sabine, C. | 1 | Mooring |
| Atlantic Explorer | North Atlantic, tropical Atlantic, coastal | 34,399 | Bates, N. R. | 16 | Ship |
| Atlantic Sail | North Atlantic, coastal | 27,496 | Steinhoff, T.; Körtzinger, A. | 7 | Ship |
| BlueFin | Tropical Pacific | 60,606 | Alin, S. R.; Feely, R. A. | 11 | Ship |
| Cap San Lorenzo | North Atlantic, tropical Atlantic, coastal | 44,281 | Lefèvre, N. | 7 | Ship |
| CCE2_121W_34N | Coastal | 1,333 | Sutton, A.; Send, U.; Ohman, M. | 1 | Mooring |
| Celtic Explorer | North Atlantic, coastal | 61,118 | Cronin, M. | 10 | Ship |
| F.G. Walton Smith | Coastal | 38,375 | Rodriguez, C.; Millero, F. J.; Pierrot, D.; Wanninkhof, R. | 14 | Ship |
| Finnmaid | Coastal | 223,438 | Rehder, G.; Bittig, H. C.; Glockzin, M. | 1 | Ship |
| FRA56 | Coastal | 5,652 | Tanhua, T. | 1 | Ship |
| G.O. Sars | Arctic, north Atlantic, coastal | 82,607 | Skjelvan, I. | 9 | Ship |
| GAKOA_149W_60N | Coastal | 402 | Monacci, N.; Cross, J.; Musielewicz, S.; Sutton, A. | 1 | Mooring |
| Gordon Gunter | North Atlantic, coastal | 36,058 | Wanninkhof, R.; Pierrot, D. | 6 | Ship |
| Gulf Challenger | Coastal | 6,375 | Salisbury, J.; Vandemark, D.; Hunt, C. W. | 6 | Ship |
| Healy | Arctic, north Atlantic, coastal | 28,998 | Sweeney, C.; Newberger, T.; Sutherland, S. C.; Munro, D. R. | 5 | Ship |
| Henry B. Bigelow | North Atlantic, coastal | 67,399 | Wanninkhof, R.; Pierrot, D. | 8 | Ship |
| Heron Island | Coastal | 989 | Tilbrook, B.; Neill, C.; van Oojen, E.; Passmore, A.; Black, J. | 1 | Mooring |
| Investigator | Southern Ocean, coastal, tropical Pacific, Indian Ocean | 120,782 | Tilbrook, B.; Akl, J.; Neill, C. | 6 | Ship |
| KC_BUOY | Coastal | 2,860 | Evans, W.; Pocock, K. | 1 | Mooring |
| Keifu Maru II | North Pacific, tropical Pacific, coastal | 10,053 | Kadono, K. | 8 | Ship |
| Laurence M. Gould | Southern Ocean | 2,604 | Sweeney, C.; Newberger, T.; Sutherland, S. C.; Munro, D. R. | 1 | Ship |
| Marion Dufresne | Indian Ocean, Southern Ocean, coastal | 9,911 | Lo Monaco, C.; Metzl, N. | 1 | Ship |
| Nathaniel B. Palmer | Southern Ocean | 2,376 | Sweeney, C.; Newberger, T.; Sutherland, S. C.; Munro, D. R. | 1 | Ship |
| New Century 2 | North Pacific, tropical Pacific, north Atlantic, coastal | 198,293 | Nakaoka, S.-I.; Takao, S. | 10 | Ship |
| Newrest - Art and Fenetres | North Atlantic, tropical Atlantic, south Atlantic, coastal | 17,699 | Tanhua, T. | 2 | Ship |
| Quadra Island Field Station | Coastal | 81,201 | Evans, W.; Pocock, K. | 1 | Mooring |
| Ronald H. Brown | North Atlantic, coastal | 31,661 | Wanninkhof, R.; Pierrot, D. | 3 | Ship |
| Ryofu Maru III | North Pacific, tropical Pacific, coastal | 10,464 | Kadono, K. | 8 | Ship |
| Sea Explorer | Southern Ocean, north Atlantic, coastal, tropical Atlantic | 37,027 | Landschützer, P.; Tanhua, T. | 2 | Ship |

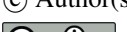



| | | | | | |
|---|---|---|---|---|---|
| Sikuliaq | Arctic, north Pacific, coastal | 60,549 | Sweeney, C.; Newberger, T.; Sutherland, S. C.; Munro, D. R. | 13 | Ship |
| Simon Stevin | Coastal | 57,055 | Gkritzalis, T.; Theetaert, H.; Cattrijsse, A.; T´Jampens, M. | 11 | Ship |
| Sitka Tribe of Alaska Environmental Research Laboratory | Coastal | 19,086 | Whitehead, C.; Evans, W.; Lanphier, K.; Peterson, W.; Kennedy, E.; Hales, B. | 1 | Mooring |
| SOFS_142E_46S | Southern Ocean | 894 | Sutton, A.; Trull, T.; Shadwick, E. | 1 | Mooring |
| Soyo Maru | Tropical Pacific, coastal | 33,234 | Ono, T. | 3 | Ship |
| Station M | North Atlantic | 447 | Skjelvan, I. | 1 | Mooring |
| Statsraad Lehmkuhl | North Atlantic, tropical Atlantic, coastal | 47,881 | Becker, M.; Olsen, A. | 3 | Ship |
| TAO125W_0N | Tropical Pacific | 241 | Sutton, A. | 1 | Mooring |
| Tavastland | Coastal | 48,421 | Willstrand Wranne, A.; Steinhoff, T. | 17 | Ship |
| Thomas G. Thompson | North Atlantic, tropical Atlantic, north Pacific, tropical Pacific, coastal | 47,073 | Alin, S. R. ; Feely, R. A. | 5 | Ship |
| Trans Future 5 | Southern Ocean, north Pacific, tropical Pacific, coastal | 257,424 | Nakaoka, S.-I.; Takao, S. | 22 | Ship |
| Tukuma Arctica | North Atlantic, coastal | 70,033 | Becker, M.; Olsen, A. | 23 | Ship |
| Wakataka Maru | North Pacific, coastal | 13,392 | Tadokoro, K. | 2 | Ship |

**Table A6.** Aircraft measurement programs archived by Cooperative Global Atmospheric Data Integration Project (CGADIP; Schuldt et al. 2022a and 2022b) that contribute to the evaluation of the atmospheric inversions (Figure B4).

| Site code | Measurement program name in Obspack | Specific doi | Data providers |
|---|---|---|---|
| AAO | Airborne Aerosol Observatory, Bondville, Illinois | | Sweeney, C.; Dlugokencky, E.J. |
| ABOVE | Carbon in Arctic Reservoirs Vulnerability Experiment (CARVE) | https://doi.org/10.3334/ORNLDAAC/1404 | Sweeney, C., J.B. Miller, A. Karion, S.J. Dinardo, and C.E. Miller. 2016. CARVE: L2 Atmospheric Gas Concentrations, Airborne Flasks, Alaska, 2012-2015. ORNL DAAC, Oak Ridge, Tennessee, USA. |
| ACG | Alaska Coast Guard | | Sweeney, C.; McKain, K.; Karion, A.; Dlugokencky, E.J. |
| ACT | Atmospheric Carbon and Transport - America | | Sweeney, C.; Dlugokencky, E.J.; Baier, B; Montzka, S.; Davis, K. |
| AIRCORENOAA | NOAA AirCore | | Colm Sweeney (NOAA) AND Bianca Baier (NOAA) |
| ALF | Alta Floresta | | Gatti, L.V.; Gloor, E.; Miller, J.B.; |
| AOA | Aircraft Observation of Atmospheric trace gases by JMA | | ghg_obs@met.kishou.go.jp |
| BGI | Bradgate, Iowa | | Sweeney, C.; Dlugokencky, E.J. |
| BNE | Beaver Crossing, Nebraska | | Sweeney, C.; Dlugokencky, E.J. |
| BRZ | Berezorechka, Russia | | Sasakama, N.; Machida, T. |
| CAR | Briggsdale, Colorado | | Sweeney, C.; Dlugokencky, E.J. |
| CMA | Cape May, New Jersey | | Sweeney, C.; Dlugokencky, E.J. |
| CON | CONTRAIL (Comprehensive Observation Network for TRace gases by AIrLiner) | http://dx.doi.org/10.17595/20180208.001 | Machida, T.; Matsueda, H.; Sawa, Y. Niwa, Y. |
| CRV | Carbon in Arctic Reservoirs Vulnerability Experiment (CARVE) | | Sweeney, C.; Karion, A.; Miller, J.B.; Miller, C.E.; Dlugokencky, E.J. |
| DND | Dahlen, North Dakota | | Sweeney, C.; Dlugokencky, E.J. |
| ECO | East Coast Outflow | | Sweeney, C.; McKain, K. |
| ESP | Estevan Point, British Columbia | | Sweeney, C.; Dlugokencky, E.J. |
| ETL | East Trout Lake, Saskatchewan | | Sweeney, C.; Dlugokencky, E.J. |
| FWI | Fairchild, Wisconsin | | Sweeney, C.; Dlugokencky, E.J. |
| GSFC | NASA Goddard Space Flight Center Aircraft Campaign | | Kawa, S.R.; Abshire, J.B.; Riris, H. |
| HAA | Molokai Island, Hawaii | | Sweeney, C.; Dlugokencky, E.J. |
| HFM | Harvard University Aircraft Campaign | | Wofsy, S.C. |
| HIL | Homer, Illinois | | Sweeney, C.; Dlugokencky, E.J. |
| HIP | HIPPO (HIAPER Pole-to-Pole Observations) | https://doi.org/10.3334/CDIAC/HIPPO_010 | Wofsy, S.C.; Stephens, B.B.; Elkins, J.W.; Hintsa, E.J.; Moore, F. |
| IAGOS-CARIBI | In-service Aircraft for a Global Observing System | | Obersteiner, F.; Boenisch., H; Gehrlein, T.; Zahn, A.; Schuck, T. |



| C | | | |
|---|---|---|---|
| INX | INFLUX (Indianapolis Flux Experiment) | | Sweeney, C.; Dlugokencky, E.J.; Shepson, P.B.; Turnbull, J. |
| LEF | Park Falls, Wisconsin | | Sweeney, C.; Dlugokencky, E.J. |
| NHA | Offshore Portsmouth, New Hampshire (Isles of Shoals) | | Sweeney, C.; Dlugokencky, E.J. |
| OIL | Oglesby, Illinois | | Sweeney, C.; Dlugokencky, E.J. |
| ORC | ORCAS (O2/N2 Ratio and CO2 Airborne Southern Ocean Study) | https://doi.org/10.5065/D6S B445X | Stephens, B.B, Sweeney, C., McKain, K., Kort, E. |
| PFA | Poker Flat, Alaska | | Sweeney, C.; Dlugokencky, E.J. |
| RBA-B | Rio Branco | | Gatti, L.V.; Gloor, E.; Miller, J.B. |
| RTA | Rarotonga | | Sweeney, C.; Dlugokencky, E.J. |
| SCA | Charleston, South Carolina | | Sweeney, C.; Dlugokencky, E.J. |
| SGP | Southern Great Plains, Oklahoma | | Sweeney, C.; Dlugokencky, E.J.; Biraud, S. |
| TAB | Tabatinga | | Gatti, L.V.; Gloor, E.; Miller, J.B. |
| TGC | Offshore Corpus Christi, Texas | | Sweeney, C.; Dlugokencky, E.J. |
| THD | Trinidad Head, California | | Sweeney, C.; Dlugokencky, E.J. |
| WBI | West Branch, Iowa | | Sweeney, C.; Dlugokencky, E.J. |



**Table A7.** Main methodological changes in the global carbon budget since first publication. Methodological changes introduced in one year are kept for the following years unless noted. Empty cells mean there were no methodological changes introduced that year.

| Publication year | Fossil fuel emissions | | | LUC emissions | Reservoirs | | | Uncertainty & other changes |
|---|---|---|---|---|---|---|---|---|
| | Global | Country (territorial) | Country (consumption) | | Atmosphere | Ocean | Land | |
| 2006 (a) | | Split in regions | | | | | | |
| 2007 (b) | | | | ELUC based on FAO-FRA 2005; constant ELUC for 2006 | 1959-1979 data from Mauna Loa; data after 1980 from global average | Based on one ocean model tuned to reproduced observed 1990s sink | | ±1σ provided for all components |
| 2008 (c) | | | | Constant ELUC for 2007 | | | | |
| 2009 (d) | | Split between Annex B and non-Annex B | Results from an independent study discussed | Fire-based emission anomalies used for 2006-2008 | | Based on four ocean models normalised to observations with constant delta | First use of five DGVMs to compare with budget residual | |
| 2010 (e) | Projection for current year based on GDP | Emissions for top emitters | | ELUC updated with FAO-FRA 2010 | | | | |
| 2011 (f) | | | Split between Annex B and non-Annex B | | | | | |
| 2012 (g) | | 129 countries from 1959 | 129 countries and regions from 1990-2010 based on GTAP8.0 | ELUC for 1997-2011 includes interannual anomalies from fire-based emissions | All years from global average | Based on 5 ocean models normalised to observations with ratio | Ten DGVMs available for SLAND; First use of four models to compare with ELUC | |
| 2013 (h) | | 250 countriesb | 134 countries and regions 1990-2011 based on GTAP8.1, with detailed estimates for years 1997, 2001, 2004, and 2007 | ELUC for 2012 estimated from 2001-2010 average | | Based on six models compared with two data-products to year 2011 | Coordinated DGVM experiments for SLAND and ELUC | Confidence levels; cumulative emissions; budget from 1750 |





| | | | | | | | |
|---|---|---|---|---|---|---|---|
| 2014 (i) | Three years of BP data | Three years of BP data | Extended to 2012 with updated GDP data | ELUC for 1997-2013 includes interannual anomalies from fire-based emissions | | Based on seven models | Based on ten models | Inclusion of breakdown of the sinks in three latitude bands and comparison with three atmospheric inversions |
| 2015 (j) | Projection for current year based Jan-Aug data | National emissions from UNFCCC extended to 2014 also provided | Detailed estimates introduced for 2011 based on GTAP9 | | | Based on eight models | Based on ten models with assessment of minimum realism | The decadal uncertainty for the DGVM ensemble mean now uses ±1σ of the decadal spread across models |
| 2016 (k) | Two years of BP data | Added three small countries; China's emissions from 1990 from BP data (this release only) | | Preliminary ELUC using FRA-2015 shown for comparison; use of five DGVMs | | Based on seven models | Based on fourteen models | Discussion of projection for full budget for current year |
| 2017 (l) | Projection includes India-specific data | | | Average of two bookkeeping models; use of 12 DGVMs | | Based on eight models that match the observed sink for the 1990s; no longer normalised | Based on 15 models that meet observation-based criteria (see Sect. 2.5) | Land multi-model average now used in main carbon budget, with the carbon imbalance presented separately; new table of key uncertainties |

a Raupach et al. (2007)

b Canadell et al. (2007)

c GCP (2008)

d Le Quéré et al. (2009)

e Friedlingstein et al. (2010)

f Peters et al. (2012b)

g Le Quéré et al. (2013), Peters et al. (2013)

h Le Quéré et al. (2014)

i Le Quéré et al. (2015a)

j Le Quéré et al. (2015b)

k Le Quéré et al. (2016)

l Le Quéré et al. (2018a)





**Table A8: Mapping of global carbon cycle models' land flux definitions to the definition of the LULUCF net flux used in national reporting to UNFCCC. Non-intact lands are used here as proxy for "managed lands" in the country reporting, national Greenhouse Gas Inventories (NGHGI) are gap-filled (see Sec. C.2.3 for details). Where available, we provide independent estimates of certain fluxes for comparison.**

| | | | 2002-2011 | 2012-2021 |
|---|---|---|---|---|
| ELUC from bookkeeping estimates (from Tab. 5) | | | 1.36 | 1.24 |
| SLAND | Total (from Tab. 5) | from DGVMs | -2.85 | -3.10 |
| | in non-forest lands | from DGVMs | -0.74 | -0.83 |
| | in non-intact forest | from DGVMs | -1.67 | -1.80 |
| | in intact forests | from DGVMs | -0.44 | -0.47 |
| | in intact land | from ORCHIDEE-MICT | -1.34 | -1.38 |
| ELUC plus SLAND on non-intact lands | considering non-intact forests only | from bookkeeping ELUC and DGVMs | -0.31 | -0.56 |
| | considering all non-intact land | from ORCHIDEE-MICT | 0.90 | 0.60 |
| National Greenhouse Gas Inventories (LULUCF) | | | -0.37 | -0.54 |
| FAOSTAT (LULUCF) | | | 0.39 | 0.24 |



| Table A9. Funding supporting the production of the various components of the global carbon budget in addition to the authors' supporting institutions (see also acknowledgements). | |
|---|---|
| **Funder and grant number (where relevant)** | **Author Initials** |
| Australia, Integrated Marine Observing System (IMOS) | BT |
| Australian National Environment Science Program (NESP) | JGC |
| Belgium, FWO (Flanders Research Foundation, contract GN I001821N) | TGk |
| BNP Paribas Foundation through Climate & Biodiversity initiative, philanthropic grant for developments of the Global Carbon Atlas | PC |
| Canada, Tula Foundation | WE, KP |
| China, National Natural Science Foundation (grant no. 41975155) | XY |
| China, National Natural Science Foundation (grant no. 42141020) | WY |
| China, National Natural Science Foundation of China (grant no. 41921005) | BZ |
| China, Scientific Research Start-up Funds (grant no. QD2021024C) from Tsinghua Shenzhen International Graduate School | BZ |
| China, Second Tibetan Plateau Scientific Expedition and Research Program (SQ2022QZKK0101) | XT |
| China, Young Elite Scientists Sponsorship Program by CAST (grant no. YESS20200135) | BZ |
| EC Copernicus Atmosphere Monitoring Service implemented by ECMWF | FC |
| EC Copernicus Marine Environment Monitoring Service implemented by Mercator Ocean | MG |
| EC H2020 (4C; grant no 821003) | PF, MOS, RMA, SS, GPP, PC, JIK, TI, LB, AJ, PL, LGr, NG, NMa, SZ |
| EC H2020 (CoCO2: grant no. 958927) | RMA, GPP, JIK |
| EC H2020 (COMFORT: grant no. 820989) | LGr, MG, NG |
| EC H2020 (CONSTRAIN: grant no 820829) | RS, TGa |
| EC H2020 (ESM2025 – Earth System Models for the Future; grant agreement No 101003536). | RS, TGa, TI, LB, BD |
| EC H2020 (JERICO-S3: grant no. 871153) | HCB |
| EC H2020 (VERIFY: grant no. 776810) | MWJ, RMA, GPP, PC, JIK, MJM |
| Efg International | TT, MG |
| European Space Agency Climate Change Initiative ESA-CCI RECCAP2 project 655 (ESRIN/4000123002/18/I-NB) | SS, PC |
| European Space Agency OceanSODA project (grant no. 4000137603/22/I-DT) | LGr, NG |
| France, French Oceanographic Fleet (FOF) | NMe |
| France, ICOS (Integrated Carbon Observation System) France | NL |
| France, Institut National des Sciences de l'Univers (INSU) | NMe |
| France, Institut polaire français Paul-Emile Victor(IPEV) | NMe |
| France, Institut de recherche français sur les ressources marines (IFREMER) | NMe |
| France, Institut de Recherche pour le Développement (IRD) | NL |
| France, Observatoire des sciences de l'univers Ecce-Terra (OSU at | NMe |



| | |
|---|---|
| Sorbonne Université) | |
| Germany, Deutsche Forschungsgemeinschaft (DFG) under Germany's Excellence Strategy – EXC 2037 'Climate, Climatic Change, and Society' – Project Number: 390683824 | TI |
| Germany, Federal Ministry for Education and Research (BMBF) | HCB |
| Germany, German Federal Ministry of Education and Research under project "DArgo2025" (03F0857C) | TS |
| Germany, Helmholtz Association ATMO programme | AA |
| Germany, Helmholtz Young Investigator Group Marine Carbon and Ecosystem Feedbacks in the Earth System (MarESys), grant number VH-NG-1301 | JH, OG |
| Germany, ICOS (Integrated Carbon Observation System) Germany | HCB |
| Hapag-Lloyd | TT, MG |
| Ireland, Marine Institute | MC |
| Japan, Environment Research and Technology Development Fund of the Ministry of the Environment (JPMEERF21S20810) | YN |
| Japan, Global Environmental Research Coordination System, Ministry of the Environment (grant number E1751) | SN, ST, TO |
| Japan, Environment Research and Technology Development Fund of the Ministry of the Environment (JPMEERF21S20800) | HT |
| Japan, Japan Meteorological Agency | KK |
| Kuehne + Nagel International AG | TT, MG |
| Mediterranean Shipping Company (MSc) | TT, MG |
| Monaco, Fondation Prince Albert II de Monaco | TT, MG |
| Monaco, Yacht Club de Monaco | TT, MG |
| Netherlands, ICOS (Integrated Carbon Observation System) | WP |
| Norway, Research Council of Norway (N-ICOS-2, grant no. 296012) | AO, MB, IS |
| Norway, Norwegian Research Council (grant no. 270061) | JS |
| Sweden, ICOS (Integrated Carbon Observation System) | AW |
| Sweden, Swedish Meteorological and Hydrological Institute | AW |
| Sweden, The Swedish Research Council | AW |
| Swiss National Science Foundation (grant no. 200020-200511) | QS |
| Tibet, Second Tibetan Plateau Scientific Expedition and Research Program (SQ2022QZKK0101) | TX |
| UK Royal Society (grant no. RP\R1\191063) | CLQ |
| UK, Natural Environment Research Council (SONATA: grant no. NE/P021417/1) | RW |
| UK, Natural Environmental Research Council (NE/R016518/1) | PIP |
| UK, Natural Environment Research Council (NE/V01417X/1) | MWJ |
| UK, Royal Society: The European Space Agency OCEANFLUX projects | JDS |
| UK Royal Society (grant no. RP\R1\191063) | CLQ |
| USA, BIA Tribal Resilience | CW |
| USA, Cooperative Institute for Modeling the Earth System between the National Oceanic and Atmospheric Administration Geophysical | LR |

| | |
|---|---|
| Fluid Dynamics Laboratory and Princeton University and the High Meadows Environmental Institute | |
| USA, Cooperative Institute for Climate, Ocean, & Ecosystem Studies (CIOCES) under NOAA Cooperative Agreement NA20OAR4320271 | KO |
| USA, Department of Energy, Biological and Evironmental Research | APW |
| USA, Department of Energy, SciDac (DESC0012972) | GCH, LPC |
| USA, Energy Exascale Earth System Model (E3SM) project, Department of Energy, Office of Science, Office of Biological and Environmental Research | GCH, LPC |
| USA, EPA Indian General Assistance Program | CW |
| USA, NASA Carbon Monitoring System probram and OCO Science team program (80NM0018F0583) . | JL |
| USA, NASA Interdisciplinary Research in Earth Science (IDS) (80NSSC17K0348) | GCH, LPC, BP |
| USA, National Center for Atmospheric Research (NSF Cooperative Agreement No. 1852977) | DK |
| USA, National Oceanic and Atmospheric Administration, Ocean Acidification Program | DP, RW, SRA, RAF, AJS, NMM |
| USA, National Oceanic and Atmospheric Administration, Global Ocean Monitoring and Observing Program | DRM, CSw, NRB, CRodr, DP, RW, SRA, RAF, AJS |
| USA, National Science Foundation (grant number 1903722) | HT |
| USA, State of Alaska | NMM |
| **Computing resources** | |
| ADA HPC cluster at the University of East Anglia | MWJ |
| CAMS inversion was granted access to the HPC resources of TGCC under the allocation A0110102201 | FC |
| Cheyenne supercomputer (doi:10.5065/D6RX99HX), were provided by the Computational and Information Systems Laboratory (CISL) at NCAR | DK |
| HPC cluster Aether at the University of Bremen, financed by DFG within the scope of the Excellence Initiative | ITL |
| MRI (FUJITSU Server PRIMERGY CX2550M5) | YN |
| NIES (SX-Aurora) | YN |
| NIES supercomputer system | EK |
| UNINETT Sigma2, National Infrastructure for High Performance Computing and Data Storage in Norway (NN2980K/NS2980K) | JS |

**Appendix B. Supplementary Figures**

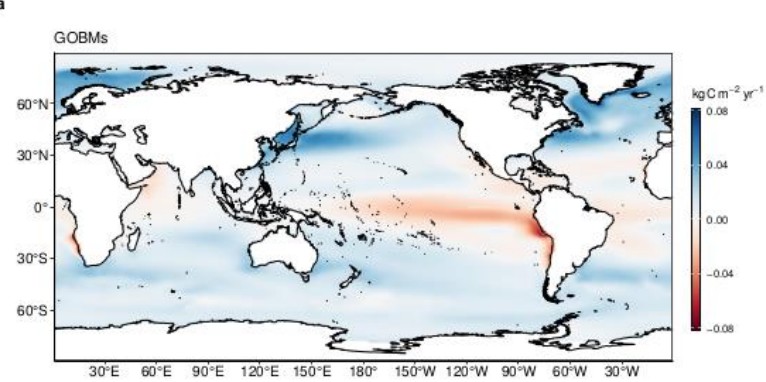

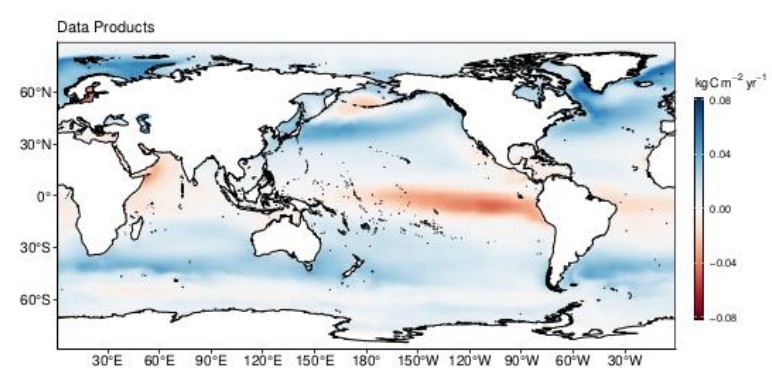

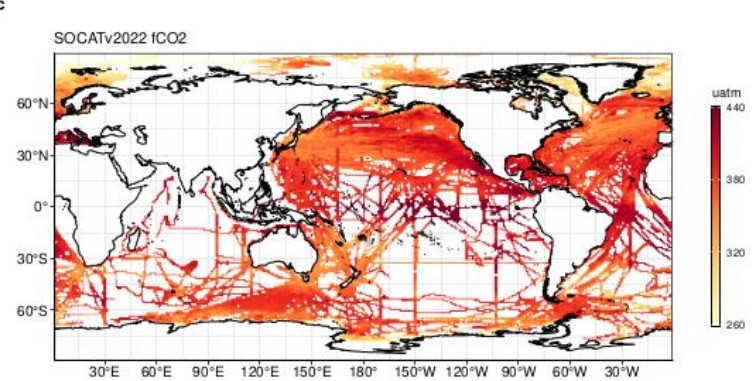

**Figure B1. Ensemble mean air-sea $CO_2$ flux from a) global ocean biogeochemistry models and b) $fCO_2$ based data products,**
**averaged over 2012-2021 period (kgC m$^{-2}$ yr$^{-1}$). Positive numbers indicate a flux into the ocean. c) gridded SOCAT v2022**
**$fCO_2$ measurements, averaged over the 2012-2021 period (µatm). In (a) model simulation A is shown. The data-products**



represent the contemporary flux, i.e. including outgassing of riverine carbon, which is estimated to amount to 0.65 GtC yr$^{-1}$
globally.



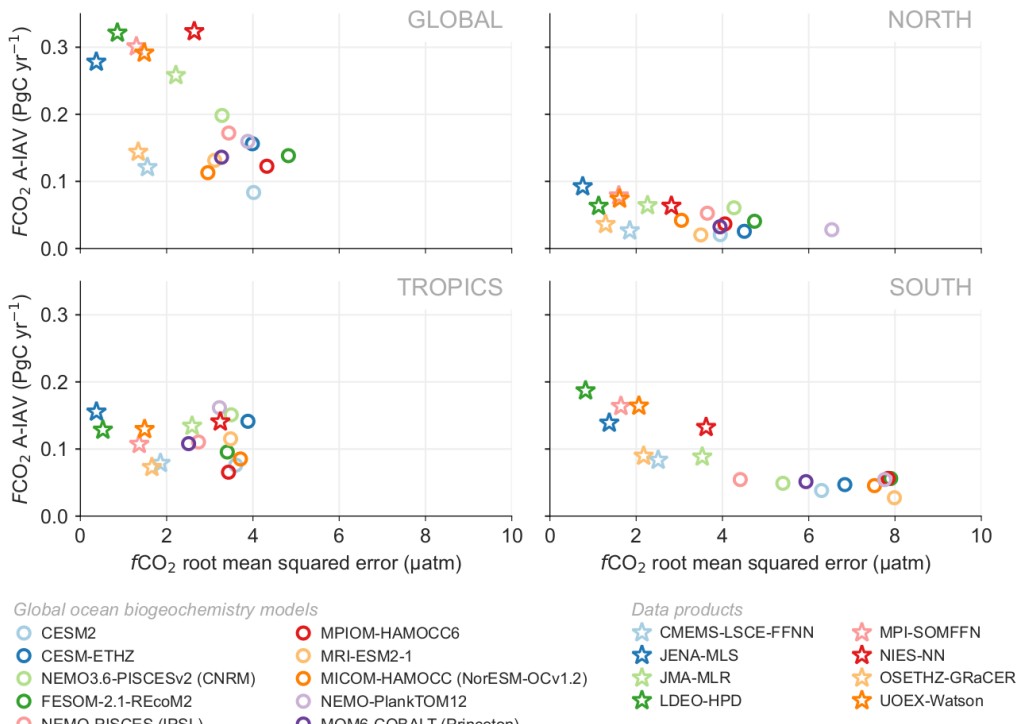

Figure B2. Evaluation of the GOBMs and data products using the root mean squared error (RMSE) for the period 1990 to 2021,
between the individual surface ocean $fCO_2$ mapping schemes and the SOCAT v2022 database. The y-axis shows the amplitude of
the interannual variability of the air-sea $CO_2$ flux (A-IAV, taken as the standard deviation of the detrended annual time series.
Results are presented for the globe, north (>30°N), tropics (30°S-30°N), and south (<30°S) for the GOBMs (see legend, circles) and
for the $fCO_2$-based data products (star symbols). The $fCO_2$-based data products use the SOCAT database and therefore are not
independent from the data (see section 2.4.1).



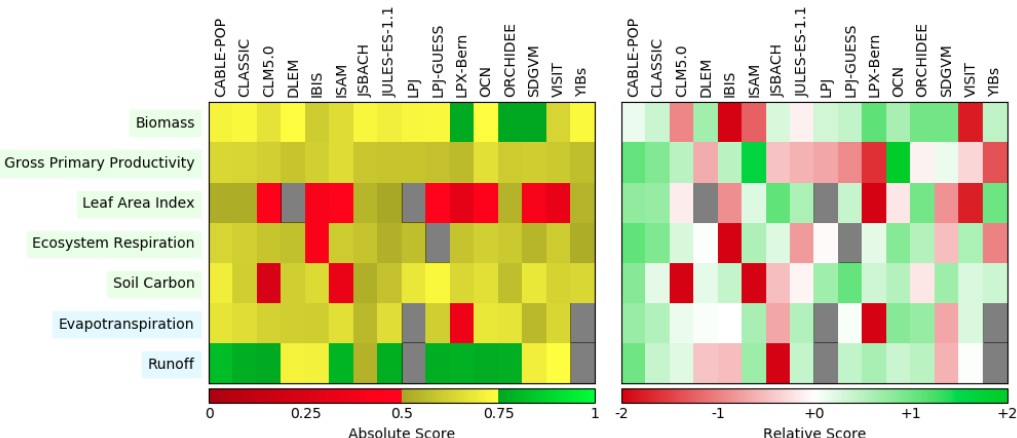

**Figure B3. Evaluation of the DGVMs using the International Land Model Benchmarking system (ILAMB; Collier et al., 2018) (left) absolute skill scores and (right) skill scores relative to other models. The benchmarking is done with observations for vegetation biomass (Saatchi et al., 2011; and GlobalCarbon unpublished data; Avitabile et al., 2016), GPP (Jung et al., 2010; Lasslop et al., 2010), leaf area index (De Kauwe et al., 2011; Myneni et al., 1997), ecosystem respiration (Jung et al., 2010; Lasslop et al., 2010), soil carbon (Hugelius et al., 2013;Todd-Brown et al., 2013), evapotranspiration (De Kauwe et al., 2011), and runoff (Dai and Trenberth, 2002). For each model-observation comparison a series of error metrics are calculated, scores are then calculated as an exponential function of each error metric, finally for each variable the multiple scores from different metrics and observational data sets are combined to give the overall variable scores shown in the left panel. Overall variable scores increase from 0 to 1 with improvements in model performance. The set of error metrics vary with data set and can include metrics based on the period mean, bias, root mean squared error, spatial distribution, interannual variability and seasonal cycle. The relative skill score shown in the right panel is a Z-score, which indicates in units of standard deviation the model scores relative to the multi-model mean score for a given variable. Grey boxes represent missing model data.**

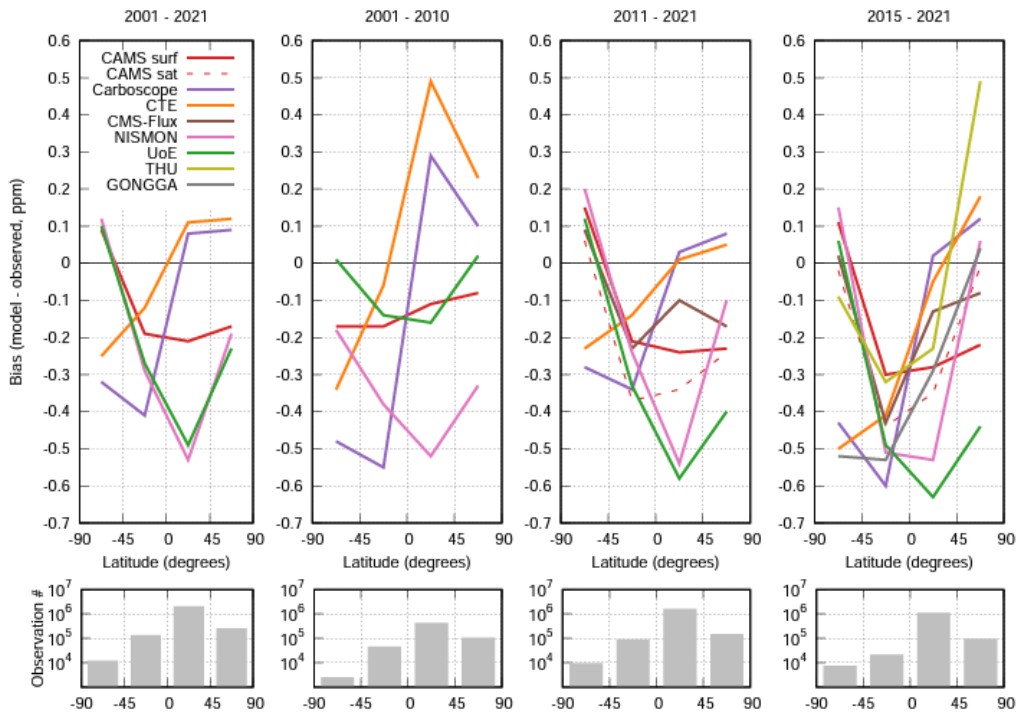

**Figure B4. Evaluation of the atmospheric inversion products. The mean of the model minus observations is shown for four latitude bands in four periods: (first panel) 2001-2021, (second panel) 2001-2010, (third panel) 2011-2021, (fourth panel) 2015-2021. The 9 systems are compared to independent CO2 measurements made onboard aircraft over many places of the world between 2 and 7 km above sea level. Aircraft measurements archived in the Cooperative Global Atmospheric Data Integration Project (Schuldt et al. 2021, Schuldt et al. 2022) from sites, campaigns or programs that have not been assimilated and cover at least 9 months (except for SH programs) between 2001 and 2021, have been used to compute the biases of the differences in four 45° latitude bins. Land and ocean data are used without distinction, and observation density varies strongly with latitude and time as seen on the lower panels.**

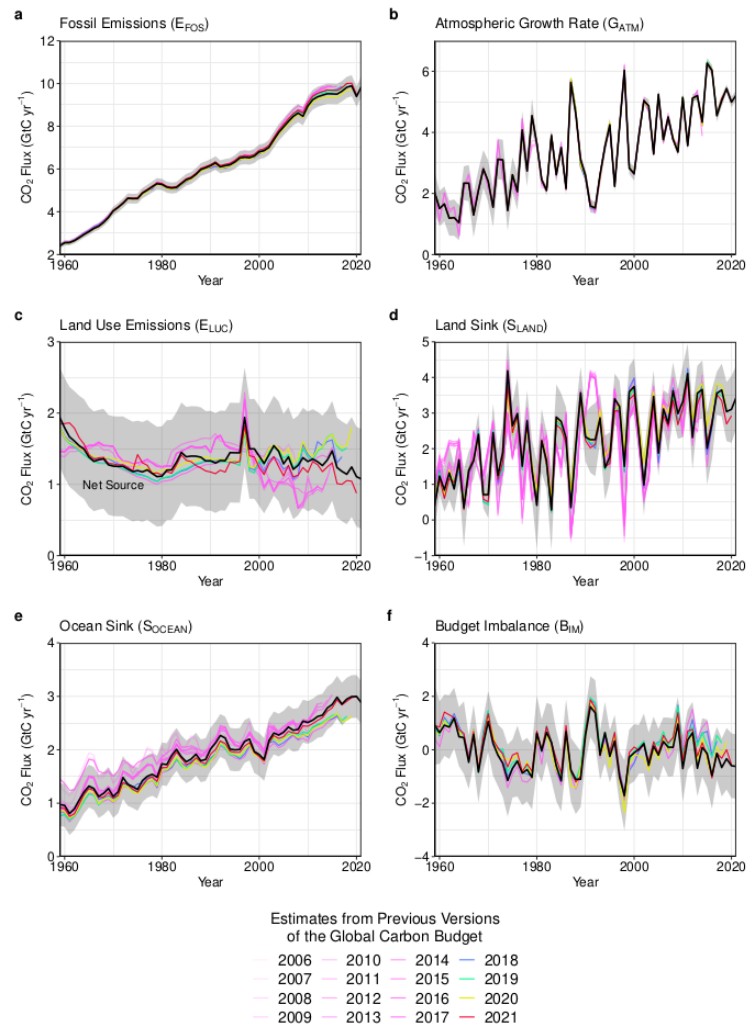

**Figure B5. Comparison of the estimates of each component of the global carbon budget in this study (black line) with the estimates released annually by the GCP since 2006. Grey shading shows the uncertainty bounds representing ±1 standard deviation of the current global carbon budget, based on the uncertainty assessments described in Appendix C. $CO_2$ emissions from (a) fossil $CO_2$ emissions ($E_{FOS}$), and (b) land-use change ($E_{LUC}$), as well as their partitioning among (c) the atmosphere ($G_{ATM}$), (d) the land ($S_{LAND}$), and (e) the ocean ($S_{OCEAN}$). See legend for the corresponding years, and Tables 3 and A7 for references. The budget year corresponds to the year when the budget was first released. All values are in GtC yr$^{-1}$.**

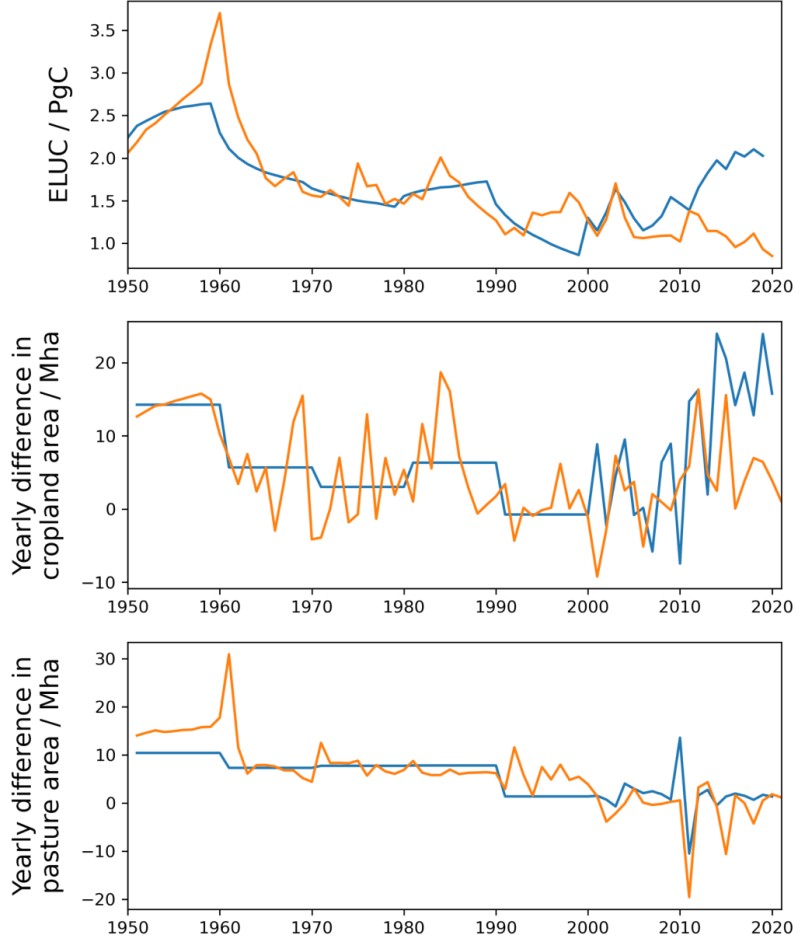

**Figure B6. Differences in the HYDE/LUH2 land-use forcing used for the global carbon budgets GCB2020 (Friedlingstein et**
**al., 2021), and for GCB2021/GCB2022 (Friedlingstein et al., 2022a, Friedlingstein et al., 2022b). Shown are year-to-year**
**changes in cropland area (middle panel) and pasture area (bottom panel). To illustrate the relevance of the update in the**
**land-use forcing to the recent trends in $E_{LUC}$, the top panel shows the land-use emission estimate from the bookkeeping model**
**BLUE (original model output, i.e. excluding peat fire and drainage emissions).**



**Appendix C. Extended Methodology**
**C.1 Methodology Fossil Fuel $CO_2$ emissions ($E_{FOS}$)**
**C.1.1 Cement carbonation**
From the moment it is created, cement begins to absorb $CO_2$ from the atmosphere, a process known as 'cement
carbonation'. We estimate this $CO_2$ sink, as the average of two studies in the literature (Cao et al., 2020; Guo et al.,
2021). Both studies use the same model, developed by Xi et al. (2016), with different parameterisations and input data,
with the estimate of Guo and colleagues being a revision of Xi et al (2016). The trends of the two studies are very
similar. Modelling cement carbonation requires estimation of a large number of parameters, including the different
types of cement material in different countries, the lifetime of the structures before demolition, of cement waste after
demolition, and the volumetric properties of structures, among others (Xi et al., 2016). Lifetime is an important
parameter because demolition results in the exposure of new surfaces to the carbonation process. The main reasons for
differences between the two studies appear to be the assumed lifetimes of cement structures and the geographic
resolution, but the uncertainty bounds of the two studies overlap. In the present budget, we include the cement
carbonation carbon sink in the fossil $CO_2$ emission component ($E_{FOS}$).
**C.1.2 Emissions embodied in goods and services**
CDIAC, UNFCCC, and BP national emission statistics 'include greenhouse gas emissions and removals taking place
within national territory and offshore areas over which the country has jurisdiction' (Rypdal et al., 2006), and are called
territorial emission inventories. Consumption-based emission inventories allocate emissions to products that are
consumed within a country, and are conceptually calculated as the territorial emissions minus the 'embodied' territorial
emissions to produce exported products plus the emissions in other countries to produce imported products
(Consumption = Territorial – Exports + Imports). Consumption-based emission attribution results (e.g. Davis and
Caldeira, 2010) provide additional information to territorial-based emissions that can be used to understand emission
drivers (Hertwich and Peters, 2009) and quantify emission transfers by the trade of products between countries (Peters
et al., 2011b). The consumption-based emissions have the same global total, but reflect the trade-driven movement of
emissions across the Earth's surface in response to human activities. We estimate consumption-based emissions from
1990-2020 by enumerating the global supply chain using a global model of the economic relationships between
economic sectors within and between every country (Andrew and Peters, 2013; Peters et al., 2011a). Our analysis is
based on the economic and trade data from the Global Trade and Analysis Project (GTAP; Narayanan et al., 2015), and
we make detailed estimates for the years 1997 (GTAP version 5), 2001 (GTAP6), and 2004, 2007, 2011, and 2014
(GTAP10.0a), covering 57 sectors and 141 countries and regions. The detailed results are then extended into an annual
time series from 1990 to the latest year of the Gross Domestic Product (GDP) data (2020 in this budget), using GDP
data by expenditure in current exchange rate of US dollars (USD; from the UN National Accounts main Aggregates
database; UN, 2021) and time series of trade data from GTAP (based on the methodology in Peters et al., 2011a). We
estimate the sector-level $CO_2$ emissions using the GTAP data and methodology, add the flaring and cement emissions
from our fossil $CO_2$ dataset, and then scale the national totals (excluding bunker fuels) to match the emission estimates
from the carbon budget. We do not provide a separate uncertainty estimate for the consumption-based emissions, but



based on model comparisons and sensitivity analysis, they are unlikely to be significantly different than for the
territorial emission estimates (Peters et al., 2012a).
**C.1.3 Uncertainty assessment for E$_{FOS}$**
We estimate the uncertainty of the global fossil CO2 emissions at ±5% (scaled down from the published ±10 % at ±2σ
to the use of ±1σ bounds reported here; Andres et al., 2012). This is consistent with a more detailed analysis of
uncertainty of ±8.4% at ±2σ (Andres et al., 2014) and at the high-end of the range of ±5-10% at ±2σ reported by
(Ballantyne et al., 2015). This includes an assessment of uncertainties in the amounts of fuel consumed, the carbon and
heat contents of fuels, and the combustion efficiency. While we consider a fixed uncertainty of ±5% for all years, the
uncertainty as a percentage of emissions is growing with time because of the larger share of global emissions from
emerging economies and developing countries (Marland et al., 2009). Generally, emissions from mature economies
with good statistical processes have an uncertainty of only a few per cent (Marland, 2008), while emissions from
strongly developing economies such as China have uncertainties of around ±10% (for ±1σ; Gregg et al., 2008; Andres
et al., 2014). Uncertainties of emissions are likely to be mainly systematic errors related to underlying biases of energy
statistics and to the accounting method used by each country.
**C.1.4 Growth rate in emissions**
We report the annual growth rate in emissions for adjacent years (in percent per year) by calculating the difference
between the two years and then normalising to the emissions in the first year: (EFOS(t0+1)-
EFOS(t0))/EFOS(t0)×100%. We apply a leap-year adjustment where relevant to ensure valid interpretations of annual
growth rates. This affects the growth rate by about 0.3% yr-1 (1/366) and causes calculated growth rates to go up
approximately 0.3% if the first year is a leap year and down 0.3% if the second year is a leap year.
The relative growth rate of $E_{FOS}$ over time periods of greater than one year can be rewritten using its logarithm
equivalent as follows:
$$\frac{1}{E_{FOS}}\frac{dE_{FOS}}{dt} = \frac{d(lnE_{FOS})}{dt}$$   (2)
Here we calculate relative growth rates in emissions for multi-year periods (e.g. a decade) by fitting a linear trend to
$ln(E_{FOS})$ in Eq. (2), reported in percent per year.
**C.1.5 Emissions projection for 2022**
To gain insight on emission trends for 2022, we provide an assessment of global fossil CO$_2$ emissions, $E_{FOS}$, by
combining individual assessments of emissions for China, USA, the EU, and India (the four countries/regions with the
largest emissions), and the rest of the world.
The methods are specific to each country or region, as described in detail below.
**China**: We use a regression between monthly data for each fossil fuel and cement, and annual data for consumption of
fossil fuels / production of cement to project full-year growth in fossil fuel consumption and cement production. The
monthly data for each product consists of the following:

125        ● Coal: Proprietary estimate for monthly consumption of main coal types, from SX Coal



● Oil: Production data from the National Bureau of Statistics (NBS), plus net imports from the China Customs
127        Administration (i.e., gross supply of oil, not including inventory changes)
● Natural gas: Same as for oil
● Cement: Production data from NBS
For oil, we use data for production and net imports of refined oil products rather than crude oil. This choice is made
because refined products are one step closer to actual consumption, and because crude oil can be subject to large
market-driven and strategic inventory changes that are not captured by available monthly data.
For each fuel and cement, we make a Bayesian linear regression between year-on-year cumulative growth in supply
(production for cement) and full-year growth in consumption (production for cement) from annual consumption data. In
the regression model, the growth rate in annual consumption (production for cement) is modelled as a regression
parameter multiplied by the cumulative year-on-year growth rate from the monthly data through July of each year for
past years (through 2021). We use broad Gaussian distributions centered around 1 as priors for the ratios between
annual and through-July growth rates. We then use the posteriors for the growth rates together with cumulative monthly
supply/production data through July of 2022 to produce a posterior predictive distribution for the full-year growth rate
for fossil fuel consumption / cement production in 2022.
If the growth in supply/production through July were an unbiased estimate of the full-year growth in
consumption/production, the posterior distribution for the ratio between the monthly and annual growth rates would be
centered around 1. However, in practice the ratios are different from 1 (in most cases below 1). This is a result of
various biasing factors such as uneven evolution in the first and second half of each year, inventory changes that are
somewhat anti-correlated with production and net imports, differences in statistical coverage, and other factors that are
not captured in the monthly data.
For fossil fuels, the mean of the posterior distribution is used as the central estimate for the growth rate in 2022, while
the edges of a 68% credible interval (analogous to a 1-sigma confidence interval) are used for the upper and lower
bounds.
For cement, the evolution from January to July has been highly atypical owing to the ongoing turmoil in the
construction sector, and the results of the regression analysis are heavily biased by equally atypical but different
dynamics in 2021. For this reason, we use an average of the results of the regression analysis and the plain growth in
cement production through July 2022, since this results in a growth rate that seems more plausible and in line with
where the cumulative cement production appears to be headed at the time of writing.
**USA**: We use emissions estimated by the U.S. Energy Information Administration (EIA) in their Short-Term Energy
Outlook (STEO) for emissions from fossil fuels to get both YTD and a full year projection (EIA, 2022). The STEO also
includes a near-term forecast based on an energy forecasting model which is updated monthly (last update with
preliminary data through August 2022), and takes into account expected temperatures, household expenditures by fuel
type, energy markets, policies, and other effects. We combine this with our estimate of emissions from cement
production using the monthly U.S. cement clinker production data from USGS for January-June 2022, assuming
changes in cement production over the first part of the year apply throughout the year.





**India**: We use monthly emissions estimates for India updated from Andrew (2020b) through July 2022. These
estimates are derived from many official monthly energy and other activity data sources to produce direct estimates of
national $CO_2$ emissions, without the use of proxies. Emissions from coal are then extended to August using a regression
relationship based on power generated from coal, coal dispatches by Coal India Ltd., the composite PMI, time, and days
per month. For the last 3-5 months of the year, each series is extrapolated assuming typical trends.
**EU**: We use a refinement to the methods presented by Andrew (2021), deriving emissions from monthly energy data
reported by Eurostat. Some data gaps are filled using data from the Joint Organisations Data Initiative (JODI, 2022).
Sub-annual cement production data are limited, but data for Germany and Poland, the two largest producers, suggest a
small decline. For fossil fuels this provides estimates through July. We extend coal emissions through August using a
regression model built from generation of power from hard coal, power from brown coal, total power generation, and
the number of working days in Germany and Poland, the two biggest coal consumers in the EU. These are then
extended through the end of the year assuming typical trends. We extend oil emissions by building a regression model
between our monthly $CO_2$ estimates and oil consumption reported by the EIA for Europe in its Short-Term Energy
Outlook (September edition), and then using this model with EIA's monthly forecasts. For natural gas, the strong
seasonal signal allows the use of the bias-adjusted Holt-Winters exponential smoothing method (Chatfield, 1978).
**Rest of the world**: We use the close relationship between the growth in GDP and the growth in emissions (Raupach et
al., 2007) to project emissions for the current year. This is based on a simplified Kaya Identity, whereby $E_{FOS}$ (GtC yr$^{-1}$)
is decomposed by the product of GDP (USD yr$^{-1}$) and the fossil fuel carbon intensity of the economy ($I_{FOS}$; GtC USD$^{-1}$)
as follows:
$$E_{FOS} = GDP \times I_{FOS} \tag{3}$$
Taking a time derivative of Equation (3) and rearranging gives:
$$\frac{1}{E_{FOS}}\frac{dE_{FOS}}{dt} = \frac{1}{GDP}\frac{dGDP}{dt} + \frac{1}{I_{FOS}}\frac{dI_{FOS}}{dt} \tag{4}$$
where the left-hand term is the relative growth rate of $E_{FOS}$, and the right-hand terms are the relative growth rates of
GDP and $I_{FOS}$, respectively, which can simply be added linearly to give the overall growth rate.
The $I_{FOS}$ is based on GDP in constant PPP (Purchasing Power Parity) from the International Energy Agency (IEA) up to
2017 (IEA/OECD, 2019) and extended using the International Monetary Fund (IMF) growth rates through 2021 (IMF,
2022). Interannual variability in $I_{FOS}$ is the largest source of uncertainty in the GDP-based emissions projections. We
thus use the standard deviation of the annual IFOS for the period 2012-2021 as a measure of uncertainty, reflecting a
±1σ as in the rest of the carbon budget. For rest-of-world oil emissions growth, we use the global oil demand forecast
published by the EIA less our projections for the other four regions, and estimate uncertainty as the maximum absolute
difference over the period available for such forecasts using the specific monthly edition (e.g. August) compared to the
first estimate based on more solid data in the following year (April).
**World**: The global total is the sum of each of the countries and regions.





**C.2 Methodology CO₂ emissions from land-use, land-use change and forestry (E_LUC)**
The net $CO_2$ flux from land-use, land-use change and forestry ($E_{LUC}$, called land-use change emissions in the rest of the
text) includes $CO_2$ fluxes from deforestation, afforestation, logging and forest degradation (including harvest activity),
shifting cultivation (cycle of cutting forest for agriculture, then abandoning), and regrowth of forests following wood
harvest or abandonment of agriculture. Emissions from peat burning and drainage are added from external datasets (see
section C.2.1 below). Only some land-management activities are included in our land-use change emissions estimates
(Table A1). Some of these activities lead to emissions of $CO_2$ to the atmosphere, while others lead to $CO_2$ sinks. $E_{LUC}$ is
the net sum of emissions and removals due to all anthropogenic activities considered. Our annual estimate for 1960-
2021 is provided as the average of results from three bookkeeping approaches (Section C.2.1 below): an estimate using
the Bookkeeping of Land Use Emissions model (Hansis et al., 2015; hereafter BLUE) and one using the compact Earth
system model OSCAR (Gasser et al., 2020), both BLUE and OSCAR being updated here to new land-use forcing
covering the time period until 2021, and an updated version of the estimate published by Houghton and Nassikas (2017)
(hereafter updated H&N2017). All three data sets are then extrapolated to provide a projection for 2022 (Section C.2.5
below). In addition, we use results from Dynamic Global Vegetation Models (DGVMs; see Section 2.5 and Table 4) to
help quantify the uncertainty in $E_{LUC}$ (Section C.2.4), and thus better characterise our understanding. Note that in this
budget, we use the scientific $E_{LUC}$ definition, which counts fluxes due to environmental changes on managed land
towards $S_{LAND}$, as opposed to the national greenhouse gas inventories under the UNFCCC, which include them in $E_{LUC}$
and thus often report smaller land-use emissions (Grassi et al., 2018; Petrescu et al., 2020). However, we provide a
methodology of mapping of the two approaches to each other further below (Section C.2.3).
**C.2.1 Bookkeeping models**
Land-use change $CO_2$ emissions and uptake fluxes are calculated by three bookkeeping models. These are based on the
original bookkeeping approach of Houghton (2003) that keeps track of the carbon stored in vegetation and soils before
and after a land-use change (transitions between various natural vegetation types, croplands, and pastures). Literature-
based response curves describe decay of vegetation and soil carbon, including transfer to product pools of different
lifetimes, as well as carbon uptake due to regrowth. In addition, the bookkeeping models represent long-term
degradation of primary forest as lowered standing vegetation and soil carbon stocks in secondary forests, and include
forest management practices such as wood harvests.
BLUE and the updated H&N2017 exclude land ecosystems' transient response to changes in climate, atmospheric $CO_2$
and other environmental factors, and base the carbon densities on contemporary data from literature and inventory data.
Since carbon densities thus remain fixed over time, the additional sink capacity that ecosystems provide in response to
$CO_2$-fertilisation and some other environmental changes is not captured by these models (Pongratz et al., 2014). On the
contrary, OSCAR includes this transient response, and it follows a theoretical framework (Gasser and Ciais, 2013) that
allows separating bookkeeping land-use emissions and the loss of additional sink capacity. Only the former is included
here, while the latter is discussed in Appendix D4. The bookkeeping models differ in (1) computational units (spatially
explicit treatment of land-use change for BLUE, country-level for the updated H&N2017 and OSCAR), (2) processes
represented (see Table A1), and (3) carbon densities assigned to vegetation and soil of each vegetation type (literature-
based for the updated H&N2017 and BLUE, calibrated to DGVMs for OSCAR). A notable difference between models
exists with respect to the treatment of shifting cultivation. The update of H&N2017, introduced for the GCB2021
(Friedlingstein et al., 2022) changed the approach over the earlier H&N2017 version: H&N2017 had assumed the



"excess loss" of tropical forests (i.e., when FRA indicated a forest loss larger than the increase in agricultural areas
from FAO) resulted from converting forests to croplands at the same time older croplands were abandoned. Those
abandoned croplands began to recover to forests after 15 years. The updated H&N2017 now assumes that forest loss in
excess of increases in cropland and pastures represented an increase in shifting cultivation. When the excess loss of
forests was negative, it was assumed that shifting cultivation was returned to forest. Historical areas in shifting
cultivation were extrapolated taking into account country-based estimates of areas in fallow in 1980 (FAO/UNEP,
1981) and expert opinion (from Heinimann et al., 2017). In contrast, the BLUE and OSCAR models include sub-grid-
scale transitions between all vegetation types. Furthermore, the updated H&N2017 assume conversion of natural
grasslands to pasture, while BLUE and OSCAR allocate pasture proportionally on all natural vegetation that exists in a
grid-cell. This is one reason for generally higher emissions in BLUE and OSCAR. Bookkeeping models do not directly
capture carbon emissions from peat fires, which can create large emissions and interannual variability due to synergies
of land-use and climate variability in Southeast Asia, particularly during El-Niño events, nor emissions from the
organic layers of drained peat soils. To correct for this, we add peat fire emissions based on the Global Fire Emission
Database (GFED4s; van der Werf et al., 2017) to the bookkeeping models' output. As these satellite-derived estimates
start in 1997 only, we follow the approach by Houghton and Nassikas (2017) for earlier years, which ramps up from
zero emissions in 1980 to 0.04 Pg C yr 1 in 1996, reflecting the onset of major clearing of peatlands in equatorial
Southeast Asia in the 1980s. Similarly, we add estimates of  peat drainage emissions. In recent years, more peat
drainage estimates that provide spatially explicit data have become available, and we thus extended the number of peat
drainage datasets considered: We employ FAO peat drainage emissions 1990–2019 from croplands and grasslands
(Conchedda and Tubiello, 2020), peat drainage emissions 1700–2010 from simulations with the DGVM ORCHIDEE-
PEAT (Qiu et al., 2021), and peat drainage emissions 1701–2021 from simulations with the DGVM LPX-Bern (Lienert
and Joos, 2018; Müller and Joos, 2021) applying the updated LUH2 forcing as also used by BLUE, OSCAR and the
DGVMs. We extrapolate the FAO data to 1850-2021 by keeping the post-2019 emissions constant at 2019 levels, by
linearly increasing tropical drainage emissions between 1980 and 1990 starting from 0 GtC yr-1 in 1980, consistent
with H&N2017's assumption (Houghton and Nassikas, 2017), and by keeping pre-1990 emissions from the often old
drained areas of the extra-tropics constant at 1990 emission levels. ORCHIDEE-PEAT data are extrapolated to 2011-
2021 by replicating the average emissions in 2000-2010 (pers. comm. C. Qiu). Further, ORCHIDEE-PEAT only
provides peat drainage emissions north of 30°N, and thus we fill the regions south of 30°N by the average peat drainage
emissions from FAO and LPX-Bern. The average of the carbon emission estimates by the three different peat drainage
dataset is added to the bookkeeping models to obtain net ELUC and gross sources.
The three bookkeeping estimates used in this study differ with respect to the land-use change data used to drive the
models. The updated H&N2017 base their estimates directly on the Forest Resource Assessment of the FAO which
provides statistics on forest-area change and management at intervals of five years currently updated until 2020 (FAO,
2020). The data is based on country reporting to FAO and may include remote-sensing information in more recent
assessments. Changes in land-use other than forests are based on annual, national changes in cropland and pasture areas
reported by FAO (FAOSTAT, 2021). On the other hand, BLUE uses the harmonised land-use change data LUH2-
GCB2022 covering the entire 850-2021 period (an update to the previously released LUH2 v2h dataset; Hurtt et al.,
2017; Hurtt et al., 2020), which was also used as input to the DGVMs (Section C.2.2). It describes land-use change,
also based on the FAO data as described in Section C.2.2 as well as the HYDE3.3 dataset (Klein Goldewijk et al.,
2017a, 2017b), but provided at a quarter-degree spatial resolution, considering sub-grid-scale transitions between



primary forest, secondary forest, primary non-forest, secondary non-forest, cropland, pasture, rangeland, and urban land
(Hurtt et al., 2020; Chini et al., 2021). LUH2-GCB2022 provides a distinction between rangelands and pasture, based
on inputs from HYDE. To constrain the models' interpretation on whether rangeland implies the original natural
vegetation to be transformed to grassland or not (e.g., browsing on shrubland), a forest mask was provided with LUH2-
GCB2021; forest is assumed to be transformed to grasslands, while other natural vegetation remains (in case of
secondary vegetation) or is degraded from primary to secondary vegetation (Ma et al., 2020). This is implemented in
BLUE. OSCAR was run with both LUH2-GCB2022 and FAO/FRA (as used with the updated H&N2017), where the
drivers of the latter were linearly extrapolated to 2021 using their 2015–2020 trends. The best-guess OSCAR estimate
used in our study is a combination of results for LUH2-GCB2022 and FAO/FRA land-use data and a large number of
perturbed parameter simulations weighted against a constraint (the cumulative $S_{LAND}$ over 1960-2020 of last year's
GCB) . As the record of the updated H&N2017 ends in 2020, we extend it to 2021 by adding the difference of the
emissions from tropical deforestation and degradation, peat drainage, and peat fire between 2020 and 2021 to the
model's estimate for 2020 (i.e. considering the yearly anomalies of the emissions from tropical deforestation and
degradation, peat drainage, and peat fire). The same method is applied to all three bookkeeping estimates to provide a
projection for 2022.
For $E_{LUC}$ from 1850 onwards we average the estimates from BLUE, the updated H&N2017 and OSCAR. For the
cumulative numbers starting 1750 an average of four earlier publications is added (30 ± 20 PgC 1750-1850, rounded to
nearest 5; Le Quéré et al., 2016).
We provide estimates of the gross land use change fluxes from which the reported net land-use change flux, $E_{LUC}$, is
derived as a sum. Gross fluxes are derived internally by the three bookkeeping models: Gross emissions stem from
decaying material left dead on site and from products after clearing of natural vegetation for agricultural purposes or
wood harvesting, emissions from peat drainage and peat burning, and, for BLUE, additionally from degradation from
primary to secondary land through usage of natural vegetation as rangeland. Gross removals stem from regrowth after
agricultural abandonment and wood harvesting. Gross fluxes for the updated H&N2017 for 2020 and for the 2022
projection of all three models were calculated by the change in emissions from tropical deforestation and degradation
and peat burning and drainage as described for the net ELUC above: As tropical deforestation and degradation and peat
burning and drainage all only lead to gross emissions to the atmosphere, only gross (and net) emissions are adjusted this
way, while gross sinks are assumed to remain constant over the previous year. .
This year, we provide an additional split of the net $E_{LUC}$ into component fluxes to better identify reasons for divergence
between bookkeeping estimates and to give more insight into the drivers of sources and sinks. This split distinguishes
between fluxes from deforestation (including due to shifting cultivation), fluxes from organic soils (i.e., peat drainage
and fires), fluxes on forests (slash and product decay following wood harvesting; regrowth associated with wood
harvesting or after abandonment, including reforestation and in shifting cultivation cycles; afforestation) and fluxes
associated with all other transitions.

### C.2.2 Dynamic Global Vegetation Models (DGVMs)

Land-use change $CO_2$ emissions have also been estimated using an ensemble of 16 DGVMs simulations. The DGVMs
account for deforestation and regrowth, the most important components of $E_{LUC}$, but they do not represent all processes
resulting directly from human activities on land (Table A1). All DGVMs represent processes of vegetation growth and



mortality, as well as decomposition of dead organic matter associated with natural cycles, and include the vegetation
and soil carbon response to increasing atmospheric $CO_2$ concentration and to climate variability and change. Most
models explicitly simulate the coupling of carbon and nitrogen cycles and account for atmospheric N deposition and N
fertilisers (Table A1). The DGVMs are independent from the other budget terms except for their use of atmospheric
$CO_2$ concentration to calculate the fertilisation effect of $CO_2$ on plant photosynthesis.
All DGVMs use the LUH2-GCB2022 dataset as input, which includes the HYDE cropland/grazing land dataset (Klein
Goldewijk et al., 2017a, 2017b), and additional information on land-cover transitions and wood harvest. DGVMs use
annual, half-degree (regridded from 5 minute resolution),  fractional data on cropland and pasture from HYDE3.3.
DGVMs that do not simulate subgrid scale transitions (i.e., net land-use emissions; see Table A1) used the HYDE
information on agricultural area change. For all countries, with the exception of Brazil and the Democratic Republic of
the Congo (DRC), these data are based on the available annual FAO statistics of change in agricultural land area
available from 1961 up to and including 2017. The FAO retrospectively revised their reporting for DRC, which was
newly available until 2020. In addition to FAO country-level statistics the HYDE3.3 cropland/grazing land dataset is
constrained spatially based on multi-year satellite land cover maps from ESA CCI LC (see below). . After the year
2017, LUH2 extrapolates, on a gridcell-basis, the cropland, pasture, and urban data linearly based on the trend over the
previous 5 years, to generate data until the year 2021. This extrapolation methodology is not appropriate for countries
which have experienced recent rapid changes in the rate of land-use change, e.g. Brazil which has experienced a recent
upturn in deforestation. Hence, for Brazil we replace FAO state-level data for cropland and grazing land in HYDE by
those from in-country land cover dataset MapBiomas (collection 6) for 1985-2020 (Souza et al. 2020). ESA-CCI is
used to spatially disaggregate as described below. Similarly, an estimate for the year 2021 is based on the MapBiomas
trend 2015-2020. The pre-1985 period is scaled with the per capita numbers from 1985 from MapBiomas, so this
transition is smooth.
HYDE uses satellite imagery from ESA-CCI from 1992 – 2018 for more detailed yearly allocation of cropland and
grazing land, with the ESA area data scaled to match the FAO annual totals at country-level. The original 300 metre
spatial resolution data from ESA was aggregated to a 5 arc minute resolution according to the classification scheme as
described in Klein Goldewijk et al (2017a).
DGVMs that simulate subgrid scale transitions (i.e., gross land-use emissions; see Table A1) use more detailed land use
transition and wood harvest information from the LUH2-GCB2022 data set. LUH2-GCB2022 is an update of the more
comprehensive harmonised land-use data set (Hurtt et al., 2020), that further includes fractional data on primary and
secondary forest vegetation, as well as all underlying transitions between land-use states (850-2020; Hurtt et al., 2011,
2017, 2020; Chini et al., 2021; Table A1). This data set is of quarter degree fractional areas of land-use states and all
transitions between those states, including a new wood harvest reconstruction, new representation of shifting
cultivation, crop rotations, management information including irrigation and fertiliser application. The land-use states
include five different crop types in addition to splitting grazing land into managed pasture and rangeland. Wood harvest
patterns are constrained with Landsat-based tree cover loss data (Hansen et al. 2013). Updates of LUH2-GCB2022 over
last year's version (LUH2-GCB2021) are using the most recent HYDE release (covering the time period up to 2017,
revision to Brazil and DRC as described above). We use the same FAO wood harvest data as last year for all dataset
years from 1961 to 2019, and extrapolate to the year 2022. The HYDE3.3 population data is also used to extend the
wood harvest time series back in time. Other wood harvest inputs (for years prior to 1961) remain the same in LUH2.



These updates in the land-use forcing are shown in comparison to the more pronounced version change from  the
GCB2020 (Friedlingstein et al., 2020) to GCB2021, which was discussed in Friedlingstein et al. (2022a) in Figure B6
and their relevance for land-use emissions discussed in Section 3.2.2. DGVMs implement land-use change differently
(e.g., an increased cropland fraction in a grid cell can either be at the expense of grassland or shrubs, or forest, the latter
resulting in deforestation; land cover fractions of the non-agricultural land differ between models). Similarly, model-
specific assumptions are applied to convert deforested biomass or deforested area, and other forest product pools into
carbon, and different choices are made regarding the allocation of rangelands as natural vegetation or pastures.
The difference between two DGVMs simulations (See Section C4.1 below), one forced with historical changes in land-
use and a second with time-invariant pre-industrial land cover and pre-industrial wood harvest rates, allows
quantification of the dynamic evolution of vegetation biomass and soil carbon pools in response to land-use change in
each model ($E_{LUC}$). Using the difference between these two DGVMs simulations to diagnose $E_{LUC}$ means the DGVMs
account for the loss of additional sink capacity (around $0.4 \pm 0.3$ GtC yr-1; see Section 2.7.4, Appendix D4), while the
bookkeeping models do not.
As a criterion for inclusion in this carbon budget, we only retain models that simulate a positive $E_{LUC}$ during the 1990s,
as assessed in the IPCC AR4 (Denman et al., 2007) and AR5 (Ciais et al., 2013).  All DGVMs met this criterion,
although one model was not included in the $E_{LUC}$ estimate from DGVMs as it exhibited a spurious response to the
transient land cover change forcing after its initial spin-up.
**C.2.3 Mapping of national GHG inventory data to $E_{LUC}$**
An approach was implemented to reconcile the large gap between ELUC from bookkeeping models and land use, land-
use change and forestry (LULUCF) from national GHG Inventories (NGHGI) (see Tab. A8). This gap is due to
different approaches to calculating "anthropogenic" $CO_2$ fluxes related to land-use change and land management
(Grassi et al. 2018). In particular, the land sinks due to environmental change on managed lands are treated as non-
anthropogenic in the global carbon budget, while they are generally considered as anthropogenic in NGHGIs ("indirect
anthropogenic fluxes"; Eggleston et al., 2006). Building on previous studies (Grassi et al. 2021), the approach
implemented here adds the DGVMs estimates of $CO_2$ fluxes due to environmental change from countries' managed
forest area (part of the $S_{LAND}$) to the original $E_{LUC}$ flux. This sum is expected to be conceptually more comparable to
LULUCF than simply $E_{LUC}$.
ELUC data are taken from bookkeeping models, in line with the global carbon budget approach. To determine $S_{LAND}$ on
managed forest, the following steps were taken: Spatially gridded data of "natural" forest NBP ($S_{LAND}$ i.e., due to
environmental change and excluding land use change fluxes) were obtained with S2 runs from DGVMs up to 2021
from the TRENDY v11 dataset. Results were first masked with a forest map that is based on Hansen (Hansen et
al.2013) tree cover data. To do this conversion ("tree" cover to "forest" cover), we exclude gridcells with less than 20%
tree cover and isolated pixels with maximum connectivity less than 0.5 ha following the FAO definition of forest.
Forest NBP are then further masked with the "intact" forest map for the year 2013, i.e. forest areas characterised by no
remotely detected signs of human activity (Potapov et al. 2017). This way, we obtained the SLAND in "intact" and
"non-intact" forest area, which previous studies (Grassi et al. 2021) indicated to be a good proxy, respectively, for
"unmanaged" and "managed" forest area in the NGHGI. Note that only 4 models (CABLE-POP, CLASSIC, JSBACH
and YIBs) had forest NBP at grid cell level. For the other DGVMs, when a grid cell had forest, all the NBP was



allocated to forest. However, since S2 simulations use pre-industrial forest cover masks that are at least 20% larger than
today's forest (Hurtt et al. 2020), we corrected this NBP by a ratio between observed (based on Hansen) and prescribed
(from DGVMs) forest cover. This ratio is calculated for each individual DGVM that provides information on prescribed
forest cover (LPX-Bern, OCN, JULES, VISIT, VISIT-NIES, SDGVM). For the others (IBIS, CLM5.0, ORCHIDEE,
ISAM, DLEM, LPJ-GUESS) a common ratio (median ratio of all the 10 models that provide information on prescribed
forest cover) is used. The details of the method used are explained here:
https://github.com/RamAlkama/LandCarbonBudget_IntactAndNonIntactForest
LULUCF data from NGHGIs are from Grassi et al. (2022a). While Annex I countries report a complete time series
1990-2020, for Non-Annex I countries gap-filling was applied through linear interpolation between two points and/or
through extrapolation backward (till 1990) and forward (till 2020) using the single closest available data. For all
countries, the year 2021 is assumed to be equal to 2020.. This data includes all CO2 fluxes from land considered
managed, which in principle encompasses all land uses (forest land, cropland, grassland, wetlands, settlements, and
other land), changes among them, emissions from organic soils and from fires. In practice, although almost all Annex I
countries report all land uses, many non-Annex I countries report only on deforestation and forest land, and only few
countries report on other land uses. In most cases, NGHGI include most of the natural response to recent environmental
change, because they use direct observations (e.g., national forest inventories) that do not allow separating direct and
indirect anthropogenic effects (Eggleston et al., 2006).
To provide additional, largely independent assessments of fluxes on unmanaged vs managed lands, we include a
DGVM that allows diagnosing fluxes from unmanaged vs managed lands by tracking vegetation cohorts of different
ages separately. This model, ORCHIDEE-MICT (Yue et al., 2018), was run using the same LUH2 forcing as the
DGVMs used in this budget (Section 2.5) and the bookkeeping models BLUE and OSCAR (Section 2.2). Old-aged
forest was classified as primary forest after a certain threshold of carbon density was reached again, and the model-
internal distinction between primary and secondary forest used as proxies for unmanaged vs managed forests;
agricultural lands are added to the latter to arrive at total managed land.
Tab. A8 shows the resulting mapping of global carbon cycle models' land flux definitions to that of the NGHGI
(discussed in Section 3.2.2). ORCHIDEE-MICT estimates for SLAND on intact forests are expected to be higher than
based on DGVMs in combination with the NGHGI managed/unmanaged forest data because the unmanaged forest
area, with about 27 mio km2, is estimated to be substantially larger by ORCHIDEE-MICT than, with less than 10 mio
km2, by the NGHGI, while managed forest area is estimated to be smaller (22 compared to 32 mio km2). Related to
this, $E_{LUC}$ plus $S_{LAND}$ on non-intact lands is a larger source estimated by ORCHIDEE-MICT compared to NGHGI. We
also show as comparison FAOSTAT emissions totals (FAO, 2021), which include emissions from net forest conversion
and fluxes on forest land (Tubiello et al., 2021) as well as $CO_2$ emissions from peat drainage and peat fires. The 2021
data was estimated by including actual 2021 estimates for peatlands drainage and fire and a carry forward from 2020 to
2021 for the forest land stock change. The FAO data shows a global source of 0.24 GtC yr$^{-1}$ averaged over 2012-2021,
in contrast to the sink of -0.54 GtC yr$^{-1}$ of the gap-filled NGHGI data. Most of this difference is attributable to different
scopes: a focus on carbon fluxes for the NGHGI and a focus on area and biomass for FAO. In particular, the NGHGI
data includes a larger forest sink for non-Annex 1 countries resulting from a more complete coverage of non-biomass
carbon pools and non-forest land uses. NGHGI and FAO data also differ in terms of underlying data on forest land
(Grassi et al., 2022a).

### C.2.4 Uncertainty assessment for $E_{LUC}$

Differences between the bookkeeping models and DGVMs models originate from three main sources: the different methodologies, which among others lead to inclusion of the loss of additional sink capacity in DGVMs (see Appendix D1.4), the underlying land-use/land cover data set, and the different processes represented (Table A1). We examine the results from the DGVMs models and of the bookkeeping method and use the resulting variations as a way to characterise the uncertainty in $E_{LUC}$.

Despite these differences, the $E_{LUC}$ estimate from the DGVMs multi-model mean is consistent with the average of the emissions from the bookkeeping models (Table 5). However there are large differences among individual DGVMs (standard deviation at around 0.5 GtC yr$^{-1}$; Table 5), between the bookkeeping estimates (average difference 1850-2020 BLUE-updated H&N2017 of 0.8 GtC yr$^{-1}$, BLUE-OSCAR of 0.4 GtC yr$^{-1}$, OSCAR-updated H&N2017 of 0.3 GtC yr$^{-1}$), and between the updated estimate of H&N2017 and its previous model version (Houghton et al., 2012). A factorial analysis of differences between BLUE and H&N2017 attributed them particularly to differences in carbon densities between natural and managed vegetation or primary and secondary vegetation (Bastos et al., 2021). Earlier studies additionally showed the relevance of the different land-use forcing as applied (in updated versions) also in the current study (Gasser et al., 2020). Ganzenmüller et al. (2022) recently showed that $E_{LUC}$ estimates with BLUE are substantially smaller when the model is driven by a new high-resolution land-use dataset (HILDA+). They identified shifting cultivation and the way it is implemented in LUH2 as a main reason for this divergence. They further showed that a higher spatial resolution reduces the estimates of both sources and sinks because successive transitions are not adequately represented at coarser resolution, which has the effect that—despite capturing the same extent of transition areas—overall less area remains pristine at the coarser compared to the higher resolution.

The uncertainty in $E_{LUC}$ of ±0.7 GtC yr$^{-1}$ reflects our best value judgement that there is at least 68% chance (±1σ) that the true land-use change emission lies within the given range, for the range of processes considered here. Prior to the year 1959, the uncertainty in $E_{LUC}$ was taken from the standard deviation of the DGVMs. We assign low confidence to the annual estimates of $E_{LUC}$ because of the inconsistencies among estimates and of the difficulties to quantify some of the processes in DGVMs.

### C.2.5 Emissions projections for $E_{LUC}$

We project the 2022 land-use emissions for BLUE, the updated H&N2017 and OSCAR, starting from their estimates for 2021 assuming unaltered peat drainage, which has low interannual variability, and the highly variable emissions from peat fires, tropical deforestation and degradation as estimated using active fire data (MCD14ML; Giglio et al., 2016). Those latter scale almost linearly with GFED over large areas (van der Werf et al., 2017), and thus allows for tracking fire emissions in deforestation and tropical peat zones in near-real time.

### C.3 Methodology Ocean $CO_2$ sink

### C.3.1 Observation-based estimates

We primarily use the observational constraints assessed by IPCC of a mean ocean $CO_2$ sink of 2.2 ± 0.7 GtC yr$^{-1}$ for the 1990s (90% confidence interval; Ciais et al., 2013) to verify that the GOBMs provide a realistic assessment of $S_{OCEAN}$.

This is based on indirect observations with seven different methodologies and their uncertainties, and further using
three of these methods that are deemed most reliable for the assessment of this quantity (Denman et al., 2007; Ciais et
al., 2013). The observation-based estimates use the ocean/land $CO_2$ sink partitioning from observed atmospheric $CO_2$
and $O_2/N_2$ concentration trends (Manning and Keeling, 2006; Keeling and Manning, 2014), an oceanic inversion
method constrained by ocean biogeochemistry data (Mikaloff Fletcher et al., 2006), and a method based on penetration
time scale for chlorofluorocarbons (McNeil et al., 2003). The IPCC estimate of 2.2 GtC yr$^{-1}$ for the 1990s is consistent
with a range of methods (Wanninkhof et al., 2013). We refrain from using the IPCC estimates for the 2000s (2.3 ± 0.7
GtC yr$^{-1}$), and the period 2002-2011 (2.4 ± 0.7 GtC yr$^{-1}$, Ciais et al., 2013) as these are based on trends derived mainly
from models and one data-product (Ciais et al., 2013). Additional constraints summarised in AR6 (Canadell et al.,
2021) are the interior ocean anthropogenic carbon change (Gruber et al., 2019) and ocean sink estimate from
atmospheric $CO_2$ and $O_2/N_2$ (Tohjima et al., 2019) which are used for model evaluation and discussion, respectively.
We also use eight estimates of the ocean $CO_2$ sink and its variability based on surface ocean f$CO_2$ maps obtained by the
interpolation of surface ocean f$CO_2$ measurements from 1990 onwards due to severe restriction in data availability prior
to 1990 (Figure 10). These estimates differ in many respects: they use different maps of surface f$CO_2$, different
atmospheric $CO_2$ concentrations, wind products and different gas-exchange formulations as specified in Table A3. We
refer to them as f$CO_2$-based flux estimates. The measurements underlying the surface f$CO_2$ maps are from the Surface
Ocean $CO_2$ Atlas version 2022 (SOCATv2022; Bakker et al., 2022), which is an update of version 3 (Bakker et al.,
2016) and contains quality-controlled data through 2021 (see data attribution Table A5). Each of the estimates uses a
different method to then map the SOCAT v2022 data to the global ocean. The methods include a data-driven diagnostic
method combined with a multi linear regression approach to extend back to 1957 (Rödenbeck et al., 2022; referred to
here as Jena-MLS), three neural network models (Landschützer et al., 2014; referred to as MPI-SOMFFN; Chau et al.,
2022; Copernicus Marine Environment Monitoring Service, referred to here as CMEMS-LSCE-FFNN; and Zeng et al.,
2014; referred to as NIES-NN), one cluster regression approaches (Gregor and Gruber, 2021, referred to as OS-ETHZ-
GRaCER), and a multi-linear regression method (Iida et al., 2021; referred to as JMA-MLR), and one method that
relates the fCO2 misfit between GOBMs and SOCAT to environmental predictors using the extreme gradient boosting
method (Gloege et al., 2022). The ensemble mean of the f$CO_2$-based flux estimates is calculated from these seven
mapping methods. Further, we show the flux estimate of Watson et al. (2020) who also use the MPI-SOMFFN method
to map the adjusted f$CO_2$ data to the globe, but resulting in a substantially larger ocean sink estimate, owing to a
number of adjustments they applied to the surface ocean f$CO_2$ data. Concretely, these authors adjusted the SOCAT
f$CO_2$ downward to account for differences in temperature between the depth of the ship intake and the relevant depth
right near the surface, and included a further adjustment to account for the cool surface skin temperature effect. The
Watson et al. flux estimate hence differs from the others by their choice of adjusting the flux to a cool, salty ocean
surface skin. Watson et al. (2020) showed that this temperature adjustment leads to an upward correction of the ocean
carbon sink, up to 0.9 GtC yr$^{-1}$, that, if correct, should be applied to all f$CO_2$-based flux estimates. A reduction of this
adjustment to 0.6 GtC yr$^{-1}$ was proposed by Dong et al. (2022). The impact of the cool skin effect on air-sea $CO_2$ flux is
based on established understanding of temperature gradients (as discussed by Goddijn-Murphy et al 2015), and
laboratory observations (Jähne and Haussecker, 1998; Jähne, 2019), but in situ field observational evidence is lacking
(Dong et al., 2022). The Watson et al flux estimate presented here is therefore not included in the ensemble mean of the
f$CO_2$-based flux estimates. This choice will be re-evaluated in upcoming budgets based on further lines of evidence.

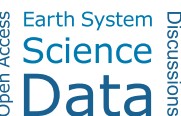

Typically, data products do not cover the entire ocean due to missing coastal oceans and sea ice cover. The $CO_2$ flux
from each $fCO_2$-based product is already at or above 99% coverage of the ice-free ocean surface area in two products
(Jena-MLS, OS-ETHZ-GRaCER), and filled by the data-provider in three products (using Fay et al., 2021a, method for
JMA-MLR and LDEO-HPD; and Landschützer et al., 2020, methodology for MPI-SOMFFN). The products that
remained below 99% coverage of the ice-free ocean (CMEMS-LSCE-FFNN, MPI-SOMFFN, NIES-NN, UOx-Watson)
were scaled by the following procedure.
In previous versions of the GCB, the missing areas were accounted for by scaling the globally integrated fluxes by the
fraction of the global ocean coverage (361.9e6 km$^2$ based on ETOPO1, Amante and Eakins, 2009; Eakins and Sharman,
2010) with the area covered by the $CO_2$ flux predictions. This approach may lead to unnecessary scaling when the
majority of the missing data are in the ice-covered region (as is often the case), where flux is already assumed to be
zero. To avoid this unnecessary scaling, we now scale fluxes regionally (North, Tropics, South) to match the ice-free
area (using NOAA's OISSTv2, Reynolds et al., 2002):
$$FCO_2^{reg-scaled} = \frac{A_{(1-ice)}^{region}}{A_{FCO_2}^{region}} \cdot FCO_2^{region}$$
In the equation, $A$ represents area, $(1 - ice)$ represents the ice free ocean, $A_{FCO_2}^{region}$ represents the coverage
of the data product for a region, and $FCO_2^{region}$ is the integrated flux for a region.
We further use results from two diagnostic ocean models, Khatiwala et al. (2013) and DeVries (2014), to estimate the
anthropogenic carbon accumulated in the ocean prior to 1959. The two approaches assume constant ocean circulation
and biological fluxes, with $S_{OCEAN}$ estimated as a response in the change in atmospheric $CO_2$ concentration calibrated to
observations. The uncertainty in cumulative uptake of ±20 GtC (converted to ±1σ) is taken directly from the IPCC's
review of the literature (Rhein et al., 2013), or about ±30% for the annual values (Khatiwala et al., 2009).
**C.3.2 Global Ocean Biogeochemistry Models (GOBMs)**
The ocean $CO_2$ sink for 1959-20121 is estimated using ten GOBMs (Table A2). The GOBMs represent the physical,
chemical, and biological processes that influence the surface ocean concentration of $CO_2$ and thus the air-sea $CO_2$ flux.
The GOBMs are forced by meteorological reanalysis and atmospheric $CO_2$ concentration data available for the entire
time period. They mostly differ in the source of the atmospheric forcing data (meteorological reanalysis), spin up
strategies, and in their horizontal and vertical resolutions (Table A2). All GOBMs except two (CESM-ETHZ, CESM2)
do not include the effects of anthropogenic changes in nutrient supply (Duce et al., 2008). They also do not include the
perturbation associated with changes in riverine organic carbon (see Section 2.7.3).
Four sets of simulations were performed with each of the GOBMs. Simulation A applied historical changes in climate
and atmospheric $CO_2$ concentration. Simulation B is a control simulation with constant atmospheric forcing (normal
year or repeated year forcing) and constant pre-industrial atmospheric $CO_2$ concentration. Simulation C is forced with
historical changes in atmospheric $CO_2$ concentration, but repeated year or normal year atmospheric climate forcing.
Simulation D is forced by historical changes in climate and constant pre-industrial atmospheric $CO_2$ concentration. To
derive $S_{OCEAN}$ from the model simulations, we subtracted the slope of a linear fit to the annual time series of the control
simulation B from the annual time series of simulation A. Assuming that drift and bias are the same in simulations A
and B, we thereby correct for any model drift. Further, this difference also removes the natural steady state flux
(assumed to be 0 GtC yr$^{-1}$ globally without rivers) which is often a major source of biases. This approach works for all
model set-ups, including IPSL, where simulation B was forced with constant atmospheric $CO_2$ but observed historical



changes in climate (equivalent to simulation D). This approach assures that the interannual variability is not removed
from IPSL simulation A.
The absolute correction for bias and drift per model in the 1990s varied between <0.01 GtC yr$^{-1}$ and 0.41 GtC yr$^{-1}$, with
seven models having positive biases, two having negative biases and one model having essentially no bias (NorESM).
The MPI model uses riverine input and therefore simulates outgassing in simulation B. By subtracting simulation B,
also the ocean carbon sink of the MPI model follows the definition of $S_{OCEAN}$. This correction reduces the model mean
ocean carbon sink by 0.04 GtC yr$^{-1}$ in the 1990s. The ocean models cover 99% to 101% of the total ocean area, so that
area-scaling is not necessary.

**C.3.3 GOBM evaluation and uncertainty assessment for $S_{OCEAN}$**

The ocean $CO_2$ sink for all GOBMs and the ensemble mean falls within 90% confidence of the observed range, or 1.5
to 2.9 GtC yr$^{-1}$ for the 1990s (Ciais et al., 2013) before and after applying adjustments. An exception is the MPI model,
which simulates a low ocean carbon sink of 1.38 GtC yr$^{-1}$ for the 1990s in simulation A owing to the inclusion of
riverine carbon flux. After adjusting to the GCB's definition of $S_{OCEAN}$ by subtracting simulation B, the MPI model falls
into the observed range with an estimated sink of 1.69 GtC yr$^{-1}$.
The GOBMs and data products have been further evaluated using the fugacity of sea surface $CO_2$ (fCO$_2$) from the
SOCAT v2022 database (Bakker et al., 2016, 2022). We focused this evaluation on the root mean squared error
(RMSE) between observed and modelled fCO$_2$ and on a measure of the amplitude of the interannual variability of the
flux (modified after Rödenbeck et al., 2015).  The RMSE is calculated from detrended, annually and regionally
averaged time series calculated from GOBMs and data-product fCO$_2$ subsampled to SOCAT sampling points to
measure the misfit between large-scale signals (Hauck et al., 2020). To this end, we apply the following steps: (i)
subsample data points for where there are observations (GOBMs/data-products as well as SOCAT), (ii) average
spatially, (iii) calculate annual mean, (iv) detrend both time-series (GOBMs/data-products as well as SOCAT), (v)
calculate RMSE.  This year, we do not apply an open ocean mask of 400 m, but instead a mask based on the minimum
area coverage of the data-products. This ensures a fair comparison over equal areas. The amplitude of the $S_{OCEAN}$
interannual variability (A-IAV) is calculated as the temporal standard deviation of the detrended annual $CO_2$ flux time
series after area-scaling (Rödenbeck et al., 2015, Hauck et al., 2020). These metrics are chosen because RMSE is the
most direct measure of data-model mismatch and the A-IAV is a direct measure of the variability of $S_{OCEAN}$ on
interannual timescales. We apply these metrics globally and by latitude bands. Results are shown in Figure B2 and
discussed in Section 3.5.5.
We quantify the 1-σ uncertainty around the mean ocean sink of anthropogenic $CO_2$ by assessing random and systematic
uncertainties for the GOBMs and data-products. The random uncertainties are taken from the ensemble standard
deviation (0.3 GtC yr$^{-1}$ for GOBMs, 0.3  GtC yr$^{-1}$ for data-products). We derive the GOBMs systematic uncertainty by
the deviation of the DIC inventory change 1994-2007 from the Gruber et al (2019) estimate (0.4 GtC yr$^{-1}$) and suggest
these are related to physical transport (mixing, advection) into the ocean interior. For the data-products, we consider
systematic uncertainties stemming from uncertainty in fCO$_2$ observations (0.2 GtC yr$^{-1}$ , Takahashi et al., 2009;
Wanninkhof et al., 2013), gas-transfer velocity (0.2 GtC yr$^{-1}$ , Ho et al., 2011; Wanninkhof et al., 2013; Roobaert et al.,
2018), wind product (0.1 GtC yr$^{-1}$, Fay et al., 2021a), river flux adjustment (0.3 GtC yr$^{-1}$, Regnier et al., 2022, formally
2-σ uncertainty), and fCO$_2$ mapping (0.2 GtC yr$^{-1}$, Landschützer et al., 2014). Combining these uncertainties as their



squared sums, we assign an uncertainty of $\pm 0.5$ GtC yr$^{-1}$ to the GOBMs ensemble mean and an uncertainty of $\pm 0.6$
GtC yr$^{-1}$ to the data-product ensemble mean. These uncertainties are propagated as $\sigma(S_{OCEAN}) = (1/2^2 * 0.5^2 + 1/2^2 *$
$0.6^2)^{1/2}$ GtC yr$^{-1}$ and result in an $\pm 0.4$ GtC yr$^{-1}$ uncertainty around the best estimate of $S_{OCEAN}$.
We examine the consistency between the variability of the model-based and the $fCO_2$-based data products to assess
confidence in $S_{OCEAN}$. The interannual variability of the ocean fluxes (quantified as A-IAV, the standard deviation after
detrending, Figure B2) of the seven $fCO_2$-based data products plus the Watson et al. (2020) product for 1990-2021,
ranges from 0.12 to 0.32 GtC yr$^{-1}$ with the lower estimates by the two ensemble methods (CMEMS-LSCE-FFNN, OS-
ETHZ-GRaCER). The inter-annual variability in the GOBMs ranges between 0.09 and 0.20 GtC yr$^{-1}$, hence there is
overlap with the lower A-IAV estimates of two data-products.
Individual estimates (both GOBMs and data products) generally produce a higher ocean $CO_2$ sink during strong El
Niño events. There is emerging agreement between GOBMs and data-products on the patterns of decadal variability of
$S_{OCEAN}$ with a global stagnation in the 1990s and an extra-tropical strengthening in the 2000s (McKinley et al., 2020,
Hauck et al., 2020). The central estimates of the annual flux from the GOBMs and the $fCO_2$-based data products have a
correlation $r$ of 0.94 (1990-2021). The agreement between the models and the data products reflects some consistency
in their representation of underlying variability since there is little overlap in their methodology or use of observations.

**C.4 Methodology Land $CO_2$ sink**
**C.4.1 DGVM simulations**
The DGVMs model runs were forced by either the merged monthly Climate Research Unit (CRU) and 6 hourly
Japanese 55-year Reanalysis (JRA-55) data set or by the monthly CRU data set, both providing observation-based
temperature, precipitation, and incoming surface radiation on a 0.5°x0.5° grid and updated to 2021 (Harris et al., 2014,
2020). The combination of CRU monthly data with 6 hourly forcing from JRA-55 (Kobayashi et al., 2015) is performed
with methodology used in previous years (Viovy, 2016) adapted to the specifics of the JRA-55 data.
Introduced in GCB2021 (Friedlingstein et al., 2022a), incoming short-wave radiation fields to take into account aerosol
impacts and the division of total radiation into direct and diffuse components as summarised below.
The diffuse fraction dataset offers 6-hourly distributions of the diffuse fraction of surface shortwave fluxes over the
period 1901-2021. Radiative transfer calculations are based on monthly-averaged distributions of tropospheric and
stratospheric aerosol optical depth, and 6-hourly distributions of cloud fraction. Methods follow those described in the
Methods section of Mercado et al. (2009), but with updated input datasets.
The time series of speciated tropospheric aerosol optical depth is taken from the historical and RCP8.5 simulations by
the HadGEM2-ES climate model (Bellouin et al., 2011). To correct for biases in HadGEM2-ES, tropospheric aerosol
optical depths are scaled over the whole period to match the global and monthly averages obtained over the period
2003-2020 by the CAMS Reanalysis of atmospheric composition (Inness et al., 2019), which assimilates satellite
retrievals of aerosol optical depth.
The time series of stratospheric aerosol optical depth is taken from the by Sato et al. (1993) climatology, which has
been updated to 2012. Years 2013-2020 are assumed to be background years so replicate the background year 2010.
That assumption is supported by the Global Space-based Stratospheric Aerosol Climatology time series (1979-2016;



Thomason et al., 2018). The time series of cloud fraction is obtained by scaling the 6-hourly distributions simulated in
the Japanese Reanalysis (Kobayashi et al., 2015) to match the monthly-averaged cloud cover in the CRU TS v4.06
dataset (Harris et al., 2020). Surface radiative fluxes account for aerosol-radiation interactions from both tropospheric
and stratospheric aerosols, and for aerosol-cloud interactions from tropospheric aerosols, except mineral dust.
Tropospheric aerosols are also assumed to exert interactions with clouds.
The radiative effects of those aerosol-cloud interactions are assumed to scale with the radiative effects of aerosol-
radiation interactions of tropospheric aerosols, using regional scaling factors derived from HadGEM2-ES. Diffuse
fraction is assumed to be 1 in cloudy sky. Atmospheric constituents other than aerosols and clouds are set to a constant
standard mid-latitude summer atmosphere, but their variations do not affect the diffuse fraction of surface shortwave
fluxes.
In summary, the DGVMs forcing data include time dependent gridded climate forcing, global atmospheric $CO_2$
(Dlugokencky and Tans, 2022), gridded land cover changes (see Appendix C.2.2), and gridded nitrogen deposition and
fertilisers (see Table A1 for specific models details).
Four simulations were performed with each of the DGVMs. Simulation 0 (S0) is a control simulation which uses fixed
pre-industrial (year 1700) atmospheric CO2 concentrations, cycles early 20th century (1901-1920) climate and applies a
time-invariant pre-industrial land cover distribution and pre-industrial wood harvest rates. Simulation 1 (S1) differs
from S0 by applying historical changes in atmospheric CO2 concentration and N inputs. Simulation 2 (S2) applies
historical changes in atmospheric $CO_2$ concentration, N inputs, and climate, while applying time-invariant pre-
industrial land cover distribution and pre-industrial wood harvest rates. Simulation 3 (S3) applies historical changes in
atmospheric CO2 concentration, N inputs, climate, and land cover distribution and wood harvest rates.
S2 is used to estimate the land sink component of the global carbon budget ($S_{LAND}$). S3 is used to estimate the total land
flux but is not used in the global carbon budget. We further separate $S_{LAND}$ into contributions from $CO_2$ (=S1-S0) and
climate (=S2-S1+S0).

**C.4.2 DGVM evaluation and uncertainty assessment for $S_{LAND}$**

We apply three criteria for minimum DGVMs realism by including only those DGVMs with (1) steady state after
spin up, (2) global net land flux ($S_{LAND} – E_{LUC}$) that is an atmosphere-to-land carbon flux over the 1990s ranging
between -0.3 and 2.3 GtC yr$^{-1}$, within 90% confidence of constraints by global atmospheric and oceanic observations
(Keeling and Manning, 2014; Wanninkhof et al., 2013), and (3) global $E_{LUC}$ that is a carbon source to the atmosphere
over the 1990s, as already mentioned in section C.2.2. All DGVMs meet these three criteria.
In addition, the DGVMs results are also evaluated using the International Land Model Benchmarking system (ILAMB;
Collier et al., 2018). This evaluation is provided here to document, encourage and support model improvements through
time. ILAMB variables cover key processes that are relevant for the quantification of $S_{LAND}$ and resulting aggregated
outcomes. The selected variables are vegetation biomass, gross primary productivity, leaf area index, net ecosystem
exchange, ecosystem respiration, evapotranspiration, soil carbon, and runoff (see Figure B3 for the results and for the
list of observed databases). Results are shown in Figure B3 and discussed in Section 3.6.5.
For the uncertainty for $S_{LAND}$, we use the standard deviation of the annual $CO_2$ sink across the DGVMs, averaging to
about ± 0.6 GtC yr$^{-1}$ for the period 1959 to 2021. We attach a medium confidence level to the annual land $CO_2$ sink and





its uncertainty because the estimates from the residual budget and averaged DGVMs match well within their respective
uncertainties (Table 5).

**C.5 Methodology Atmospheric Inversions**
**C.5.1 Inversion System Simulations**
Nine atmospheric inversions (details of each in Table A4) were used to infer the spatio-temporal distribution of the $CO_2$
flux exchanged between the atmosphere and the land or oceans. These inversions are based on Bayesian inversion
principles with prior information on fluxes and their uncertainties. They use very similar sets of surface measurements
of $CO_2$ time series (or subsets thereof) from various flask and in situ networks. One inversion system also used satellite
$xCO_2$ retrievals from GOSAT and OCO-2.
Each inversion system uses different methodologies and input data but is rooted in Bayesian inversion principles. These
differences mainly concern the selection of atmospheric $CO_2$ data and prior fluxes, as well as the spatial resolution,
assumed correlation structures, and mathematical approach of the models. Each system uses a different transport model,
which was demonstrated to be a driving factor behind differences in atmospheric inversion-based flux estimates, and
specifically their distribution across latitudinal bands (Gaubert et al., 2019; Schuh et al., 2019).
The inversion systems all prescribe similar global fossil fuel emissions for $E_{FOS}$; specifically, the GCP's Gridded Fossil
Emissions Dataset version 2022 (GCP-GridFEDv2022.2; Jones et al., 2022), which is an update through 2021 of the
first version of GCP-GridFED presented by Jones et al. (2021), or another recent version of GCP-GridFED (Table A4).
All GCP-GridFED versions scale gridded estimates of $CO_2$ emissions from EDGARv4.3.2 (Janssens-Maenhout et al.,
2019) within national territories to match national emissions estimates provided by the GCP for the years 1959-2021,
which are compiled following the methodology described in Appendix C.1. GCP-GridFEDv2022.2 adopts the
seasonality of emissions (the monthly distribution of annual emissions) from the Carbon Monitor (Liu et al., 2020a,b;
Dou et al., 2022) for Brazil, China, all EU27 countries, the United Kingdom, the USA and shipping and aviation bunker
emissions. The seasonality present in Carbon Monitor is used directly for years 2019-2021, while for years 1959-2018
the average seasonality of 2019 and 2021 are applied (avoiding the year 2020 during which emissions were most
impacted by the COVID-19 pandemic). For all other countries, seasonality of emissions is taken from EDGAR
(Janssens-Maenhout et al., 2019; Jones et al., 2022), with small annual correction to the seasonality present in year
2010 based on heating or cooling degree days to account for the effects of inter-annual climate variability on the
seasonality of emissions (Jones et al., 2021). Earlier versions of GridFED used Carbon Monitor-based seasonality only
during the years 2019 onwards. In addition, we note that GCP-GridFEDv2022.1 and v2022.2 include emissions from
cement production and the cement carbonation $CO_2$ sink (Appendix C.1.1), whereas earlier versions of GCP-GridFED
did not include the cement carbonation $CO_2$ sink.
The consistent use of recent versions of GCP-GridFED for $E_{FOS}$ ensures a close alignment with the estimate of $E_{FOS}$
used in this budget assessment, enhancing the comparability of the inversion-based estimate with the flux estimates
deriving from DGVMs, GOBMs and $fCO_2$-based methods. To ensure that the estimated uptake of atmospheric $CO_2$ by
the land and oceans was fully consistent with the sum of the fossil emissions flux from GCP-GridFEDv2022.2 and the
atmospheric growth rate of $CO_2$, small corrections to the fossil fuel emissions flux were applied to inversions systems
using other versions of GCP-GridFED.



The land and ocean $CO_2$ fluxes from atmospheric inversions contain anthropogenic perturbation and natural pre-
industrial $CO_2$ fluxes. On annual time scales, natural pre-industrial fluxes are primarily land $CO_2$ sinks and ocean $CO_2$
sources corresponding to carbon taken up on land, transported by rivers from land to ocean, and outgassed by the
ocean. These pre-industrial land $CO_2$ sinks are thus compensated over the globe by ocean $CO_2$ sources corresponding to
the outgassing of riverine carbon inputs to the ocean, using the exact same numbers and distribution as described for the
oceans in Section 2.4. To facilitate the comparison, we adjusted the inverse estimates of the land and ocean fluxes per
latitude band with these numbers to produce historical perturbation $CO_2$ fluxes from inversions.
**C.5.2 Inversion System Evaluation**
All participating atmospheric inversions are checked for consistency with the annual global growth rate, as both are
derived from the global surface network of atmospheric CO2 observations. In this exercise, we use the conversion
factor of 2.086 GtC/ppm to convert the inverted carbon fluxes to mole fractions, as suggested by Prather (2012). This
number is specifically suited for the comparison to surface observations that do not respond uniformly, nor
immediately, to each year's summed sources and sinks. This factor is therefore slightly smaller than the GCB
conversion factor in Table 1 (2.142 GtC/ppm, Ballantyne et al., 2012). Overall, the inversions agree with the growth
rate with biases between 0.03-0.08 ppm (0.06-0.17 GtCyr$^{-1}$) on the decadal average.
The atmospheric inversions are also evaluated using vertical profiles of atmospheric $CO_2$ concentrations (Figure B4).
More than 30 aircraft programs over the globe, either regular programs or repeated surveys over at least 9 months, have
been used in order to draw a robust picture of the system performance (with space-time data coverage irregular and
denser in the 0-45°N latitude band; Table A6). The nine systems are compared to the independent aircraft $CO_2$
measurements between 2 and 7 km above sea level between 2001 and 2021. Results are shown in Figure B4, where the
inversions generally match the atmospheric mole fractions to within 0.7 ppm at all latitudes, except for CT Europe in
2011-2021 over the more sparsely sampled southern hemisphere.

**Appendix D: Processes not included in the global carbon budget**
**D.1 Contribution of anthropogenic CO and $CH_4$ to the global carbon budget**
Equation (1) includes only partly the net input of $CO_2$ to the atmosphere from the chemical oxidation of reactive
carbon-containing gases from sources other than the combustion of fossil fuels, such as: (1) cement process emissions,
since these do not come from combustion of fossil fuels, (2) the oxidation of fossil fuels, (3) the assumption of
immediate oxidation of vented methane in oil production. However, it omits any other anthropogenic carbon-containing
gases that are eventually oxidised in the atmosphere, such as anthropogenic emissions of CO and $CH_4$. An attempt is
made in this section to estimate their magnitude and identify the sources of uncertainty. Anthropogenic CO emissions
are from incomplete fossil fuel and biofuel burning and deforestation fires. The main anthropogenic emissions of fossil
$CH_4$ that matter for the global (anthropogenic) carbon budget are the fugitive emissions of coal, oil and gas sectors (see
below). These emissions of CO and $CH_4$ contribute a net addition of fossil carbon to the atmosphere.
In our estimate of $E_{FOS}$ we assumed (Section 2.1.1) that all the fuel burned is emitted as $CO_2$, thus CO anthropogenic
emissions associated with incomplete fossil fuel combustion and its atmospheric oxidation into $CO_2$ within a few



months are already counted implicitly in $E_{FOS}$ and should not be counted twice (same for $E_{LUC}$ and anthropogenic CO
emissions by deforestation fires). Anthropogenic emissions of fossil $CH_4$ are however not included in $E_{FOS}$, because
these fugitive emissions are not included in the fuel inventories. Yet they contribute to the annual $CO_2$ growth rate after
$CH_4$ gets oxidized into $CO_2$. Emissions of fossil $CH_4$ represent 30% of total anthropogenic $CH_4$ emissions (Saunois et
al. 2020; their top-down estimate is used because it is consistent with the observed $CH_4$ growth rate), that is 0.083 GtC
$yr^{-1}$ for the decade 2008-2017. Assuming steady state, an amount equal to this fossil $CH_4$ emission is all converted to
$CO_2$ by OH oxidation, and thus explain 0.083 GtC $yr^{-1}$ of the global $CO_2$ growth rate with an uncertainty range of 0.061
to 0.098 GtC $yr^{-1}$ taken from the min-max of top-down estimates in Saunois et al. (2020). If this min-max range is
assumed to be 2 σ because Saunois et al. (2020) did not account for the internal uncertainty of their min and max top-
down estimates, it translates into a 1-σ uncertainty of 0.019 GtC $yr^{-1}$.
Other anthropogenic changes in the sources of CO and $CH_4$ from wildfires, vegetation biomass, wetlands, ruminants, or
permafrost changes are similarly assumed to have a small effect on the $CO_2$ growth rate. The $CH_4$ and CO emissions
and sinks are published and analysed separately in the Global Methane Budget and Global Carbon Monoxide Budget
publications, which follow a similar approach to that presented here (Saunois et al., 2020; Zheng et al., 2019).
**D.2 Contribution of other carbonates to CO₂ emissions**
Although we do account for cement carbonation (a carbon sink), the contribution of emissions of fossil carbonates
(carbon sources) other than cement production is not systematically included in estimates of $E_{FOS,}$ except for Annex I
countries and lime production in China (Andrew and Peters, 2021). The missing processes include $CO_2$ emissions
associated with the calcination of lime and limestone outside of cement production. Carbonates are also used in various
industries, including in iron and steel manufacture and in agriculture. They are found naturally in some coals. $CO_2$
emissions from fossil carbonates other than cement not included in our dataset are estimated to amount to about 0.3%
of $E_{FOS}$ (estimated based on Crippa et al., 2019).
**D.3 Anthropogenic carbon fluxes in the land-to-ocean aquatic continuum**
The approach used to determine the global carbon budget refers to the mean, variations, and trends in the perturbation
of $CO_2$ in the atmosphere, referenced to the pre-industrial era. Carbon is continuously displaced from the land to the
ocean through the land-ocean aquatic continuum (LOAC) comprising freshwaters, estuaries, and coastal areas (Bauer et
al., 2013; Regnier et al., 2013). A substantial fraction of this lateral carbon flux is entirely 'natural' and is thus a steady
state component of the pre-industrial carbon cycle. We account for this pre-industrial flux where appropriate in our
study (see Appendix C.3). However, changes in environmental conditions and land-use change have caused an increase
in the lateral transport of carbon into the LOAC – a perturbation that is relevant for the global carbon budget presented
here.
The results of the analysis of Regnier et al. (2013) can be summarised in two points of relevance for the anthropogenic
$CO_2$ budget. First, the anthropogenic perturbation of the LOAC has increased the organic carbon export from terrestrial
ecosystems to the hydrosphere by as much as $1.0 \pm 0.5$ GtC $yr^{-1}$ since pre-industrial times, mainly owing to enhanced
carbon export from soils. Second, this exported anthropogenic carbon is partly respired through the LOAC, partly
sequestered in sediments along the LOAC and to a lesser extent, transferred to the open ocean where it may accumulate
or be outgassed. The increase in storage of land-derived organic carbon in the LOAC carbon reservoirs (burial) and in





the open ocean combined is estimated by Regnier et al. (2013) at $0.65 \pm 0.35$ GtC yr$^{-1}$. The inclusion of LOAC related
anthropogenic $CO_2$ fluxes should affect estimates of $S_{LAND}$ and $S_{OCEAN}$ in Eq. (1) but does not affect the other terms.
Representation of the anthropogenic perturbation of LOAC $CO_2$ fluxes is however not included in the GOBMs and
DGVMs used in our global carbon budget analysis presented here.
**D.4 Loss of additional land sink capacity**
Historical land-cover change was dominated by transitions from vegetation types that can provide a large carbon sink
per area unit (typically, forests) to others less efficient in removing $CO_2$ from the atmosphere (typically, croplands).
The resultant decrease in land sink, called the 'loss of additional sink capacity', can be calculated as the difference
between the actual land sink under changing land-cover and the counterfactual land sink under pre-industrial land-
cover. This term is not accounted for in our global carbon budget estimate. Here, we provide a quantitative estimate of
this term to be used in the discussion. Seven of the DGVMs used in Friedlingstein et al. (2019) performed additional
simulations with and without land-use change under cycled pre-industrial environmental conditions. The resulting loss
of additional sink capacity amounts to $0.9 \pm 0.3$ GtC yr$^{-1}$ on average over 2009-2018 and $42 \pm 16$ GtC accumulated
between 1850 and 2018 (Obermeier et al., 2021). OSCAR, emulating the behaviour of 11 DGVMs finds values of the
loss of additional sink capacity of $0.7 \pm 0.6$ GtC yr$^{-1}$ and $31 \pm 23$ GtC for the same time period (Gasser et al., 2020).
Since the DGVM-based ELUC estimates are only used to quantify the uncertainty around the bookkeeping models'
ELUC, we do not add the loss of additional sink capacity to the bookkeeping estimate.