# Peer review of "Global Carbon Budget 2022"

_Earth System Science Data, 2022_

## Referee Comment (RC2)

Review ESSD-2022-328

Thanks for opportunity to again review important document. I will certainly recommend publication.

Document has gotten longer, while time to review - esp this year - has gotten shorter. Too much to ask, even of strong advocates? Authors and journal need to consider alternate approaches!

I evaluate mostly marked changes. I read full text to understand context for changes. Line number references refer to full 2022 document.

Basic question, perhaps not answerable here but also not addressed: Why do atmos CO2 concentrations (e.g. Figure 1) rise continually apparently without any influence from substantial changes documented in Efos etc. as in Figure 3?

Technical issues/changes:

Line 159: "synthesise" - ESSD editors will know correct spelling

Lines 169 to 171: Efos = 10.1 + Eluc 1.1 = Etotal of 11.1? As in prior years, small offset must represent rounding error? Scientific readers should understand uncertainty limits (1 sigma) but casual readers will see this as math error? Also, because of qualification due to cement carbonation, does this total include or not include the carbonation term? One case Etotal = 11.0, in the other 11.2, both within uncertainty limits but both will confuse some readers?

Line 176: "50% above pre-industrial" this means pre-industrial = 270 ppm? We know this but casual readers may not. Likewise at lines 221. Not defined (277 ppm) until line 239.

Line 191: "pre-COVID" not well defined (and, in any case, not determined by carbon community). Insert '2019' so that readers will understand changes on your terms.

Lines 190 to 199: this paragraph seems slightly confusing? Efos for 2021 known (reported), now [quantified] as slightly below Efos 2019. (Why not specify Efos 2019 here?). Efos increase expected for 2022 [estimate growth], to above Efos 2019 (now Efos 2019 itemized but why here rather than earlier?).Then sectors then regions, but all as Efos 2022 estimates. Some readers may feel that authors jumped past solid 2021data?

Lines 203 to 211: now, appropriately, Eluc but on decadal rather than annual terms; tell us why? In line 202, authors revert to annual estimate. If not in fact significant, say so while omitting the numbers?

Lines 206, 207: "highlighting substantial mitigation potential" this seems like IPCC-speak. For casual readers (of Exec Summary!) call out the desired change: less logging (deforestation)?

Line 207: "sequestration of 0.9" a quantified report or an estimate, not clear.

Line 208: other 'land-use' transitions?

Line 209: now reader confronts 6 decades (1959 to 2021)? Need consistent time-scales or explicit refocussing.

Line 215: rounding errors again induce confusion. 2021 Efos + cement at 11.1 while 2022 increase to 11.1? And what happened to carbonation?

Line 217: release of IPCC AR6 WG1 in 2019? Formal citation = 2021? Please clarify?

Line 219: cumulatively for each of the next 28 years? 28*0.4 = 11.2. What (again) about carbonation? In the noise?

[Speaking personally, these numbers assume we continue in a stable economic and social system? Very unlikely? Can authors not inject some note of enhanced uncertainty?]

Line 223: following from prior paragraph, this growth rate will need to trend toward zero? But, despite quantified decreased Efos for 2020, atmos concentrations of CO2 showed no downward deflection?

Lines 224 to 230: Decadal records vs annual predictions? Can authors justify annual predictions in view of 3x uncertainty obs to models?

Lines 235, 236: If annual 1 Gt changes in Sland do not impact global atmos CO2 concentrations, same also true for Socean? What are we missing as global atmos CO2 concentrations continue to rise regardless of changes in Efos, Sland, Socean?

Line 254: parentheses rather than commas (e.g. "since the year 1750 (the pre-industrial period) and")? Authors and editors will know.

Line 302: "characterising" looks strange to my eye but authors and editors will know.

Lines 363 to 369: this paragraph implies that authors have applied carbonation corrections consistently to prior data? Not clear to this reader.

Line 403: information also for peat drainage  three independent datasets for peat drainage?

Line 545: for widely-diverging GOBM and obs-based estimates, should this be median rather than average? Or, authors already selected against extreme values?

Line 632: here, carbonation corrections applied only since 2021 in atmos inversions? Somewhere this reader would like to find a statement about when corrections applied, over what time periods, or - as minor - not applied.

Line 697: here reader finds/learns that carbonation correction applies since "1960s". Need this clarification earlier to resolve previous issues.

Line 789, 790: "emissions from organic soils contribute over proportionally to interannual variability" Something wrong somewhere?

Line 784-809: sorry, but the authors lost me completely in this discussion of Eluc. I suspect authors could rewrite at half the length with twice the clarity. Not useful as written.

Line 815: " to a substantial part for export" not clear what authors mean here? Land cleared of forest, converted to cropland, but crops then exported?

Line 829: NGHGI defined here but acronym used several time previously?

Line 863: " relatively wet dry season" I know what you mean but highly awkward as written

Lines 901 to 903: why not these lines (about unprecedented atmos $CO_2$ concentrations) in exec summary?

Page 87-88, Table 3: Excellent, should be required in any repeating global estimate in ESSD.

Page 94, Table 6: Legend includes a disclaimer about rounding to 0.1 GtC. Such a disclaimer should occur at top of manuscript text?

Page 97-98, Table 10: Interesting approach to show major uncertainties in one table. Appears much more orderly and organised here than this reader found in text?

Page 107, Figure 9: Why can't these panels appear as large and clear as panel in Figure 10?

Page 110, Figure 12: Thought-provoking. In last year's version, Efos - Gatm occurred as a broad range and doubled line diagonally across the graphic, allowing this viewer to assume annual variations. In present version, Efos - Gatm follows a single linear line without variation. I need to think whether invariant Efos - Gatm across the ranges of Sland and Socan is even possible? Shouldn't individual Efos - Gatm points vary as BIM?

Page 137, Figure B4: Good that authors include but suggests very weak relation atmospheric inversions to airborne measurements. Problem with obs or models or both? I understand why co-authors want inclusion in this notable effort but this suggests a substantive piece that could be eliminated, treated elsewhere, relegated to a supplement, etc. Really no change, and certainly no improvement, from prior version! This version could reference all of Appendix B by citation to previous rather than inclusion in every iteration?

Page 140, Appendix C, lines 59 to 69: About carbonation. Text here describes (and, to certain extent, repeats information already provided) carbonation process. Final sentence describes where carbonation processes fit in budget. Nothing here, however, about time span for applying these corrections, or of impact of leaving them out?

Other sections of Appendix C, Methods, quite necessary, particularly when external economic or social factors force change in annual estimates. Likewise for Appendix D. But, Appendix B?

---

## Referee Comment (RC3)

Review on "Global Carbon Budget 2022" by Friedlingstein submitted to ESSD
(#essd-2022-328/)

General comment

The Global Carbon Program (GCP) annually publishes a detailed analysis of the global carbon cycle budget using simulation model results, observational and statistical data. This manuscript is the latest edition of such analysis, and many parts of the methodology have been thoroughly reviewed in previous editions. The manuscript is deemed acceptable assuming that the authors provide appropriate responses to these comments.

Specific comments

1. ll.214-216: In line 190, the authors excluded cement carbonation. It is not quite clear whether the "total anthropogenic emissions" include cement carbonation or not. Please clarify.

2. section 2.5: The authors mention several times on emissions from peat fire, but not on those from natural and anthropogenic biomass burning. How are they treated in this estimate?

3. Some of the DGVMs list in Table A1 are not DGVMs in its narrow sense, i.e., models that predict the distribution of plant types. VIST, for example, deals with biomass variation with a fixed distribution of plant types. Note somewhere that the term "DGVMs" in this manuscript simply means vegetation models. Also, it would be more user-friendly if Table 4 is referred to in addition to Table A1.

4. l.562: ONI index -> The acronym ONI includes the word "index". Please just say "ONI" instead of "ONI index". There are a few other places where the same expression is used. I am afraid that the authors are well aware of this, but I presume many of the readers will feel uncomfortable with this expression.

5. l. 891: This sentence says that 30% is from LUC and 79% is from fossil fuel. The sum exceeds 100%.

6. l. 976: The text says "one new model is included" but Table 4 shows there are two models that are new this year. Perhaps the authors meant something like "one of the new models bears an estimate higher than the average"?

7. ll.1034-1035: "This suggests... by the ocean." This sentence casts doubt on the scheme adopted in the manuscript to calculate ocean uptake as an average between GOBM estimates and observation-based products. Isn't it more suitable to put some weight on the observation-based products when the authors are so sure that GOBMs underestimate the uptake? Explanation from the authors on this point would be appreciated.

8. ll. 1398-1399: It is stated that the importance of ELUC is increasing, but given the fact that the fraction of ELUC to the entire

---

## Referee Comment (RC4)

**Review of the Global Carbon Budget 2022**

Thank you for the opportunity I was offered to be reviewer and I would also like to thank the authors for this excellent study. This latest version of the global carbon budget study is a useful and comprehensive work for the carbon community. Please find below a few comments.

**General and technical comments:**

ln.333. How the added decomposition of $E_{LUC}$ into its main component improve or change the Glocal Carbon Budget?

ln.353. It should be BP energy company.

ln.401-403. these sentences could be rearrange and rewrote in one sentence. The information seem redundant.

Ln.403. 3 independent datasets for peat drainage are included. It is not well clear which are the corresponding datasets in the section 2.2.1.

ln.539. A fourth simulation has been added in this 2022 paper compared to the 2021 paper. It is not clear what the added simulation bring to the study in comparison to the previous one.

ln.617. CMS-Flux is assimilating both GOSAT and OCO-2 simultaneously. Even if these sensors have similar spectral bands, their calibration are not perform in similar ways which could bring non-negligeable biases in the inversion result. It would, hence, be interesting to know (useful for the inversion results of this manuscript) how the biases resulted from the joint GOSAT and OCO-2 assimilation was considered in the CMS-Flux inversion.

Section 3.1.1 Even though the values have been updated, the text is similar to the 2021 paper. It would have been interested, for instance, to add further information comparing the 1850-2021 (including the post-covid lock-down) and 1850-2020 (including the covid lock-down) periods. Some of the difference between 2021 and 2020 are mentioned in section 3.1.3 but this could be mentioned in section 3.1.1 as well. Ln 672, in comparison to the 1850-2020 period, the 1850-2021 one has only a decrease of 1%

from natural gas but the contribution of the other sources have not changed. Do you know if the reduction in natural gas emission is coming from a specific region or not?

ln.679. need to remove a parenthesis after Hoesly et al., 2018.

ln.773. You mention "these changes […] lead to higher net emissions in Brazil in the last decades compared to last year's GCB". It would be useful here to add some carbon emission values. How much carbon emission are you talking about?

ln.777. You mention that the increase in deforestation over Brazil and the associated carbon emission is not well capture. Do you have an estimation of how much carbon emission from the deforestation in Brazil is missing in your estimation? For future GCB, do you consider additional measurements (i.e. chlorophyll fluorescence or vegetation canopy from spaceborne platforms) to help better address and monitor deforestation related to the global carbon budget?

ln.1049-1050. Why not using an other dataset independent from the data products?

ln.1242. I could not find Section 2.7.4.

ln.1345. CO2 should be $CO_2$

---

## Author Comment (AC1)

**RC1 Ana Maria Roxana Petrescu**

The authors are to be greatly complimented on their work, for the outstanding number of data sources used, performed analysis, as well as for the continuous inclusion of new products. This series of studies represent useful resource for scientists but to a lesser extent to policymakers, given the content, depth of treatment and length. However, the authors do their best in disseminating the scientific findings and convert it into policy messages (e.g., COP meetings). A broader dissemination of results and conclusions could be done for NGOs, stakeholders and the non-expert citizens i.e., common language press releases, focusing on key findings like the reduction of the period needed to limiting global warming to 1.5 degree, reduction of fossil emissions in 24 countries etc.

*Thank you for these suggestions, most of them being already implemented in our communication strategy when the paper is released (provision of key messages from key findings, slide deck and short slide presentations, key datasets in user friendly format, data for key countries, etc).*

**General comments**

Given its increasing length, an option for future versions would be to transform it into a communication like paper, highlighting key messages and the paper as it is now to be the Supplement. The GCB it is a very well known study, and in a shorter format would beneficiate of a broader audience.

*Point taken but we want to state that for the carbon cycle community, it is important for this paper to report the latest results and key messages but also to report and to be transparent on the methodology used, assumptions made, models/data used, assessment of uncertainty, etc. We did revise the format of the paper last year and we moved most of the methodology in the supplement section (Appendix C). We don't think changing the format to a short "communication-like paper" is what the community wants, keeping in mind that we publish in Earth System Science Data.*

A great improvement is the inclusion of country level ELUC estimates and decomposition of the flux, the regional discussion, as well as the inclusion of peat degradation data sets.

*Thank you*

Do simulations and projections for 2022 take into account the long drought period in some regions?

*It does for land-use changes emissions as discussed in section 3.2.4, and also to some extent for the land and ocean sink 2022 projections as they use the ONI ENSO index as predictor.*

Why Results fossil and ELUC chapters do not have model (data sets) evaluation sections?

*For land use emissions, there are simply no independent observations of land use CO2 emissions that we could use to evaluate the models. The DGVMs land use estimate is used as independent supporting evidence of the ELUC estimate from the bookkeeping models.*

*For fossil emissions, as discussed in section 3.1.1, there are other datasets (CDIAC-FF, CEDS, and PRIMAP-hist), although not fully independent as ultimately relying on the same raw energy data. We provide a brief comparison, for the global historical emissions.*

To follow-up on a recent "VERIFY" project discussion, would the "median" instead of mean (average) be more appropriate for large ensembles (e.g. DGVMs, inversions) where the min/max show large ranges?

*Nonparametric statistics would only be relevant if the distributions of models were skewed, and significantly departed from the implicit gaussian assumption. Figure 12 suggests this is not the case for DGVMs (see green shaded distribution). For Inversion models, we present the range from the models (see Table 6), rather than the average as these estimates are only based on 7 models.*

Given the emphasis on using more and more inversions in the future using satellite data, it is great to have OCO-2 based products included in this study, however I find it poorly highlighted in the discussion.

*We indeed do not highlight this specifically in the text, as we prefer to first have a scientific highlight that derives from the inclusion of OCO-2 based inversions. This would be a stronger reason to label them great than just the fact that they are now used. With the inversion group, we expect to have a larger ensemble of OCO-2 based estimates in the future, and to specifically assess their added value for GCP.*

A more focused discussion/conclusions based on this year's budget and updates would be welcomed. Few points have been added to discussions but conclusions from previous work still repeat, giving more the feeling of "aim of the study" and "general statements" than concluding upon this year's findings. Perhaps authors could keep their general remarks and add a short and more results-focused paragraph.

*Thank you for this suggestion. We feel we already partly do what the reviewer is suggesting. Most of the new findings is the current year (2022) projection, which is already clearly highlighted (separate sections for each component). The executive summary also tries to step back and highlight the key findings from this year's manuscript (as for example the 2022 emissions, the country level land use estimates). Other changes from year to year are more minor and mainly methodological (summarised in Table 3).*

**Line by line suggestions**

Abstract: If 2020 registered a decline of 5% fossil emissions compared to pre-pandemic (2019) and in 2021 was noted an increase of 5.1% relative to 2020, I think would be good to clearly state that EFOS are almost back to the levels of 2019, with 2020 being an atypical year.

*We don't mention the 2019-2020 change in the abstract, hence it's probably not the best place to add this statement. Note that we have such a statement in the Executive Summary (first paragraph).*

L 190-191: start with "...further increased in 2021" with values and then refer to the 2022 projection

*The bold headline is about 2022, which is what really matters this year. The 2021 increase and the further 2022 increase are presented in the following sentence.*

L 191: add 2019 "...above their 2019 pre-COVID19 levels"

*Done, thank you.*

L 193: "Preliminary estimates" I think the end month for the data availability should be mentioned, is it July, August 2022?

*Data availability varies across countries and fuel type. It would be misleading to give one single month (e.g. July or August) here.*

`L 200: why is the decade till 2019 when all analysis is focuses on 2012-2021? If some data sets are not available should be mentioned.

*Sorry, that was a typo, it is 2012-2021 indeed.*

L 201: is "only" needed? A quarter or world's fossil emissions is pretty significative.
*Agreed, "only" removed, to be more neutral*

L234: why previous decade is 2000-2009? Should not be 2002-2011? for oceans we have 2011-2020, the executive summary talks about 2012- 2021.

*Across the entire manuscript (see table 5 and 6 for example) we present the last decade, i.e,the last 10 years: 2012-2021. However, when presenting previous decades we use the conventional time period (1960-1969, 1970-1979, …, 2000-2009). To avoid confusion we rephrased the sentence on the land sink as ".. larger than during the 2000-2009 decade." We also rephrased the ocean sink sentence as "The ocean CO2 sink resumed a more rapid growth in the past two decades after low or no growth during the 1991-2002 period". Thanks also for spotting that in the ocean section (line 229) the decade 2011-2020 should read 2012-2021. This typo is now corrected and the ocean sink number for this decade was correct.*

L 338: 2.1.1. the period "1850-2021" appears only here, I would suggest to add this historical period to the general paragraph in Methods or add it to all sections (2.2.1, 2.3.1 etc.) as done in the Results sections

*Done, thank you.*

L 348-350 mention the number of fossil data sets used (N=7 as in Fig 12?) Why Table 4 does not include the fossil sources? It is not clear from the main text which seven data sets are used.

*There is a confusion here. We do not have 7 datasets for fossil fuel data. The N=7 in figure 12 refers to the atmospheric inversions (described in Table 4).*

L 371: Peters et al.,

*Done, thank you*

L 539 and L1029: please add (Appendix C.3.2) for sim D

*We already refer to Appendix C3 at the end of section 2.3: "More details on the $S_{OCEAN}$ methodology can be found in Appendix C.3.". Not sure why it would be needed specifically for sim. D.*

L 582-584: 16 DGVMs in total, only 11 include the effect of N input, what happens to the other 5? Are estimates comparable? Was the N effect quantified in terms of sink between DGVMs with and without N?

*11 models represent the nitrogen cycle in addition to the carbon cycle. This is clarified now.*

*We have further analysed the 16 DGVMs grouping into -CN (11) and -C only models (4). We find that over different time horizons the NBP results are not substantially different, e.g. for the decade 2012-2021 the annual mean SLAND and ELUC for the two model groups are not significantly different (P>0.05). With a small set of -C only models it will be difficult to attribute differences to inclusion of a N-cycle alone. A more informative study would be to compare -CN vs -C only of the same models, but that is a major endeavour and beyond the scope of GCB2022. However, we are also in the process of preparing a new study on benchmarking -CN models from GCB2022.*

L 610-611: Why talking about refining, aren't the a-priori fluxes harmonized for all inversion systems? at least I would believe so when reading L 626-L 630.

*The atmospheric inversions prescribe the fossil fuel emissions, as described in that section. other apriori fluxes for land and ocean uptake are not harmonised, these are part of the specific design of each inversion model.*

L 678: three global data sets additional to which ones ?

*Clarified: "In addition to the estimates of fossil CO2 emissions that we provide here (see Methods), …"*

L 715: can you quantify the least accurate? How large were the changes?

*This is presented in Table 7, for the rest of the world, the prediction was 3.2% while the actual change was 4.5%. We now report the error in the text: "Of the regions, the projection for the 'rest of world' region was least accurate (off by -1.3%), largely because of poorly projected emissions from international transport (bunker fuels), which were subject to very large changes during this period. "*

L 774 DR Congo, L210 and L812 Democratic Republic of the Congo and (DRC) L324

*Changed all instances to "Democratic Republic of the Congo"*

L 791: "Deforestation is thus the main driver of global gross sources" – an important message to be highlighted in conclusions

*Thank you, added in the executive summary*

L 826, L 1226 and references: please update Ciais et al 2020 with Ciais et al., 2022

*Thanks, updated.*

L 939-949: how does this one model simulating a strength changes the average (N=10) when the other 9 simulate weakening?

*Sorry, there was a missing negative sign ahead of the "4.2%" value, now added. Should clarify that the effect of climate is to reduce the sink on average: "The effect of climate change is much weaker, reducing the ocean sink globally by 0.11 ± 0.09 GtC yr$^{-1}$ (-4.2%) during 2012-2021 (nine models simulate a weakening of the ocean sink by climate change, range -3.2 to -8.9% and only one model simulates a strengthening by 4.8%)"*

L 1193: are these biases known? Perhaps add few in brackets (parametrization (T), tiers?)

*The biases are not known. This is work in progress within the RECCAP (REgional Carbon Cycle Assessment and Processes) project. Model biases were investigated in Earth System Models, but the forced ocean models used here have smaller biases. This is work in progress. We have changed 'model biases' to 'potential biases', added a sentence on the Earth System Model biases, and will revisit next year.*

*"Dominant biases in Earth System Models are related to mode water formation, stratification, and the chemical buffer capacity (Terhaar et al. 2021, Bourgeois et al. 2022, Terhaar et al., 2022)."*

L 1411: Totally agree with the "pragmatic fix" in Grassi et al., shifting and adding-up numbers from different BU data sets is not a long-term solution to solve the reconciliation between BU and inventories...I believe the two perspectives should only inform/complement each other and remain two different entities.

*Thank you*

**Tables and figures**

Table 4: add if possible the fossil data sets

*Please see our response above to comments regarding L 348-350*

Table 5 there is no column 2022 (Projection)?

*Indeed, the 2022 projection of the land sink is not from DGVMs (see section 2.5.2), so we feel that including the projections here would add unnecessary complexity to the table.*

Figure 3 caption: again not clear what this mosaic of data sets is for the fossil emissions (Andrew and Peters 2021?)

*Please see our response above to comments regarding L 348-350*

Figure 12 caption: In the main text you talk about nine inversions, here the caption talks about six and the figure about seven. Also perhaps informative to add in the figure the value for the GCB grey point (in brackets).

*Thank you, there are 7 inversions shown on Figure 12, there was a typo in the caption. Note that we have 9 inversions but two inversions based on OCO-2 only cover the 2015-2021 period, hence not shown on this figure.*

Figure 15: Y-axis should be the same for all panels. Interesting to see 2009 has similar behavior as 2020 (was it a consequence of the economic recession felt strongly by developed countries (not seen much in India, China))?

*Y-axis are different because magnitude of annual changes varies widely across countries. Indeed 2008-2009 was the result of the global financial crisis, which primarily impacted USA and EU.*

Appendix C.2.4. Reference for HILDA+ (https://landchangestories.org/hildaplus/, Ganzenmüller et al. 2022)

*We do cite the paper, Ganzenmüller et al. 2022, not clear why we should also refer to the website.*

Appendix D: Can you please explain what do you mean by: "Anthropogenic emissions of fossil CH4 are however not included in EFOS, because these fugitive emissions are not included in the fuel inventories" Fugitives are reported in the CRF tables, 1B (1B1 and 1B2), see chapter 4, ipcc 2006 https://www.ipcc-nggip.iges.or.jp/public/2006gl/pdf/2_Volume2/V2_4_Ch4_Fugitive_Emissions.pdf Or are you referring to inventories as to other BU data sets.

*Clarified and simplified to: "The diffuse atmospheric source of CO2 deriving from anthropogenic emissions of fossil CH4 is not included in EFOS. In reality, the diffuse source of CO2 from CH4 oxidation contributes to the annual CO2 growth rate."*

---

## Author Comment (AC2)

**RC2 Anonymous Referee #2**

Thanks for opportunity to again review important document. I will certainly recommend publication. Document has gotten longer, while time to review - esp this year - has gotten shorter. Too much to ask, even of strong advocates? Authors and journal need to consider alternate approaches! I evaluate mostly marked changes. I read full text to understand context for changes. Line number references refer to full 2022 document. Basic question, perhaps not answerable here but also not addressed: Why do atmos $CO_2$ concentrations (e.g. Figure 1) rise continually apparently without any influence from substantial changes documented in Efos etc. as in Figure 3?

*Atmospheric $CO_2$ increases (GATM) as long as emissions are larger than sinks, see Table 6. This is the case for the whole historical period. Unclear which "substantial changes" documented in EFos the reviewer is referring to. The small decline in fossil fuel emissions in 2020 (-5%) was to small to stop the increase of atmospheric $CO_2$. The world still emitted 10.5 GtC in 2020, which is much larger than the land and ocean sinks , hence the observed increase in atmospheric $CO_2$.*

Technical issues/changes:
Line 159: "synthesise" - ESSD editors will know correct spelling

*We use British English. We will change if requested by the editor.*

Lines 169 to 171: Efos = 10.1 + Eluc 1.1 = Etotal of 11.1? As in prior years, small offset must represent rounding error? Scientific readers should understand uncertainty limits (1 sigma) but casual readers will see this as math error? Also, because of qualification due to cement carbonation, does this total include or not include the carbonation term? One case Etotal = 11.0, in the other 11.2, both within uncertainty limits but both will confuse some readers?

*Rounding errors indeed. For 2021, Efos = 10.07 (9.85 including the carbonation sink), Eluc = 1.08, Total = 11.14 (10.93 including the carbonation sink). The total reported now includes the carbonation sink (as in Table 6 for 2021), this is clarified now.*

Line 176: "50% above pre-industrial" this means pre-industrial = 270 ppm? We know this but casual readers may not. Likewise at lines 221. Not defined (277 ppm) until line 239.

*Done, and updated to 278ppm as in IPCC AR6.*

Line 191: "pre-COVID" not well defined (and, in any case, not determined by carbon community). Insert '2019' so that readers will understand changes on your terms.

*Done*

Lines 190 to 199: this paragraph seems slightly confusing? Efos for 2021 known (reported), now [quantified] as slightly below Efos 2019. (Why not specify Efos 2019 here?). Efos increase expected for 2022 [estimate growth], to above Efos 2019 (now Efos 2019 itemized but why here rather than earlier?).Then sectors then regions, but all as Efos 2022 estimates. Some readers may feel that authors jumped past solid 2021data?

*We hope it is clearer now. The headline highlights the current year (2022), without quantification. Then the paragraph first reports the 2021 value, and second reports the 2022 estimate. Both are compared to 2019, teh pre-COVID-19 conditions.*

Lines 203 to 211: now, appropriately, Eluc but on decadal rather than annual terms; tell us why? In line 202, authors revert to annual estimate. If not in fact significant, say so while omitting the numbers?

*As explained in the text, line 201-202 refers to the fossil fuel emissions over the last decade from the 28 countries where emissions decreased.*

Lines 206, 207: "highlighting substantial mitigation potential" this seems like IPCC-speak. For casual readers (of Exec Summary!) call out the desired change: less logging (deforestation)?

*Agreed, changed as : highlighting the strong potential of halting deforestation for emissions reductions.*

Line 207: "sequestration of 0.9" a quantified report or an estimate, not clear.

*Estimate from the bookkeeping models as all numbers reported here.*

Line 208: other 'land-use' transitions?

*Indeed, now added for clarity.*

Line 209: now reader confronts 6 decades (1959 to 2021)? Need consistent time-scales or explicit refocussing.

*Indeed, we now report the top 3 countries for the last decade (2012-2021)*

Line 215: rounding errors again induce confusion. 2021 Efos + cement at 11.1 while 2022 increase to 11.1? And what happened to carbonation?

*Sorry, the cement carbonation sink had been omitted by mistake. Corrected now.*

Line 217: release of IPCC AR6 WG1 in 2019? Formal citation = 2021? Please clarify?

*The IPCC AR6 WG1 was approved in August 2021, hence a 2021 publication date, but the data IPCC used for the remaining carbon budget stop in 2019. This is clarified now.*

Line 219: cumulatively for each of the next 28 years? 28*0.4 = 11.2. What (again) about carbonation? In the noise?

*Sorry, 0.4 is simply the current emissions level (10.9 GtC) divided by 28 years, in order to quantify the linear annual emission reduction needed to reach zero in 2050. Clarified now.*

[Speaking personally, these numbers assume we continue in a stable economic and social system? Very unlikely? Can authors not inject some note of enhanced uncertainty?]

*Not clear what the reviewer suggests here. We do not want to speculate on the future of the socio-economy of the World. This is beyond the scope of the Global Carbon Budget.*

Line 223: following from prior paragraph, this growth rate will need to trend toward zero? But, despite quantified decreased Efos for 2020, atmos concentrations of $CO_2$ showed no downward deflection?

*Indeed and this is to be expected from the global carbon cycle budget. See our response to the initial comment from Reviewer 2.*

Lines 224 to 230: Decadal records vs annual predictions? Can authors justify annual predictions in view of 3x uncertainty obs to models?

*Not sure we understand the question. The methodology describing the 2022 prediction and its uncertainty is described in section 2.4.2*

Lines 235, 236: If annual 1 Gt changes in Sland do not impact global atmos $CO_2$ concentrations, same also true for Socean? What are we missing as global atmos $CO_2$ concentrations continue to rise regardless of changes in Efos, Sland, Socean?

*Same response as before, we are not "missing" anything. Atmospheric $CO_2$ continue to rise because of large anthropogenic $CO_2$ emissions (EFOS). Nevertheless, we rephrased the sentence as follows: "Year to year variability in the land sink is about 1 GtC yr-1 and*

*dominates year-to-year changes in the global atmospheric CO2 concentration, making small annual changes in anthropogenic emissions hard to detect."*

Line 254: parentheses rather than commas (e.g. "since the year 1750 (the pre-industrial period) and")? Authors and editors will know.

*Done, thank you.*

Line 302: "characterising" looks strange to my eye but authors and editors will know.

*We use British English. We will change if requested by the editor.*

Lines 363 to 369: this paragraph implies that authors have applied carbonation corrections consistently to prior data? Not clear to this reader.

*As explained in that section, we take the average of the two studies available (Cao et al., 2020 and Guo et al., 2021).*

Line 403: information also for peat drainage  three independent datasets for peat drainage?

*Done, thank you*

Line 545: for widely-diverging GOBM and obs-based estimates, should this be median rather than average? Or, authors already selected against extreme values?

*We treat GOBMs and data-products as different types of data, hence we calculate the average for each data stream before taking the average of the two averages. We only reject one "extreme" data product (Watson et al), as explained in Appendix C.3.1).*

Line 632: here, carbonation corrections applied only since 2021 in atmos inversions? Somewhere this reader would like to find a statement about when corrections applied, over what time periods, or - as minor - not applied.

*As explained in the text, small differences in Fossil datasets used by the inversion models could have occured, hence they are all adjusted to ensure agreement across inversion models and also with the estimate of EFOS in this budget*

Line 697: here reader finds/learns that carbonation correction applies since "1960s". Need this clarification earlier to resolve previous issues.

*This is not what we wrote. We only report here the magnitude of the cement carbonation flux which increased "from an average of 20 MtC yr-1 (0.02 GtC yr-1) in the 1960s to an average of 200 MtC yr-1 (0.2 GtC yr-1) during 2012-2021. The actual dataset goes back to 1931.*

Line 789, 790: "emissions from organic soils contribute over proportionally to interannual variability" Something wrong somewhere?

*Sentence clarified.*

Line 784-809: sorry, but the authors lost me completely in this discussion of Eluc. I suspect authors could rewrite at half the length with twice the clarity. Not useful as written.

*Thank you, we will rewrite this section in the revised version of the manuscript.*

Line 815: " to a substantial part for export" not clear what authors mean here? Land cleared of forest, converted to cropland, but crops then exported?

*Clarified: "export of agricultural products"*

Line 829: NGHGI defined here but acronym used several time previously?

*The acronym is defined before in section 2.2.1. but we also define it at the first instance it is used in the result section.*

Line 863: " relatively wet dry season" I know what you mean but highly awkward as written

*Sorry, we can't really think of a better way to describe a dry season that is relatively wetter than average.*

Lines 901 to 903: why not these lines (about unprecedented atmos CO2 concentrations) in exec summary?

*This is taken from IPCC AR6 WG1 chapter 5. It is there for context but we don't see this as a result from the global carbon budget that would belong to the executive summary*

Page 87-88, Table 3: Excellent, should be required in any repeating global estimate in ESSD.

*Thank you. This table is always part of the GCB paper*

Page 94, Table 6: Legend includes a disclaimer about rounding to 0.1 GtC. Such a disclaimer should occur at top of manuscript text?

*Not sure what the reviewer means by "top of manuscript text". We feel this is the right place to mention rounding numbers, as done for table 5 and table 8*

Page 97-98, Table 10: Interesting approach to show major uncertainties in one table. Appears much more orderly and organised here than this reader found in text?

*Indeed, it seems appropriate to show uncertainties for each component of the budget in one table.*

Page 107, Figure 9: Why can't these panels appear as large and clear as panel in Figure 10?

*Thanks for this. The submitted high definition file has larger panels - as in Figure 10.*

Page 110, Figure 12: Thought-provoking. In last year's version, Efos - Gatm occurred as a broad range and doubled line diagonally across the graphic, allowing this viewer to assume annual variations. In present version, Efos - Gatm follows a single linear line without variation. I need to think whether invariant Efos - Gatm across the ranges of Sland and Socan is even possible? Shouldn't individual Efos - Gatm points vary as BIM?

*Indeed, last year, we had the grey range which shows the uncertainty on EFOS-GATM (note that this has nothing to do with some combination of individual SLAND and SOCEAN). We didn't show it here, but it has been added in the revised version.*

Page 137, Figure B4: Good that authors include but suggests very weak relation atmospheric inversions to airborne measurements. Problem with obs or models or both? I understand why co-authors want inclusion in this notable effort but this suggests a substantive piece that could be eliminated, treated elsewhere, relegated to a supplement, etc. Really no change, and certainly no improvement, from prior version! This version could reference all of Appendix B by citation to previous rather than inclusion in every iteration?

*This figure is not showing relations, but biases in mole fractions. Biases are generally within 0.5ppm which is considered quite impressive for a comparison to independently gathered observations in a different part of the atmosphere. Such an "anchor" is deemed highly necessary in inverse modelling, and also used to evaluate the OCO-2 satellite inversions that the reviewer specifically highlighted in the review. We furthermore note that this figure is already part of the supplementary section, and the inversion evaluation is discussed in supplementary section C.5.2, not in the main text. We updated the reference to this discussion in Section 2.6.*

Page 140, Appendix C, lines 59 to 69: About carbonation. Text here describes (and, to certain extent, repeats information already provided) carbonation process. Final sentence

describes where carbonation processes fit in budget. Nothing here, however, about time span for applying these corrections, or of impact of leaving them out? Other sections of Appendix C, Methods, quite necessary, particularly when external economic or social factors force change in annual estimates. Likewise for Appendix D. But, Appendix B?

*Cement carbonation: there was indeed some repetition with the description in the main text (section 2.1). This has been updated now.  Time span is also mentioned now (since 1931)*

---

## Author Comment (AC3)

**RC3 Michio Kawamiya**

General comment

The Global Carbon Program (GCP) annually publishes a detailed analysis of the global carbon cycle budget using simulation model results, observational and statistical data. This manuscript is the latest edition of such analysis, and many parts of the methodology have been thoroughly reviewed in previous editions. The manuscript is deemed acceptable assuming that the authors provide appropriate responses to these comments.

Specific comments

ll.214-216: In line 190, the authors excluded cement carbonation. It is not quite clear whether the "total anthropogenic emissions" include cement carbonation or not. Please clarify.

*We now always included cement carbonation in the global EFOS estimate*

section 2.5: The authors mention several times on emissions from peat fire, but not on those from natural and anthropogenic biomass burning. How are they treated in this estimate?

*Natural wildfires are part of the natural sink terms (SLAND) as estimated by the DGVMs, as was mentioned in l. 878. However, not all DGVMs simulate fires, so our confidence in assessing this specific contribution would be low. Note that there is the FireMIP activity that assesses DGVMs wildfires. Anthropogenic biomass burning is included in the processes covered by the bookkeeping models.*

Some of the DGVMs list in Table A1 are not DGVMs in its narrow sense, i.e., models that predict the distribution of plant types. VIST, for example, deals with biomass variation with a fixed distribution of plant types. Note somewhere that the term "DGVMs" in this manuscript simply means vegetation models. Also, it would be more user-friendly if Table 4 is referred to in addition to Table A1.

*Yes the reviewer is correct that there is no one single definition of DGVM. The term is now used as generic, encompassing sense. For example, a model with seasonal phenology could be considered "dynamic vegetation". Furthermore we impose changing vegetation fractions (via LUH2), so vegetation fractions are dynamic in a sense. Hence to avoid confusion we would like to keep the use of DGVM, but we added the following sentence to Table A1. "Here we use the term "DGVM" in the broadest sense in terms of global vegetation models which are able to dynamically adjust to imposed LULCC."*

l.562: ONI index -> The acronym ONI includes the word "index". Please just say "ONI" instead of "ONI index". There are a few other places where the same expression is used. I am afraid that the authors are well aware of this, but I presume many of the readers will feel uncomfortable with this expression.

*Thank you, corrected now.*

l. 891: This sentence says that 30% is from LUC and 79% is from fossil fuel. The sum exceeds 100%.

*Typo, sorry, It it 30% and 70%. Corrected now.*

l. 976: The text says "one new model is included" but Table 4 shows there are two models that are new this year. Perhaps the authors meant something like "one of the new models bears an estimate higher than the average"?

*Good catch, thank you. Rephrased to: "because two new models are included (CESM2, MRI)". Both new models have higher than average CO2 uptake.*

ll.1034-1035: "This suggests... by the ocean." This sentence casts doubt on the scheme adopted in the manuscript to calculate ocean uptake as an average between GOBM estimates and observation-based products. Isn't it more suitable to put some weight on the observation-based products when the authors are so sure that GOBMs underestimate the uptake? Explanation from the authors on this point would be appreciated.

*This is an excellent question. The GOBMs likely underestimate the CO2 uptake by about 10% (this number was now added to the text). However, the pCO2-based data-products have large uncertainties, too; see discussion in sections 3.5.2 and 3.7.3, and Table 10. In fact, these may be even larger than the 10% underestimation by the GOBMs, but this is work in progress. We will be able to give more information on this in the next year. In view of the uncertainties in both data streams, taking the average of both ensembles seems the best approach at present.*

ll. 1398-1399: It is stated that the importance of ELUC is increasing, but given the fact that the fraction of ELUC to the entire GHGs emission is decreasing, the statement sounds somewhat contra-intuitive. Explanation on this point would be appreciated.

*That sentence refers to the climate mitigation discussions, where land based mitigations are gaining more and more interest (despite the fact that ELUC is getting smaller relative to EFOS as mentioned by the reviewer).*

---

## Author Comment (AC4)

**RC4 Hélène Peiro**

Thank you for the opportunity I was offered to be reviewer and I would also like to thank the authors for this excellent study. This latest version of the global carbon budget study is a useful and comprehensive work for the carbon community. Please find below a few comments.

*Thank you.*

ln.333. How the added decomposition of ELUC into its main component improve or change the Global Carbon Budget?

*It doesn't change the Global Carbon Budget in the sense that ELUC is still the net flux seen by the atmosphere and used in our global budget (see equation 1 in the Introduction). However, the new decomposition of ELUC allows us to better distinguish sources due to direct deforestation or to organic soil carbon loss (in peatlands) from. sinks on forest land such as through afforestation/reforestation.*

ln.353. It should be BP energy company.

*Done*

ln.401-403. these sentences could be rearrange and rewrote in one sentence. The information seem redundant.

*Not clear what information was redundant, but we slightly rephrased to shorten the sentence.*

Ln.403. 3 independent datasets for peat drainage are included. It is not well clear which are the corresponding datasets in the section 2.2.1

*These are described in the appendix, section C.2.1 (which is referenced at the end of Sec. 2.2 for further details), but we also added the reference here.*

ln.539. A fourth simulation has been added in this 2022 paper compared to the 2021 paper. It is not clear what the added simulation brings to the study in comparison to the previous one.

*As explained in the text, the 4th simulation (sim D) is used to compare the change in anthropogenic carbon inventory in the interior ocean (sim A minus sim D) to the observational estimate of Gruber et al. (2019). See also section 3.5.5 on models evaluation.*

ln.617. CMS-Flux is assimilating both GOSAT and OCO-2 simultaneously. Even if these sensors have similar spectral bands, their calibration are not perform in similar ways which could bring non- negligeable biases in the inversion result. It would, hence, be interesting to know (useful for the inversion results of this manuscript) how the biases resulted from the joint GOSAT and OCO-2 assimilation was considered in the CMS-Flux inversion.

*CMS-Flux indeed assimilates GOSAT for the period of 2010-2014, and OCO-2 for the time period of 2015-2021. But they are not assimilated simultaneously. When compared to independent observations, there is no obvious differences in performance during the GOSAT time period and OCO-2 time period (Figure 9 in Liu et al., 2021 and Figure B4 in this manuscript), which indicates that there is no obvious bias between GOSAT and OCO-2, at least at larger scale. Furthermore, both GOSAT and OCO-2 retrievals used in CMS-Flux were generated using the same retrieval ACOS retrieval algorithm and validated against the same TCCON observing network. When validating against TCCON observations, both GOSAT and OCO-2 retrievals have mean bias ~0.1-0.2 ppm, and RMS ~1.0ppm (Figure 8 in Tayler et al., 2022.*

Section 3.1.1 Even though the values have been updated, the text is similar to the 2021 paper. It would have been interested, for instance, to add further information comparing the 1850-2021 (including the post-covid lock-down) and 1850-2020 (including the covid lock-down) periods. Some of the difference between 2021 and 2020 are mentioned in section 3.1.3 but this could be mentioned in section 3.1.1 as well.

*Section 3.1.1 is about the full historical period (starting from 1850), not about specific years. We have a dedicated section 3.1.3 on the year 2021.*

Ln 672, in comparison to the 1850-2020 period, the 1850-2021 one has only a decrease of 1% from natural gas but the contribution of the other sources have not changed. Do you know if the reduction in natural gas emission is coming from a specific region or not?

*This 1% changes is due to rounding errors and minor revisions in the annual estimates of fossil fuel components, it is not due to the addition of one year.*

ln.679. need to remove a parenthesis after Hoesly et al., 2018.

*Done, thank you*

ln.773. You mention "these changes [...] lead to higher net emissions in Brazil in the last decades compared to last year's GCB"        . It would be useful here to add some carbon emission values. How much carbon emission are you talking about?

*We have now added the values of GCB2021 in comparison to GCB2022.*

ln.777. You mention that the increase in deforestation over Brazil and the associated carbon emission is not well capture. Do you have an estimation of how much carbon emission from the deforestation in Brazil is missing in your estimation? For future GCB, do you consider additional measurements (i.e. chlorophyll fluorescence or vegetation canopy from spaceborne platforms) to help better address and monitor deforestation related to the global carbon budget?

*On the first question: We cannot quantify differences to the cited Silva Junior et al study, since that study and the land use forcing underlying the GCB simulations consider different types of land use/cover changes. We therefore removed this sentence. On the second question: We re-entered information from last year's budget where we discussed that a comparison against Earth observation data is not directly possible.*

ln.1049-1050. Why not using an other dataset independent from the data products?

*Fair point. We may add comparison to pCO2 calculated from GLODAP DIC and Alkalinity, for example, as in Gregor et al., 2019 (https://doi.org/10.5194/gmd-12-5113-2019) in the future. However, the data products are all fairly close to each other, independent of the evaluation data set used, and not much information would be added by adding other datasets. SOCAT is the data set which has by far the largest number of observations and is thus the evaluation dataset of choice for the GOBMs. We thus prefer to use SOCAT in order to compare GOBMs and data-products in one figure.*

ln.1242. I could not find Section 2.7.4.

*Sorry, it should read Appendix D4. Corrected now.*

ln.1345. CO2 should be $CO_2$

*Done, thank you*

---

## Author Comment (AC5)

**RC5 Damon Mathews**

Overall, this paper continues to improve and the authors are to be congratulated for this sustained and important effort. I have only a couple of minor comments (mostly since I did not have time to more thoroughly review the entire manuscript).

1) Cumulative emissions since 1850:

On p9 the cumulative emissions since 1850 are described as aligning with the IPCC AR6 pre-industrial period, but this is not correct, since technically the pre-industrial reference period for global temperature is the 1850-1900 average (not 1850 specifically). So to align precisely with this, the cumulative emissions should be reported similarly. A quick option would to report cumulative emissions since 1975 (the midpoint of the pre-industrial period). Better however (if we want to be exactly precise) the total since the 1850-1900 reference period could be calculated as (cumulative total since 1850) - (average of cumulative emissions since 1850 for each year from 1850 to 1900). Providing this number would also be helpful for estimating the TCRE and remaining carbon budget based on historical observations, since these are the cumulative emissions that are associated with warming since the pre-industrial period, whereas cumulative emissions since 1850 would be associated with warming since some period entered on the year 1850.

*Not clear which sentence the reviewer refers to as there is no mention of IPCC on page 9. Maybe the reviewer meant page 7 where we wrote: "Finally, it provides cumulative emissions from fossil fuels and land-use change since the year 1750, the pre-industrial period; and since the year 1850, the reference year for historical simulations in IPCC AR6 (Eyring et al., 2016).". This sentence is correct, the historical simulations from CMIP6 did start in 1850. This has nothing to do with the 1850-1900 reference for temperature.*

2) National land-use CO2 emissions

The text and figures speak to national-level estimates of CO2 emissions from LULCC — this is a great addition to the carbon budget, and would be important to include if possible in the national emissions datafile that is produced. At the moment, the list of data products in this file (National_Carbon_Emissions_2022v0.1.xlsx) only include national FF emissions. Can you add a sheet to this file that gives the accompanying national land-use CO2 emissions?

*Excellent suggestion. Consider it done !*

3) Atmospheric CO2 growth rate

I have been asked many times by media why CO2 concentrations continued to grow in 2020 despite decreased CO2 emissions ... of course the answer is obvious, but nevertheless remains a question that many people ask. Might be helpful to address this specifically in the executive summary, as well as elsewhere is the manuscript -- for example, can the drop in atmospheric growth rate in 2020 be attributed to the drop in emissions? Maybe not (given other contributing factors) but in theory, all else being equal, the atmospheric growth rate should be roughly proportional to emissions, which could be highlighted for public communication purposes.

*We sympathise with the reviewer, this is one of the many common misunderstandings about the carbon cycle. There are (at least) two issues here. First issue is a confusion between*

*change in emissions and change in concentrations. An emission decrease of X% would translate into an atmospheric CO2 growth rate decrease of X% (everything else being equal), NOT into an atmospheric CO2 decrease of X%. Second issue is the natural year to year variability of the land carbon cycle which is significantly larger than any past changes in annual emissions. Even the 2020 decrease of 5.4% (about 0.5GtC) is smaller than the natural year to year swings in the land carbon sink (see Figure 4 panels b and d).*

*On the first issue, we feel there is not much we can do. The whole paper (ex figures 2, 3, 4, 14, Table 6, and related text) makes it clear that the global carbon budget is defined as: atmospheric CO2 **growth rate** = sources minus sinks (equation 1). On the more specific second issue (can we detect a change in emissions?), we changed the last sentence of the executive summary to make it clearer: " Year to year variability in the land sink is about 1 GtC yr-1 and dominates the year-to-year changes in the global atmospheric CO2 concentration, implying that small annual changes in anthropogenic emissions (such as the fossil fuel emission decrease in 2020) are hard to detect in the atmospheric CO2 observations."*